# SEGMENT ANY EVENTS WITH LANGUAGE

**Seungjun Lee,   Gim Hee Lee**
Department of Computer Science, National University of Singapore
seungjun.lee@u.nus.edu, gimhee.lee@nus.edu.sg
https://0nandon.github.io/SEAL

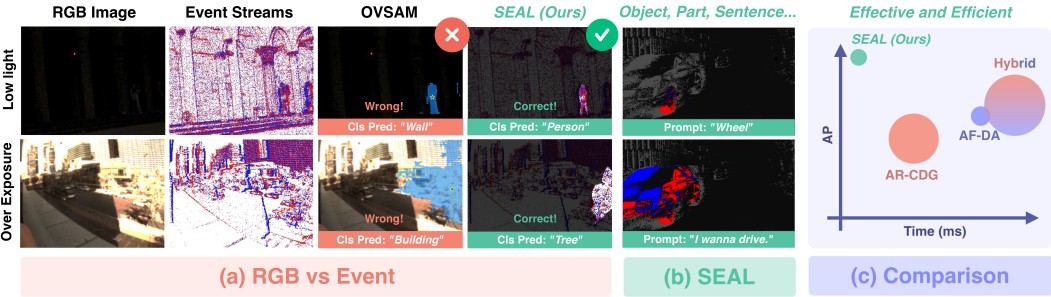

Figure 1: **a)** Image-based models are vulnerable to severe image degradation, while event-based models remain robust by leveraging event inputs. **b)** Our SEAL effectively recognizes both *part*- and *object*-level instances on both *noun*- and *sentence*-level text queries. **c)** Our SEAL outperforms existing methods in performance and inference speed with paremeter-efficient architecture.

## ABSTRACT

Scene understanding with free-form language has been widely explored within diverse modalities such as images, point clouds, and LiDAR. However, related studies on event sensors are scarce or narrowly centered on semantic-level understanding. We introduce **SEAL**, the first Semantic-aware Segment Any Events framework that addresses Open-Vocabulary Event Instance Segmentation (OV-EIS). Given the visual prompt, our model presents a unified framework to support both event segmentation and open-vocabulary mask classification at multiple levels of granularity, including instance-level and part-level. To enable thorough evaluation on OV-EIS, we curate four benchmarks that cover *label granularity* from coarse to fine class configurations and *semantic granularity* from instance-level to part-level understanding. Extensive experiments show that our SEAL largely outperforms proposed baselines in terms of performance and inference speed with a parameter-efficient architecture. In the Appendix, we further present a simple variant of our SEAL achieving generic spatiotemporal OV-EIS that does not require any visual prompts from users in the inference. The code will be publicly available.

## 1 INTRODUCTION

Event cameras, also known as bio-inspired vision sensors, differ fundamentally from conventional frame-based cameras by offering compelling advantages including unparalleled temporal resolution (Rebecq et al., 2019; Scheerlinck et al., 2020), ultra-low latency (Dimitrova et al., 2020; Lee & Lee, 2025), wide dynamic range (Gallego et al., 2020; Messikommer et al., 2022b; Schiopu & Bilcu, 2023) and commendable energy efficient (Ramesh et al., 2020). As illustrated in Fig. 1a, these advantages can mitigate false predictions of image-based models by addressing the inability of frame-based cameras to capture meaningful information in challenging conditions such as low light or overexposure.

Aforementioned competency of event cameras has motivated community to explore robust scene-understanding with events. Especially, event-based segmentation has become a key task in event vision (Alonso & Murillo, 2019; Biswas et al., 2024; Hamaguchi et al., 2023; Sun et al., 2022), where the typical challenges of image segmentation (Chen et al., 2018; 2017; Cordts et al., 2016; He et al., 2016; Long et al., 2015) are combined with additional complexities arising from the unique characteristics of event streams (Alonso & Murillo, 2019). Earlier works on Event Semantic Segmentation (ESS) has achieved significant success by using either densely annotated event labels (Alonso & Murillo, 2019; Sun et al., 2022) or employing unsupervised domain adaptation, where image-domain

knowledge is transferred to event data without using the dense event annotations (Gehrig et al., 2020; Sun et al., 2022; Messikommer et al., 2022a). Despite their effectiveness, they show two limitations: 1) Their inference is limited to a *closed*-set vocabulary which restricts their real-world application. 2) They cannot perform instance-level recognition since they focus on semantic segmentation.

Recently, several works have sought to overcome these limitations by distilling the feature space of large image foundation models (Radford et al., 2021; Kirillov et al., 2023) into event-based models or by proposing the new benchmarks for instance-level event segmentation (Hamann et al., 2024; 2025). Kong et al. (2024) proposes a novel framework for Open-Vocabulary ESS (OV-ESS) by transferring knowledge from CLIP (Radford et al., 2021) to an event backbone. However, they are limited to semantic segmentation and thus struggle to distinguish individual object instances. Chen et al. (2024) proposes a Segment Anything model for the event modality that enables instance-level event segmentation. Nonetheless, they lack the ability to recognize the semantics of events since their focus remains on class-agnostic segmentation. Hamann et al. (2024; 2025) proposes novel benchmark for spatiotemporal and instance-level segmentation for events. However, their dataset is restricted to a single class label, which falls short of the real-world requirement to recognize diverse semantics.

In this paper, we propose **SEAL**: **S**egment any **E**vents with **A**ny **L**anguage to circumvent the aforementioned issues by addressing the Open-Vocabulary Event Instance Segmentation (OV-EIS) task. There are three main objectives of our SEAL: 1) Recognize event instances across multiple levels of semantic granularity that include both *part-level* and *object-level* proposals. 2) Support free-form language queries that include but are not limited to *noun-level* and *sentence-level* expressions shown in Fig. 1b. 3) Adopt a parameter-efficient architecture with fast inference to align with the low-power consumption and high temporal resolution nature of event data. To this end, we design a Semantic-aware Segment Any Events model by introducing two key components: 1) The **Multimodal Hierarchical Semantic Guidance (MHSG)**, which is a novel multimodal learning framework that aims to learn semantic-rich event representations across multiple levels of granularity by exploiting the knowledge of large vision-language models (Radford et al., 2021; Yuan et al., 2024b). 2) To effectively learn the foundational knowledge of MHSG with parameter-efficient architecture, a **Multimodal Fusion Network** is built on top of the Segment Any Events model (Chen et al., 2024). It comprises three main components: 1) The **Backbone Feature Enhancer** explicitly integrates the language-derived semantic priors into event-domain features for better alignment between event representations and text features. 2) The **Spatial Encoding** module incorporates spatial priors derived from the foundation image segmentation model (Kirillov et al., 2023) to obtain a more discriminative event feature space. 3) The **Mask Feature Enhancer** finally refines both semantic and spatial priors encoded in the learned event features to further enhance the open-vocabulary recognition capabilities.

Since there is no existing benchmark with multiple semantics available for EIS, we further propose four benchmarks to evaluate the open-vocabulary capabilities on the diverse settings of: *label granularity* (coarse to fine-grained class annotations) and *semantic granularity* (instance-level to part-level segmentation) (*cf.* Fig. 5b). Quantitative results show that our SEAL significantly outperforms the proposed baselines across all evaluation settings and exhibits the highest efficiency in terms of inference time with efficient parameter size (*cf.* Fig. 1c). In summary, our contributions are as follows:

- We introduce SEAL, a Semantic-aware Segment Any Events framework, which is capable of generating open-world semantic predictions for event masks across the multiple levels of granularity including instance-level and part-level.

- To the best of our knowledge, this work represents the first attempt to address OV-EIS that supports free-form of language queries including noun-level and sentence-level expressions.

- We propose a novel multimodal learning framework named as MHSG, and a light-weight multimodal fusion network to effectively learn open-vocabulary capabilities.

- We propose four benchmarks to evaluate our SEAL on diverse settings. Quantitative results show that our SEAL achieves the best results in terms of effectiveness and efficiency.

## 2   RELATED WORKS

**Event-based Segmentation.** Most literature on event segmentation has focused on Event Semantic Segmentation (ESS), which assigns semantic labels to events for enhanced scene understanding. Alonso & Murillo (2019) introduces the first ESS benchmark on DDD17(Binas et al., 2017), followed

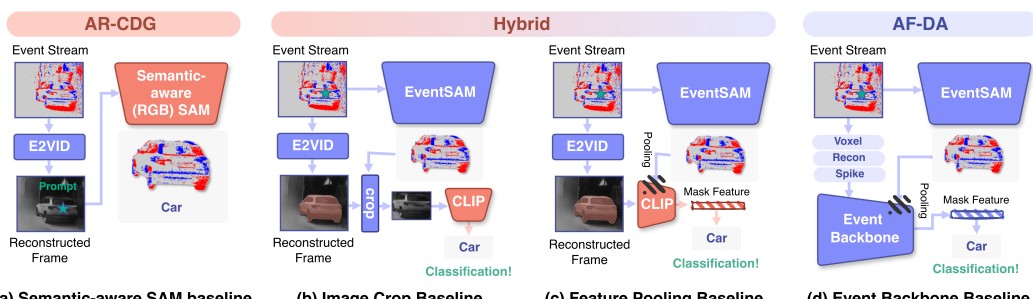

Figure 2: **Four Types of Baseline under Three Categories.** The three categories (*AR-CDG*, *Hybrid*, *AF-DA*) are designed as a baseline based on the strategy of transferring image-domain knowledge to the event domain. Refer to Sec. 3 and Supplementary material Sec. A.3 for further details.

by several methods that aim to improve accuracy while reducing the need for dense event labels. Specifically, approaches such as EvDistill (Wang et al., 2021b) and DTL (Wang et al., 2021a) leverage image frames aligned with events, while EV-Transfer (Messikommer et al., 2022a) and ESS (Sun et al., 2022) use unsupervised domain adaptation to transfer knowledge from image datasets to events. More recent work such as HALSIE (Biswas et al., 2024) and HMNet (Hamaguchi et al., 2023) explore cross-domain synthesis and memory-based encoding to further enhance mask accuracy. Another direction of research investigates the spiking neural networks for energy-efficient ESS (Che et al., 2022; Kim et al., 2022b; Neftci et al., 2019; Wu et al., 2018). Recently, EventSAM (Chen et al., 2024) introduces the Segment Anything Model (SAM) in the event modality to address class-agnostic event instance segmentation. In this paper, we extend this direction to tackle OV-EIS by designing a novel Semantic-aware Segment Any Events model.

**Open-World Events Understanding.** Recent advancements in open-world image understanding (Zhou et al., 2022; Ghiasi et al., 2022; Liang et al., 2023; Xu et al., 2023; Zhang et al., 2023) using multimodal foundation models (Radford et al., 2021; Jia et al., 2021) have inspired several works to extend this trend into the event modality. Specifically, EventCLIP (Wu et al., 2023) employs an adapter to align event features with CLIP (Radford et al., 2021), and EventBind (Zhou et al., 2024) jointly aligns event, image and text features. Ev-LaFOR (Cho et al., 2023) introduces category-guided attraction loss and category-agnostic repulsion loss to improve the alignment between the event features and CLIP. Another line of works (Li et al., 2025; Liu et al., 2025) leverage Multi-modal Large Language Model (MLLM) to understand event streams with natural language while they lack thorough exploration of pixel-level understanding such as object detection or segmentation. To this end, recent work OpenESS (Kong et al., 2024) have addressed generalizable event understanding at pixel-level granularity, termed OV-ESS, by proposing a superpixel-to-event representation learning framework. In this paper, we push the boundaries of open-world event understanding from the semantic level to the instance level by introducing a new task called OV-EIS. Our annotation-free framework enables the model to recognize individual instances across multiple levels of granularity in response to free-form language queries.

## 3 Preliminaries and Baselines.

**Event Representations and Images.** Event streams $\epsilon$ are encoded with pixel coordinates $(\mathbf{x}, \mathbf{y})$, a microsecond-level timestamp $t$, and polarity $p \in -1, +1$ that indicates brightness changes. An event $\mathbf{e} \in \epsilon$ occurs when the absolute difference in logarithmic brightness between consecutive timestamps exceeds a threshold. To process raw events $\epsilon_i$ with a neural network (Gallego et al., 2020), they are converted into regular representations $I^{evt} \in \mathbb{R}^{C \times H \times W}$. The input dimension $C$ varies by representation type: spatiotemporal voxel grids (Gehrig et al., 2019; Zihao Zhu et al., 2018; Zhu et al., 2019), frame-like reconstructions (Rebecq et al., 2019), or bio-inspired spikes (Kim et al., 2022b). Spatial alignment and temporal synchronization between events and images $I^{img}$ captured by conventional cameras enable the formation of event-image pairs.

**Open-Vocabulary Event Segmentation.** In contrast to image datasets, large-scale event–image pairs with dense pixel-level annotations are not yet widely available. To address this gap, OpenESS (Kong et al., 2024) introduces an *annotation-free* framework for OV-ESS that learns semantically rich event representations through vision–language foundation models such as CLIP (Radford et al., 2021). The event backbone is trained to align its feature space with the visual backbone of CLIP using paired event-image inputs. OpenESS also provides text-domain guidance by generating the pseudo

semantic labels with leveraging dataset-specific class lists and CLIP zero-shot classification. However, OpenESS has several limitations: 1) It lacks the ability to recognize multiple instances individually since it focuses on semantic segmentation. 2) It relies on a closed-set of predefined class candidates for text-domain guidance, which limits the open-vocabulary capabilities of the model.

**Segment Any Events.** EventSAM (Chen et al., 2024) is the first to adapt SAM for event streams to achieve EIS. Specifically, EventSAM learns an event backbone compatible with the original SAM mask decoder in an annotation-free manner by aligning its feature space with the SAM image backbone. Despite its effectiveness, EventSAM only generates class-agnostic masks and lacks the ability to recognize the semantics of the obtained proposals.

**Baselines.** Based on the preliminaries, we introduce four baselines that are grouped into three categories to address OV-EIS. Further details for baselines are in the Supplementary material Sec. A.3.

1. **Annotation-Rich Cross-Domain Generalization (*AR-CDG*):** This category focuses on cross-domain generalization by evaluating the direct transfer of pretrained image segmentation models to event streams. As shown in Fig. 2a, the **Semantic-aware SAM baseline** exemplifies this approach by leveraging SAM-based models (Yuan et al., 2024a; Li et al., 2023; Han et al., 2025; Wang et al., 2024; Chen et al., 2023) that are pre-trained on large image datasets with human annotations. To adapt event streams for these image-based models, the event data is first reconstructed into grayscale images using E2VID (Rebecq et al., 2019) and are subsequently used for inference.

2. **Annotation-Free Domain Adaptation (*AF-DA*):** This category focuses on adapting neural networks to event streams without using dense human annotations since labeling events is inherently challenging. Fig. 2d illustrates the **Event backbone baseline** in this category, where a pretrained EventSAM acts as a class-agnostic mask generator. The event backbone is learned as an open-vocabulary mask classifier via the OpenESS (Kong et al., 2024) framework while text-domain supervision is excluded based on the assumption that dataset-specific class names should also be inaccessible to better match the annotation-free setting. During inference, mask proposals from EventSAM are passed to the event backbone, where the CLIP-aligned mask features are pooled for classification.

3. **Hybrid (*AR-CDG* + *AF-DA*):** This category combines the AF-DA mask generation approach with the AR-CDG mask classification strategy. Specifically, EventSAM first generates masks that are then classified using CLIP (Radford et al., 2021) or its variants (Liang et al., 2023; Ghiasi et al., 2022; Zeng et al., 2024; Zhou et al., 2022; Li et al., 2024) which are pre-trained on large-scale image datasets with human annotations. We design two baselines in this category:

   • **Image Crop Baseline** (Fig. 2b): The E2VID-reconstructed frame is cropped using the predicted mask and feature of cropped region is extracted using CLIP or its variant (Liang et al., 2023).
   • **Feature Pooling Baseline** (Fig. 2c): Mask features are directly pooled from the pixel-wise feature maps of CLIP-based models (Ghiasi et al., 2022; Zeng et al., 2024; Zhou et al., 2022; Li et al., 2024) which are specifically designed for dense prediction. In particular, events are reconstructed into grayscale images prior to classification.

**Limitations of Baselines.** Although the proposed baselines achieve open vocabulary inference on events to some extent, they exhibit several limitations: 1) The AR-CDG and hybrid categories still suffer from a huge domain gap between image and events despite the events are reconstructed to a grayscale frame. This is because the reconstruction process often adds noise and artifacts, especially under fast motion or low event rates (Kong et al., 2024; Chen et al., 2024), driving the domain gap with natural images. 2) All baselines require two distinct backbones for mask generation and classification which degrades parameter and inference efficiency. Our work aims to address these challenges through a unified and efficient framework.

## 4    OUR METHOD

**Overview.** The overall architecture of our SEAL is illustrated in Fig. 3. Our SEAL falls under the *AF-DA* category, where only event-image pairs $(I_i^{evt}, I_i^{img})$ are used during training without access to dense event labels. During inference, the model takes the event embeddings $I_i^{evt}$ as input and generates mask and class predictions based on the visual prompts $P$ provided by the user. We design the **Multimodal Hierarchical Semantic Guidance (MHSG)** module to learn rich open-vocabulary event representations across multiple levels of granularity by exploiting the vision-language foundation models (Radford et al., 2021; Yuan et al., 2024b) (*cf.* Sec. 4.1). To

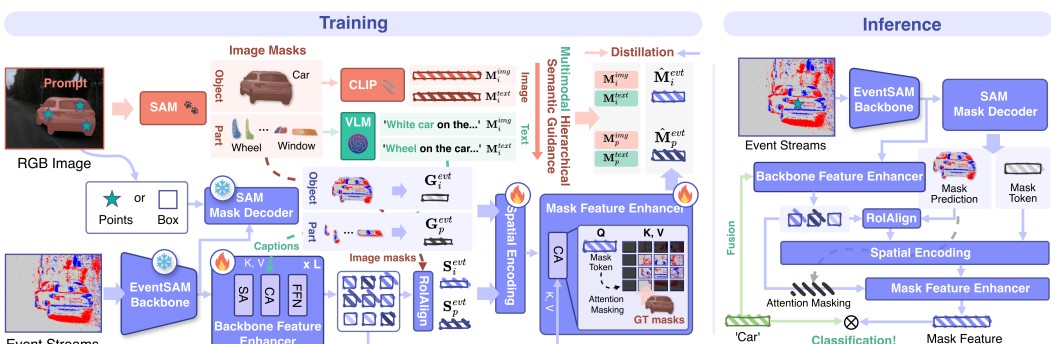

Figure 3: **Overall framework of SEAL.** The MHSG module (*Red* and *Green*) provides rich multimodal semantic guidance across multiple levels of granularity including *part*-level and *instance*-level. The multimodal fusion network of SEAL (*Purple*) enhances class prediction from a given binary mask by encoding rich *semantic* and *spatial* priors to produce a CLIP-aligned mask feature.

effectively transfer the foundational knowledge into our SEAL with parameter-efficient architecture, we introduce a light-weight **Multimodal Fusion Network** that consists of three main components: 1) Backbone Feature Enhancer, 2) Spatial Encoding, and 3) Mask Feature Enhancer (*cf.* Sec. 4.2). The training details are explained in Sec. 4.3.

## 4.1 MULTIMODAL HIERARCHICAL SEMANTIC GUIDANCE (MHSG)

In this section, we introduce MHSG module which serves as rich multimodal guidance to our SEAL, learning open-vocabulary event representations across multiple levels of granularity.

**Hierarchical Visual Guidance.** Learning the rich event representations with multiple granularity levels is challenging due to its sparse and asynchronous nature (Kong et al., 2024). To address this challenge, we leverage images $I^{img}$ and SAM (Kirillov et al., 2023) to group pixels into conceptually meaningful regions under hierarchical semantic structure. SAM can generate three different masks for a single-point prompt, where each corresponds to a different level of semantic granularity. We use the inherent property of SAM to generate three segmentation maps: $M_s^{img}$, $M_i^{img}$, and $M_p^{img}$. These maps capture the hierarchical semantic structure of $I^{img}$ at three levels: *semantic*, *instance*, and *part*, respectively.

Specifically, $M_l^{img} = \{m_l^1, m_l^2, \cdots, m_l^{K_l}\}$, where $K_l$ denotes the number of masks in $l \in \{s, i, p\}$ level of granularity. These generated masks satisfy: $m_l^1 \cup m_l^2 \cup \cdots \cup m_l^{K_l} = \{1, 2, \cdots, H \times W\}$, where $H \times W$ denotes the size of $I^{img}$. Subsequently, we get the visual features for each mask by pooling pixel-wise CLIP features: $\mathbf{I}^{img} = \mathcal{F}^{clip}(I^{img}) \in \mathbb{R}^{D_2 \times H_2 \times W_2}$. Finally, these CLIP mask features: $\mathbf{M}_l^{img} = \{\mathbf{v}_l^1, \mathbf{v}_l^2, \cdots, \mathbf{v}_l^{K_l}\}$ at multiple levels of granularity $l \in \{s, i, p\}$ serve as hierarchical visual guidance with rich open-vocabulary knowledge.

**Hierarchical Text Guidance.** Although visual guidance already provides rich semantic priors, the integration of text-domain supervision is essential to improve the responsiveness of the model to free-form language (Lee et al., 2024). To this end, we leverage LLaMA (Touvron et al., 2023)-based MLLM (Yuan et al., 2024b) to generate rich captions for each mask by feeding $I^{img}$ and $M_l^{img}$ as input. Each caption is passed to the text encoder of CLIP for the construction of the hierarchical text guidance: $\mathbf{M}_l^{text} = \{\mathbf{t}_l^1, \mathbf{t}_l^2, \cdots, \mathbf{t}_l^{K_l}\}, l \in \{s, i, p\}$.

**Discussion.** Our MHSG differs from OpenESS as follows: 1) OpenESS groups the pixels into coarse part-level instances with *single* granularity. In contrast, our MHSG defines the region into *hierarchical* semantic structure that enables the model to understand scenes with diverse levels of semantic granularity. 2) Compared to OpenESS, we do not use predefined class candidates given by the dataset to fully model annotation-free setting. Instead, we leverage pretrained MLLM to generate rich captions as supervision. This allows our model to learn from rich and diverse vocabularies.

## 4.2 MODEL ARCHITECTURE

As illustrated in Fig.3, our SEAL leverages a pretrained EventSAM backbone and a SAM mask decoder as a class-agnostic mask generator to predict instance masks. We build our light-weight Multimodal Fusion Network on top of EventSAM backbone as mask classifier to achieve a parameter-

efficient design with single backbone. The three main components of our Multimodal Fusion Network are detailed in the following sections.

**1) Backbone Feature Enhancer.** This module aims to enhance backbone features by explicitly integrating language information into event-domain features. Specifically, backbone features $\mathbf{I}^{evt} \in \mathbb{R}^{D_2 \times H_2 \times W_2}$ are first extracted from the EventSAM backbone by taking event embedding $I^{evt}$ as input. They are then fed to $L = 6$ layers of the multimodal fusion module, where each layer comprises a self-attention module, a cross-attention module, and a feed-forward network. During training, text domain guidance $\mathbf{M}_l^{text} \in \mathbb{R}^{K_l \times D}$ obtained from Sec. 4.1 are fed into the cross-attention module as keys and values to enable multimodal fusion. During inference, either the class candidates provided by the dataset or user-defined language input is given to the fusion module instead. Finally, the language-fused events features $\hat{\mathbf{I}}^{evt} \in \mathbb{R}^{D_2 \times H_2 \times W_2}$ are obtained here. Tab. 5 shows that this explicit fusion of events and language results in improved performance due to the enhancement of the alignment between event representations and the multimodal feature space of CLIP.

Guided by the binary mask predictions from the mask decoder, we extract features of corresponding regions from the language-fused backbone features $\hat{\mathbf{I}}^{evt} \in \mathbb{R}^{D_2 \times H_2 \times W_2}$ via RoI-Align (He et al., 2017) pooling operation, obtaining the mask features for classification. During training, the image masks $M_l^{img}$ from Sec. 4.1 are used to guide pooling over $\hat{\mathbf{I}}^{evt}$ to extract the mask features $\mathbf{S}_l^{evt}$, which is then aligned with multimodal semantic guidance. Despite its effectiveness, the sole reliance on the pooling method to obtain mask features yields two main issues. *1) Dead masks:* The masks for small objects, *i.e.* part-level proposals often disappear when they are downsampled to match the low-resolution of the feature map $\hat{\mathbf{I}}^{evt}$. This is due to the significant downscaling from the original mask resolution. Consequently, the pooling layer outputs zero vectors which cause unreliable predictions. The naive solution is to upscale $\hat{\mathbf{I}}^{evt}$ into the original mask resolution before pooling. However, applying RoI-Align on high-resolution feature maps substantially increases inference time, particularly when processing a large number of masks (*cf.* Fig. 7c). *2) Semantic Conflict:* At low feature-map resolution ($H_2 \times W_2$), the mask pooling operation projects multiple masks with different semantics onto the same region of $\hat{\mathbf{I}}^{evt}$. This overlap results in an indistinct event feature space (*cf.* Fig. 4a). To address this issue, we complement the pooled mask features $\mathbf{S}^{evt}$ with two additional components: *Spatial Encoding* and *Mask Feature Enhancer*. Refer to Sec. A.5 of the Supplementary material for more details about these two issues and the motivation of *Spatial Encoding* module.

**2) Spatial Encoding.** As shown in Fig. 4b, this module aims to make event representations more discriminative by encoding the spatial priors into mask features. To this end, we leverage the mask token from the SAM mask decoder that encodes rich spatial cues such as the shape and position of the mask. During training, we calculate the bounding boxes $b_l \in \mathbb{R}^{K_l \times 4}$ from the image masks $M_l^{img}$ and feed them into the SAM mask decoder as visual prompts along with the event backbone features $\mathbf{I}^{evt}$. The resulting mask tokens $\mathbf{G}_l^{evt} \in \mathbb{R}^{K_l \times D}$ are then combined with the mask features $\mathbf{S}_l^{evt} \in \mathbb{R}^{K_l \times D}$, which can be formulated as $\mathbf{M}_l^{evt} = \text{proj}(\text{concat}(\mathbf{G}_1^{evt}, \mathbf{S}_1^{evt}))$, where $\text{concat}(\cdot, \cdot)$ and $\text{proj}(\cdot)$ denote the concatenation operation and the projection layer, respectively. During inference, user-provided visual prompts are fed into the SAM mask decoder to obtain the mask token $\mathbf{G}^{evt}$ and the corresponding event mask prediction $M^{evt}$. The $M^{evt}$ is then used to pool $\mathbf{S}^{evt}$ from the enhanced backbone features $\hat{\mathbf{I}}^{evt}$, followed by integration with $\mathbf{G}^{evt}$ for spatial encoding. Note that $\mathbf{S}^{evt}$ corresponds to *semantic* features obtained from the language-fused feature representations $\hat{\mathbf{I}}^{evt}$. In comparison, $\mathbf{G}^{evt}$ corresponds to *spatial* features guided by the mask decoder. Combining these two features enables mutual compensation between *semantic* and *spatial* information.

**3) Mask Feature Enhancer.** Semantic and spatial priors encoded in mask features $\mathbf{M}^{evt}$ are further enhanced through a cross-attention layer, where the language-fused backbone features $\hat{\mathbf{I}}^{evt}$ (*semantic*) with positional encodings (*spatial*) serve as keys and values. To focus attention on relevant areas, the mask features $\mathbf{M}^{evt}$ are forced to aggregate the foreground regions of the predicted mask by applying masked attention (Cheng et al., 2022). During training, we use image masks $M_l^{img}$ as attention masks. During inference, the attention masks are instead derived from the predicted event masks $M^{evt}$ produced by the mask decoder.

## 4.3 TRAINING

**Dataset.** We collect event-image pairs by combining the training splits of two popular event-segmentation datasets (Alonso & Murillo, 2019; Sun et al., 2022), named as ***Mixed-24K***. It contains

Table 1: **Quantitative results in closed-set event instance segmentation.** Our SEAL shows superior effectiveness and efficiency in performance and inference time by outperforming all baselines across three benchmarks with parameter-efficient architecture. The colors of 1st column indicates the three categories described in Sec. 3. Best results are in **bold**, and second best are underscored.

| Method | DDD17-Ins | | | | | | DSEC11-Ins | | | | | | DSEC19-Ins | | | | | | Efficiency | |
|---|---|---|---|---|---|---|---|---|---|---|---|---|---|---|---|---|---|---|---|---|
| | Point | | | Box | | | Point | | | Box | | | Point | | | Box | | | | |
| | AP | AP$_{50}$ | AP$_{25}$ | AP | AP$_{50}$ | AP$_{25}$ | AP | AP$_{50}$ | AP$_{25}$ | AP | AP$_{50}$ | AP$_{25}$ | AP | AP$_{50}$ | AP$_{25}$ | AP | AP$_{50}$ | AP$_{25}$ | Time (ms) | #Params (M) |
| OVSAM (Yuan et al., 2024a) | 20.2 | 22.8 | 25.2 | 21.6 | 24.3 | 24.6 | 16.7 | 21.0 | 24.8 | 22.2 | 28.2 | 29.3 | 8.8 | 11.5 | 13.4 | 11.6 | 14.8 | 15.4 | 102.27 | 314.7 |
| CLIP (Radford et al., 2021) | 20.6 | 21.3 | 21.9 | 22.3 | 23.5 | 23.7 | 15.4 | 16.6 | 17.5 | 17.8 | 19.6 | 20.3 | 8.1 | 8.9 | 9.6 | 9.4 | 10.7 | 11.1 | 329.43 | 529.9 |
| OVSeg (Liang et al., 2023) | 21.1 | 21.5 | 21.7 | 21.5 | 22.1 | 22.2 | 16.2 | 17.7 | 19.0 | 18.1 | 20.3 | 21.0 | 8.6 | 9.4 | 10.0 | 9.9 | 11.0 | 11.3 | 292.86 | 530.4 |
| MaskCLIP (Zhou et al., 2022) | 19.5 | 19.9 | 20.9 | 20.0 | 21.3 | 22.4 | 13.9 | 15.6 | 16.9 | 14.1 | 17.3 | 18.2 | 6.8 | 7.2 | 8.3 | 7.0 | 9.1 | 10.2 | 296.01 | 529.9 |
| OpenSeg (Ghiasi et al., 2022) | 29.8 | 40.3 | 45.2 | 35.0 | 49.9 | 53.3 | 20.4 | 25.9 | 28.7 | 23.6 | 31.9 | 33.7 | 10.9 | 14.2 | 16.6 | 13.0 | 18.7 | 19.9 | 427.01 | 228.4 |
| MaskCLIP++ (Zeng et al., 2024) | 27.4 | 35.2 | 40.1 | 32.8 | 44.7 | 47.7 | 20.7 | 28.9 | 34.6 | 25.4 | 37.9 | 41.7 | 11.1 | 14.9 | 17.5 | 14.1 | 20.2 | 22.1 | 394.61 | 301.7 |
| Mask-Adapter (Li et al., 2024) | 27.3 | 37.0 | 41.7 | 30.0 | 41.6 | 44.3 | 18.1 | 23.1 | 27.0 | 21.3 | 28.9 | 31.1 | 10.1 | 13.5 | 15.8 | 12.0 | 16.7 | 18.0 | 287.49 | 316.7 |
| frame2recon (He et al., 2016) | 28.8 | 38.0 | 42.2 | 34.8 | 48.7 | 51.8 | 18.1 | 22.3 | 25.2 | 21.2 | 28.5 | 30.2 | 9.7 | 12.1 | 13.3 | 10.5 | 13.6 | 14.3 | 278.35 | 141.7 |
| frame2voxel (Rebecq et al., 2019) | 27.6 | 35.5 | 39.2 | 33.6 | 46.0 | 48.6 | 18.0 | 22.3 | 25.3 | 21.3 | 28.7 | 30.5 | 9.6 | 11.9 | 13.2 | 11.3 | 15.3 | 16.1 | 88.19 | 109.1 |
| frame2spike (Kim et al., 2022b) | 26.4 | 32.7 | 35.3 | 30.7 | 40.3 | 42.4 | 17.9 | 21.9 | 24.7 | 20.7 | 27.4 | 29.0 | 9.3 | 11.2 | 12.3 | 11.4 | 15.3 | 16.1 | 575.41 | 95.9 |
| SEAL (Ours) | **32.3** | **44.4** | **50.9** | **38.2** | **55.1** | **59.3** | **22.4** | **31.1** | **36.3** | **28.8** | **43.6** | **47.5** | **11.7** | **16.0** | **18.6** | **14.8** | **22.5** | **24.2** | 22.28 | 99.1 |
| | (+3.5) | (+4.1) | (+5.7) | (+3.2) | (+5.2) | (+6.0) | (+1.7) | (+2.2) | (+1.7) | (+3.4) | (+5.7) | (+5.8) | (+0.6) | (+1.1) | (+1.1) | (+0.7) | (+2.3) | (+2.1) | | |

Table 2: **Quantitative results on closed-set event part segmentation.**

| Method | DSEC-Part | | | | | |
|---|---|---|---|---|---|---|
| | Point | | | Box | | |
| | AP | AP$_{50}$ | AP$_{25}$ | AP | AP$_{50}$ | AP$_{25}$ |
| VLPart (Sun et al., 2023) | 12.9 | 15.9 | 16.8 | 16.1 | 23.2 | 24.0 |
| frame2recon (He et al., 2016) | 9.8 | 10.1 | 10.3 | 11.1 | 11.2 | 11.4 |
| frame2voxel (Rebecq et al., 2019) | 10.2 | 10.4 | 10.7 | 11.5 | 11.7 | 12.3 |
| frame2spike (Kim et al., 2022b) | 9.4 | 9.3 | 9.8 | 10.3 | 10.4 | 10.9 |
| SEAL (Ours) | **13.6** | **16.6** | **17.4** | **18.3** | **26.9** | **31.0** |
| | (+0.7) | (+0.7) | (+0.6) | (+2.2) | (+3.7) | (+7.0) |

Table 3: **Ablation study on Hierarchical Semantic Guidance.** AP scores are used as metric.

| part | instance | semantic | DDD17-Ins | | DSEC11-Ins | | DSEC-Part | |
|---|---|---|---|---|---|---|---|---|
| | | | Point | Box | Point | Box | Point | Box |
| ✓ | | | 31.2 | 37.5 | 21.0 | 27.3 | 13.5 | 18.2 |
| | ✓ | | 32.2 | **38.2** | 22.5 | 28.8 | 12.7 | 16.3 |
| | | ✓ | 31.0 | 36.9 | 22.5 | 28.2 | 11.9 | 14.4 |
| ✓ | ✓ | | 32.0 | 38.1 | 21.5 | 27.6 | 13.2 | **18.5** |
| | ✓ | ✓ | 31.5 | 37.9 | 22.4 | 28.3 | 12.4 | 15.4 |
| ✓ | ✓ | ✓ | **32.3** | **38.2** | 22.4 | 28.8 | 13.6 | 18.3 |

24,032 event-image pairs, and all of the pairs are used during the training with pre-processed MHSG. Refer to Sec. A.4 of supplementary material for more details on the training dataset and MHSG.

**Training.** Our training process consists of two stages. In stage 1, we follow the original paper (Chen et al., 2024) to train EventSAM on the *Mixed-24K* dataset. In stage 2, we train the *backbone feature enhancer* together with the *spatial encoding* and *mask feature enhancer* module. Here, we keep the pretrained EventSAM including its backbone, mask decoder, and prompt encoder frozen. For training supervision, the image masks $M_l^{img}$ from Sec. 4.1 are used as a guide where the corresponding mask features $\hat{\mathbf{M}}_l^{evt}$ are pooled from SEAL and subsequently aligned with MHSG. Specifically, the mask features are aligned with both image-domain $\mathbf{M}_l^{img}$ and text-domain $\mathbf{M}_l^{text}$ guidance for each semantic level $l \in \{s, i, p\}$ using a cosine similarity loss. We thus formulate the distillation loss function as:

$$\mathcal{L}_{distill} = \sum_l^{\{s,i,p\}} \frac{1}{K_l}(1 - \cos(\hat{\mathbf{M}}_l^{evt}, \mathbf{M}_l^{img})) + \sum_l^{\{s,i,p\}} \frac{1}{K_l}(1 - \cos(\hat{\mathbf{M}}_l^{evt}, \mathbf{M}_l^{text})), \quad (1)$$

where $\cos(\cdot, \cdot)$ is the cosine similarity function with L2 feature normalization.

## 5 EXPERIMENT

### 5.1 EXPERIMENTAL SETTINGS

**Benchmarks.** Since no existing benchmark is available for event instance segmentation, we adapt two widely used ESS datasets (Alonso & Murillo, 2019; Sun et al., 2022) to evaluate our model. Specifically, given the event-image pairs with ground-truth semantic maps, we generate ground-truth instance masks from images using SAM while the semantic label for each instance is obtained from the corresponding semantic map given by the datasets or human annotators. Finally, we construct four benchmarks to evaluate OV-EIS on diverse settings. 1) The *DDD17-Ins* is a benchmark with 6 semantic classes derived from *DDD17-Seg* (Alonso & Murillo, 2019) dataset. It consists of 3,890 testing events acquired by DAVIS346B with spatial size of $352 \times 200$, along with synchronized gray-scale frames provided by a DAVIS camera. 2) The *DSEC11-Ins* benchmark is derived from *DSEC-Semantic* (Sun et al., 2022) dataset, which contains three testing sequences comprising 2,809 events at a spatial resolution of $640 \times 440$. The class label for each instance obtained from the semantic map parsed into 11 semantic classes. 3) The *DSEC19-Ins* is a fine-grained annotated version of *DSEC11-Ins* with 19 semantic classes while sharing the same scene. 4) The *DSEC-Part* benchmark is designed to assess event part segmentation using the testing sequences of *DSEC-Semantic* (Sun et al., 2022). We choose *Vehicle* and *Building* as base categories to obtain part-level masks since they have multiple distinguishable components. We define five part-level classes for *vehicle* while four classes for *building*. The class label for each mask is manually annotated to ensure an accurate evaluation since the original dataset does not provide ground-truth part-level semantic maps. It should be noted that our benchmarks cover diverse evaluation settings including *label granularity* and *semantic granularity* (*cf.* Fig. 5b). Specifically, evaluation on *DSEC11-Ins* and *DSEC19-Ins*

simulates the *label granularity* from 11 classes to 19 classes while evaluation on *DSEC11-Ins* and *DSEC-part* captures *semantic granularity* from instance-level to part-level segmentation. Please refer to the Sec. A.1 of the supplementary material for more details about the benchmark configurations.

**Metrics.** We employ a commonly used instance segmentation metric: Average Precision (AP). AP scores are evaluated at mask overlap thresholds of 50% and 25%, and averaged over the range of $[0.5 : 0.95 : 0.05]$. The prediction confidence score is set to 1.0 in our experiments.

**Implementation Details.** We resize input images and event frames to $512{\times}512$ and use ViT-B (Dosovitskiy et al., 2020) based EventSAM (Chen et al., 2024) as the mask generator. Our SEAL is trained for 15,000 iterations with Adam (Kingma & Ba, 2014) using a batch size of 8, and an initial learning rate of 2e-4 decayed by 0.9 at the $3^{rd}$ epoch. All experiments are conducted on a single NVIDIA RTX 6000 Ada.

**Baselines.** Four types of baselines are illustrated in Fig. 2. We choose OVSAM (Yuan et al., 2024a) as the *Semantic-aware SAM Baseline* in Fig. 2a since the code is publicly available. In Fig. 2b, the original CLIP (Radford et al., 2021) and OVSeg (Liang et al., 2023) are adopted as mask classifier for the *Image Crop Baseline*, where OVSeg is a fine-tuned variant of CLIP. For *Feature Pooling Baseline* shown in Fig. 2c, we choose MaskCLIP (Zhou et al., 2022) as the structure-modified variant of CLIP and, OpenSeg (Ghiasi et al., 2022), MaskCLIP++ (Zeng et al., 2024) and MaskAdapter (Li et al., 2024) as the fine-tuned variants of CLIP to serve as mask classifier. For the *Event Backbone Baseline* in Fig. 2d, we form `frame2voxel`, `frame2recon` and `frame2spike` based on the use of event representations, where E2VID (Rebecq et al., 2019), ResNet-50 (He et al., 2016) and Spiking-DeepLab (Kim et al., 2022b) are adopted as event backbone, respectively. The event backbone is trained using only event–image pairs from *Mixed-24K* to be consistent with *AF-DA* setting.

**Evaluation with Prompts.** Following (Yuan et al., 2024a; Li et al., 2023), we evaluate our SEAL in the interactive segmentation setting using visual prompts such as boxes or points sampled from ground-truth masks. For point prompts, three points are sampled using the furthest point sampling (Qi et al., 2017). In the supplementary material, we further show the effective and efficient variant of our SEAL, named SEAL++, that supports *prompt-free* OV-EIS by combining the spatiotemporal object detection models for events (Gehrig & Scaramuzza, 2023; Zhang et al., 2025) into our framework.

## 5.2 EXPERIMENTAL RESULTS

**Closed-Set Event Instance Segmentation.** We quantitatively evaluate our approach on the closed-vocabulary event instance segmentation task on *DDD17-Ins*, *DSEC11-Ins* and *DSEC19-Ins*. As shown in Tab. 1, our SEAL achieves clear improvements over all baselines on *DDD17-Ins* and *DSEC11-Ins*. Specifically, we surpass the second-best results with box prompts by 3.2% and 3.4% AP, respectively. OVSAM in the *AR-CDG* category shows unsatisfactory performance due to the huge domain gap between images and frame-like reconstructed event frames . In the *Hybrid* category, OpenSeg (Ghiasi et al., 2022) and MaskCLIP++ (Zeng et al., 2024) outperform OVSAM by leveraging higher-quality event masks generated by an event-adapted mask generator (Chen et al., 2024). However, a domain gap remains between images and reconstructed events during mask classification leading to sub-optimal performance. We also observe that the *Feature Pooling Baselines* (Ghiasi et al., 2022; Zhou et al., 2022; Zeng et al., 2024; Li et al., 2024) achieves better performance than the *Image Crop Baselines* (Radford et al., 2021; Liang et al., 2023) since mask classifiers for Feature Pooling Baselines are explicitly designed for dense prediction tasks during pretraining. The baselines in the *AF-DA* category addressed the domain gap between events and images through domain adaptation. However, their performance remains unsatisfactory due to the absence of text-domain guidance and explicit spatial priors. Furthermore, a *semantic conflict* occurs when CLIP features from high-resolution masks are distilled into lower-resolution event feature maps due to the severe downscaling. Hence, multiple instances with distinct semantics are often projected onto the same local region which ultimately degrading performance.

Our SEAL achieves the best performance by learning discriminative event representations through the rich MHSG supervision and an effective model architecture that leverages both semantic and spatial priors. The performance gap between our SEAL and the second-best results (Zeng et al., 2024) narrows on the *DSEC19-Ins* benchmark with more fine-grained classes. This shows the advantage of annotation-rich training with large-scale images leveraged by image-pretrained baselines. Despite the absence of direct human annotations, our SEAL still achieves the best performance surpassing competitors by 0.7%, 2.3%, and 2.1% on AP, AP50, and AP25 scores, respectively. We also compare

Table 4: **Ablation study on Multimodal Guidance.** AP scores are used as metric.

| VG | TG | DDD17-Ins | | DSEC11-Ins | | DSEC19-Ins | |
|---|---|---|---|---|---|---|---|
| | | Point | Box | Point | Box | Point | Box |
| ✓ | | 30.3 | 37.0 | 19.9 | 24.0 | 10.5 | 12.9 |
| | ✓ | 31.6 | 35.2 | 21.3 | 28.0 | 10.7 | 13.0 |
| ✓ | ✓ | **32.3** | **38.2** | **22.4** | **28.8** | **11.7** | **14.8** |

Table 5: **Ablation study on Model Architecture.** AP scores are used as metric.

| Fusion | SE | MFE | DDD17-Ins | | DSEC11-Ins | | DSEC19-Ins | | DSEC-Part | |
|---|---|---|---|---|---|---|---|---|---|---|
| | | | Point | Box | Point | Box | Point | Box | Point | Box |
| ✓ | | | 29.6 | 35.5 | 20.1 | 24.3 | 10.7 | 12.9 | 12.2 | 14.9 |
| ✓ | ✓ | | 30.2 | 35.7 | 21.4 | 26.2 | 10.8 | 13.5 | 12.6 | 15.7 |
| ✓ | | ✓ | 32.1 | 38.1 | 22.0 | 27.7 | 11.4 | 14.2 | 12.3 | 16.6 |
| | ✓ | ✓ | 32.0 | 38.1 | 21.8 | 27.1 | 11.3 | 14.5 | 12.4 | 15.2 |
| ✓ | ✓ | ✓ | **32.3** | **38.2** | **22.4** | **28.8** | **11.7** | **14.8** | **13.6** | **18.3** |

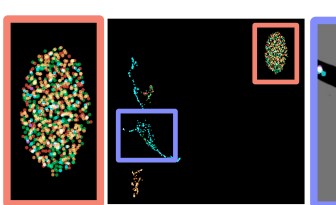
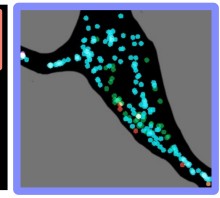
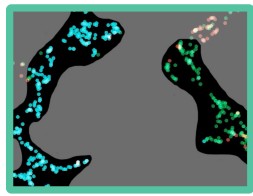
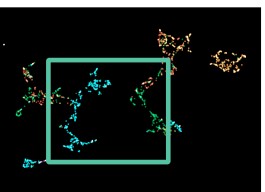

Dead masks     Semantic Conflict     Distinct Feature Space

(a) *wo* Spatial Encoding & Mask Feature Enhancer     (b) *w* Spatial Encoding & Mask Feature Enhancer

Figure 4: **Event feature space visualized by UMAP (McInnes et al., 2018).** Event representation learned without SE and MFE exhibits *dead masks* (*Red* box) and *semantic conflict* (*Purple* box) with indistinct feature space. Our model addresses these issues by encoding spatial prior via SE and MFE to give a more discriminative feature space (*Green* box). Refer to Sec. 5.3 for further details.

the efficiency on both inference time and parameter size in the *gray*-colored column of Tab. 1. Our SEAL shows significantly fewer parameters and the shortest inference time, which highlights the efficiency of its single-backbone architecture. The small model size of our SEAL aligns with practical requirements since event cameras are often deployed on edge devices with low-power constraints (Ramesh et al., 2020). The practicality of our SEAL is further demonstrated by its fast model inference that enables efficient exploitation of the asynchronous high temporal resolution of events. More qualitative results are provided in the supplementary material Sec. B.5.

**Closed-Set Event Part Segmentation.** We also evaluate our approach on the closed-vocabulary event part segmentation on *DSEC-Part* benchmark. For comparison, we adopt VLPart (Sun et al., 2023), an image-pretrained open vocabulary part segmentation model as a mask classifier within the *feature pooling baseline*. *Event backbone baseline* is further adopted for comparison. Our SEAL achieves higher AP scores than the baselines, demonstrating the effectiveness of MHSG.

## 5.3 ABLATION STUDY

**Hierarchical Semantic Guidance.** In Tab. 3, we analyze the hierarchical semantic guidance of MHSG across *part*, *instance* and *semantic* levels of granularity in *DDD17-Ins*, *DSEC11-Ins* and *DSEC-Part* benchmarks. We have following observations. **1)** We observe performance degradation on the part segmentation benchmark (*DSEC-Part*) when *part*-level guidance is excluded (2nd, 3rd, and 5th rows). **2)** Performance on instance segmentation benchmarks (*DDD17-Ins* and *DSEC11-Ins*) falls into sub-optimal when guidance from higher level of granularity such as *instance*-level or *semantic*-level is removed (1st row). **3)** Our SEAL consistently contributes the best performance across diverse benchmarks including both part and instance segmentation only when the guidance from all levels of semantic granularity is combined (4th, 5th rows vs **6th row**). This result shows the effectiveness of our hierarchical semantic guidance.

**Multimodal Semantic Guidance.** In Tab. 4, we analyze the multimodal guidance of MHSG on the *DDD17-Ins*, *DSEC11-Ins* and *DSEC19-Ins* benchmarks. Using both of Visual-domain Guidance (VG) and Text-domain Guidance (TG) yields the best performance in all the benchmarks (3rd row).

**Model Architecture.** In Tab. 5, we analyze the model components. Our SEAL shows unsatisfactory performance when Spatial Encoding (SE) and Mask Feature Enhancer (MFE) are removed (1st row). To investigate the effects of SE and MFE, we visualize the learned event feature space using UMAP (McInnes et al., 2018). Specifically, we collect the predicted mask features from the *DSEC-Part* benchmark and visualize them in 2D space, where each feature is color-coded according to its ground-truth label. Fig. 4a shows the visualization results of SEAL trained without SE and MFE. As we mentioned in Sec. 4.2, masks with small size often disappear due to the severe downscaling to feature map size. This results in zero vectors from the RoI-Align layer which we referred to as dead masks. These dead masks are shown in *Red* box, where multiple masks with different semantics

(colors) are clustered together since they are all mapped to zero values. Additionally, the *Purple* box illustrates the indistinct feature space caused by semantic conflict, where the green and blue dots lack clear separations. We address these issues by making the event feature space more discriminative with the encoding of spatial prior via the SE and MFE modules. As shown in Fig. 4b, our SEAL with ME and SFE shows more distinct feature space, where green and blue dots are clearly separated (*Green* box). Dead masks are also not observed. Additionally, Tab. 5 shows that combining SE (2nd row) or MFE (3rd row) yields a performance improvement compared to the model without either module (1st row). Using both ME and SFE achieve the best performance across all benchmarks including part and instance segmentation (1st-3rd rows vs **5th row**). Similarly, the fusion of text knowledge in the backbone feature enhancer also leads to better performance (4th row vs **5th row**) with better alignment between events and text.

## 6 CONCLUSION

In this study, we address the Open-Vocabulary Event Instance Segmentation (OV-EIS) problem, an important yet largely overlooked research challenge. We first propose four types of simple baselines that enables OV-EIS by combining the existing open vocabulary components. Then, we discuss the limitations of our proposed baselines which further motivate us to design SEAL, the novel framework to address OV-EIS that understands the event streams with free-form of language across multiple levels of granularity. Extensive experiments with our proposed EIS benchmarks show that our SEAL outperforms all of the baselines in terms of performance and inference speed with parameter-efficient architecture. We believe that SEAL can serve as a strong starting point for open-world, fine-grained event understanding,

**Acknowledgment.** This research / project is supported by the National Research Foundation, Singapore, under its NRF Investigatorship Programme (Award ID. NRF-NRFI09- 0008), and the Tier 1 grant T1-251RES2305 from the Singapore Ministry of Education.

## ETHICS STATEMENT

This research complies with the ICLR Code of Ethics. In this section, we discuss the following ethical considerations.

**Data and Privacy :** The datasets used in this research are publicly available (Sun et al., 2022; Alonso & Murillo, 2019).

**Potential Harmful Applications :** The proposed method for open-vocabulary events instance segmentation could be used in sensitive domains such as surveillance and public-space crowd monitoring. We urge future users to deploy the model responsibly, with appropriate safeguards to mitigate potential misuse.

**Research Integrity :** We confirm that all experiments were conducted in accordance with best practices in research ethics, and all results are reported with full transparency and accuracy.

## REPRODUCIBILITY STATEMENT

To ensure the reproducibility of our work, we state specific implementation details in Sec. 5.1 and Supplementary material Sec. A that cover the specifications for our proposed benchmarks, baselines, MHSG module and the model architecture of SEAL. The code will be made publicly available.

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

# Segment Any Events with Language

## Supplementary Material

TABLE OF CONTENTS

## A  IMPLEMENTATION DETAILS

In the following sections, we present additional details to facilitate the implementation and reproducibility of our proposed SEAL framework. In Sec. A.1, we outline the specific configurations and data processing pipeline of our proposed four EIS benchmarks. In Sec. A.2, we supplement the preliminaries of our work by detailing the two pioneering works in event-based scene understanding with image foundation model. Based on the preliminaries, we present the implementation details of our proposed baselines under three categories in Sec. A.3. Finally, we provide the design specifications of MHSG and our proposed SEAL framework in Sec. A.4 and Sec. A.5, respectively.

### A.1  BENCHMARKS

In this study, we use two popular event-segmentation datasets: ***DSEC-Semantic*** (Sun et al., 2022) and ***DDD17-Seg*** (Alonso & Murillo, 2019) to construct four EIS benchmarks that simulate diverse evaluation settings for open-vocabulary scene understanding. Fig. 5a illustrates the whole processing pipeline of the proposed benchmarks in two stages. 1) We first generate class-agnostic masks by leveraging the auto-segmentation capabilities of SAM (Kirillov et al., 2023). SAM can generate three different masks with different granularity for a single point where we denote *semantic-level*, *instance-level* and *part-level*. All levels of granularity are used to construct the EIS benchmarks where *DSEC11-Ins*, *DSEC19-Ins* and *DDD17-Ins* leverage *semantic-level* and *instance-level* masks while *DSEC-Part* exploits *part-level* masks. To better separate each granularity in the masks generation, we use the customized SAM proposed by Qin et al. (2024). 2) Semantic labels are subsequently assigned to SAM-generated masks by parsing the ground-truth semantic maps given by the original datasets (Alonso & Murillo, 2019; Sun et al., 2022) or employing human annotators.

In the following, we supplement the additional details of two existing ESS datasets (Sun et al., 2022; Alonso & Murillo, 2019) and the specific configurations of our proposed EIS benchmarks (*cf.* Tab. 6). Furthermore, Fig. 6 illustrates the class distributions of each EIS benchmark.

**DDD17-Seg** (Alonso & Murillo, 2019) is the first ESS benchmark built on top of the DDD17 (Binas et al., 2017) dataset. DDD17 (Binas et al., 2017) provides the large-scale event-image pairs

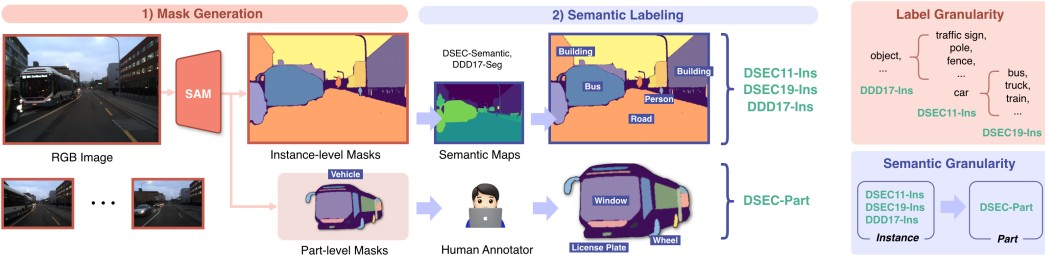

Figure 5: **(a)** We first generate class-agnostic masks using SAM, and then assign semantic labels to each mask proposal either from ground-truth semantic maps or through human annotation. **(b)** Our four EIS benchmarks simulate *label granularity* from coarse to fine-grained class configurations and *semantic granularity* from instance-level to part-level understanding.

acquired by a DAVIS346B over several hours of driving scenes. Semantic maps for the corresponding event-image pairs are additionally generated by DDD17-Seg by using a pretrained semantic segmentation model. In total, DDD17-Seg comprises 15,950 training event-image pairs from 8 driving sequences and 3,890 testing samples from a single driving sequence. Each sample has 352 × 200 spatial resolution, where each pixel is mapped to one of the six semantic classes, including `flat`, `construction+sky`, `object`, `vegetation`, `human` and `vehicle`. We further derive *DDD17-Ins* by adding the instance-level masks to DDD17-Seg. For evaluation, the background class is excluded to align with the instance segmentation setting where the "`flat`" class is removed.

**DSEC-Semantic (Sun et al., 2022)** extends the DSEC (Gehrig et al., 2021) dataset by providing additional semantic maps to its extensive collection of event–image pairs. The DSEC dataset itself is a stereo driving benchmark that includes recordings from two monochrome event cameras and two global-shutter color cameras. Specifically, Prophesee Gen3.1M sensors are used for events, and FLIR BlackFly S USB3 cameras are adopted for RGB. Based on this rich set of event–image pairs, DSEC-Semantic provides pixel-wise class annotations across 8 training sequences and 3 testing sequences. In total, the dataset comprises 8,082 training samples and 2,809 testing samples, where each sample has 640 × 440 spatial resolution. Each pixel is annotated by two types of semantic labels with a different number of classes, one with 11 classes and another with 19 classes:

- 11 classes: `background`, `building`, `fence`, `person`, `pole`, `road`, `sidewalk`, `vegetation`, `car`, `wall`, `traffic sign`.
- 19 classes: `road`, `sidewalk`, `building`, `wall`, `fence`, `pole`, `traffic light`, `traffic sign`, `vegetation`, `terrain`, `sky`, `person`, `rider`, `car`, `truck`, `bus`, `train`, `motorcycle`, `bicycle`.

Based on DSEC-Semantic, we propose three EIS benchmarks by adding either instance-level masks or part-level masks. Specifically, *DSEC11-Ins* is derived by assigning 11 classes to instance masks, while *DSEC19-Ins* adopts 19 classes to annotate the object proposals. Similarly to DDD17-Ins, background classes are excluded in the evaluation to match the instance segmentation setting. Specifically, "`background`", "`road`", "`sidewalk`" and "`wall`" classes are excluded for the DSEC11-Ins benchmark, while DSEC19-Ins additionally removes "`sky`" class as part of the background category.

One of the main objectives of this study is to enable events semantic recognition across multiple levels of granularity, encompassing both instance-level and part-level objects. To this end, we introduce a new benchmark for event part segmentation, named *DSEC11-Part*. Given the part-level mask proposals from SAM, we employ human annotators to manually label the part objects in the testing sequences of DSEC-Semantic. We first select several instance-level classes that have multiple distinguishable components and carefully design their part-level configurations according to their semantic structure. Specifically, "`building`" and "`vehicle`" are selected as base classes where five part-level classes are defined for "`vehicle`" and four classes are assigned to "`building`":

- vehicle: `wheel`, `window`, `light`, `side mirror`, `license plate`
- building: `window`, `roof`, `door`, `terrace`

Table 6: **Data configurations of the four EIS benchmarks.**

| Benchmarks | DDD17-Ins | DSEC11-Ins | | | DSEC19-Ins | | | DSEC-Part | | |
|---|---|---|---|---|---|---|---|---|---|---|
| Seq | dir1 | 13_a | 14_c | 15_a | 13_a | 14_c | 15_a | 13_a | 14_c | 15_a |
| # Frames & Events | 3,454 | 378 | 1,190 | 1,238 | 378 | 1,190 | 1,238 | 378 | 1,190 | 1,238 |
| # Mask nums | 48,303 | 6,462 | 24,572 | 27,866 | 6,399 | 23,543 | 26,681 | 3,240 | 13,144 | 15,193 |
| Resolution | $352 \times 200$ | $640 \times 440$ | | | | | | | | |
| # Classes | 5 classes | 7 Classes | | | 14 Classes | | | 9 Classes | | |

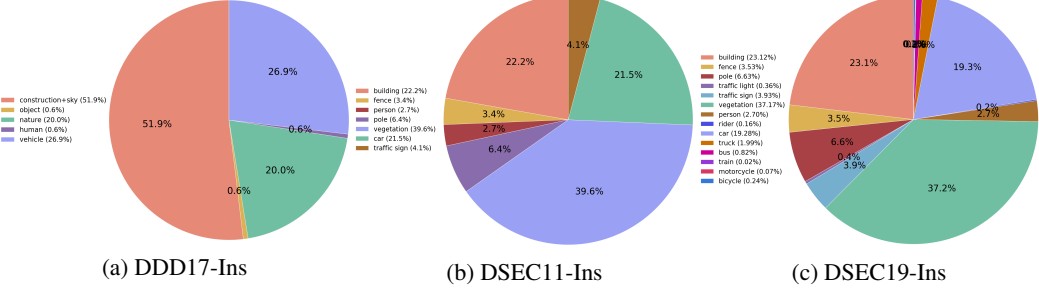

| (a) DDD17-Ins | (b) DSEC11-Ins | (c) DSEC19-Ins |

Figure 6: Class distributions for three EIS benchmarks.

**Discussion.** Our four EIS benchmarks are carefully designed to simulate the diverse evaluation settings, including *label granularity* and *semantic granularity* (*cf.* Fig. 5b). Evaluation on DDD17-Ins, DSEC11-Ins, and DSEC19-Ins models the *label granularity* where the number of classes are incremental from 5 to 14 within similar driving scenes. Specifically, "object" class in DDD17-Ins is subdivided into finer-grained categories in DDD11-Ins, such as "traffic sign", "pole", "fence", *etc*. Similarly, DSEC19-Ins further separates "car" class in DSEC11-Ins into more diverse semantics, including "bus", "truck", "train", *etc*. This enables evaluation of the model response to diverse class configurations by varying the levels of label granularity from coarse to fine. Furthermore, we consider *semantic granularity* by evaluating our SEAL on both instance-level and part-level benchmarks. This enables a thorough assessment of its semantic understanding with multiple granularities.

**Visualization.** In Fig. 9-10, we present additional visualizations of our proposed EIS benchmarks.

## A.2 PRELIMINARIES

Our SEAL is closely related to two pioneering works (Kong et al., 2024; Chen et al., 2024). These approaches transfer the foundational knowledge (Kirillov et al., 2023; Radford et al., 2021) of RGB-based models to event backbones, which enhances the generalizability and robustness of event-based scene understanding. In the following, we outline the additional details of these prior works as preliminaries.

**OpenESS (Kong et al., 2024)** proposes the first annotation-free framework to achieve open-vocabulary event semantic segmentation. Their framework only requires event-image pairs without any semantic labels, and therefore relaxes the inherent challenges of annotating asynchronous events. Given by the event-image pairs $(I^{evt}, I^{img})$, OpenESS aligns the event backbone $\mathcal{F}^{evt}$ with CLIP (Radford et al., 2021) through multimodal semantic guidance. It can be formulated as:

$$\mathcal{L}_{loss} = \mathcal{L}_{distill}(\mathcal{F}^{evt}(I^{evt}), \mathcal{F}^{clip}(I^{img})) + \alpha\mathcal{L}_{distill}(\mathcal{F}^{evt}(I^{evt}), \mathcal{T}^{clip}), \quad (2)$$

where $\mathcal{F}^{clip}$ denotes the CLIP image encoder that maps an RGB image $I^{img}$ into pixel-wise CLIP features. The event representation $I^{evt}$ is processed by an event backbone $\mathcal{F}^{evt}$ and subsequently aligned with the CLIP feature space via the distillation objective term $\mathcal{L}_{distill}$. In addition, $\mathcal{F}^{evt}$ is further guided by semantic supervision in the text domain $\mathcal{T}^{clip}$. Given by the closed-set of predefined class candidates, OpenESS generates pixel-wise class labels of event-image pairs by leveraging the multiple vision foundation models. Specifically, OpenESS first generates superpixels in the image domain using SAM and then assigns pseudo-labels to the resulting pixel groups by using CLIP as zero-shot classifier. The pixel-wise labels are subsequently mapped to the text representations $\mathcal{T}^{clip}$

Table 7: **Comparisons of four types of proposed baselines.** Mask classifiers in *AR-CDG* and *Hybrid* category are pretrained on large-scale image datasets (118K-13B) with rich human annotations and are directly applied to events. In contrast, baselines in *AF-DA* category are adapted to events domain by using relatively small set of event-image pairs (8K-15K) in an annotation-free manner.

| Category | Type | Baseline | Configuration (Segmentor \ Classifier) | Training Datasets | Data Size | Domain | Annotations |
|---|---|---|---|---|---|---|---|
| *AR-CDG* | Semantic-aware SAM Baseline | OVSAM (Yuan et al., 2024a) | OVSAM | SA-1B (1%) (Kirillov et al., 2023) | 33K | Image | ✓ |
| | | | OVSAM | COCO (Lin et al., 2014) | 118K | | |
| *Hybrid* | Image-Crop Baseline | CLIP (Radford et al., 2021) | EventSAM | DDD17-Seg (Alonso & Murillo, 2019), DSEC-Semantic (Sun et al., 2022) | 15K, 8K | Event-Image | ✗ |
| | | | CLIP | WIT (Radford et al., 2021) | 13B | Image | ✓ |
| | | OVSeg (Jia et al., 2021) | EventSAM | DDD17-Seg (Alonso & Murillo, 2019), DSEC-Semantic (Sun et al., 2022) | 15K, 8K | Event-Image | ✗ |
| | | | OVSeg | COCO (Lin et al., 2014) | 118K | Image | ✓ |
| | Feature-Crop Baseline | MaskCLIP (Zhou et al., 2022) | EventSAM | DDD17-Seg (Alonso & Murillo, 2019), DSEC-Semantic (Sun et al., 2022) | 15K, 8K | Event-Image | ✗ |
| | | | MaskCLIP | WIT (Radford et al., 2021) | 13B | Image | ✓ |
| | | OpenSeg (Ghiasi et al., 2022) | EventSAM | DDD17-Seg (Alonso & Murillo, 2019), DSEC-Semantic (Sun et al., 2022) | 15K, 8K | Event-Image | ✗ |
| | | | OpenSeg | COCO (Lin et al., 2014), Localized Narrative (Pont-Tuset et al., 2020) | 118K, 652K | Image | ✓ |
| | | MaskCLIP++ (Zeng et al., 2024) | EventSAM | DDD17-Seg (Alonso & Murillo, 2019), DSEC-Semantic (Sun et al., 2022) | 15K, 8K | Event-Image | ✗ |
| | | | MaskCLIP++ | COCO (Lin et al., 2014) | 118K | Image | ✓ |
| | | Mask-Adapter (Li et al., 2024) | EventSAM | DDD17-Seg (Alonso & Murillo, 2019), DSEC-Semantic (Sun et al., 2022) | 15K, 8K | Event-Image | ✗ |
| | | | Mask-Adapter | COCO (Lin et al., 2014) | 118K | Image | ✓ |
| *AF-DA* | Event-backbone Baseline | frame2recon (He et al., 2016) | EventSAM | DDD17-Seg (Alonso & Murillo, 2019), DSEC-Semantic (Sun et al., 2022) | 15K, 8K | Event-Image | ✗ |
| | | | ResNet-50 | | | | |
| | | frame2voxel (Rebecq et al., 2019) | EventSAM | | | | |
| | | | E2VID | | | | |
| | | frame2spike (Kim et al., 2022b) | EventSAM | | | | |
| | | | Spiking-DeepLab | | | | |

through the CLIP text encoder for additional supervision to the paired events with balancing weight $\alpha$. Kindly refer to Kong et al. (2024) for more details.

**EventSAM (Chen et al., 2024)** proposes the first Segmant Any Events framework to achieve instance-level event segmentation. Similarly to OpenESS, the event backbone $\mathcal{F}^{esam}$ is aligned with the feature space of the SAM image backbone using only event-image pairs $(I^{evt}, I^{img})$. It can be formulated as $\mathcal{L} = \mathcal{L}_{distillation}(\mathcal{F}^{evt}(I^{evt}), \mathcal{F}^{sam}(I^{img}))$, where $\mathcal{F}^{sam}$ denotes the image backbone of the original SAM (Kirillov et al., 2023). During inference, EventSAM can predict instance objects $M$ in the event domain by utilizing the SAM mask decoder $\mathcal{M}^{sam}$ as follows:

$$M = \mathcal{M}^{sam}(\mathcal{F}^{evt}(I^{evt}), \mathcal{P}^{sam}(P)), \qquad (3)$$

where $\mathcal{P}^{sam}$ is the SAM prompt encoder and $P$ represents visual prompts such as points or boxes. Kindly refer to Chen et al. (2024) for more details.

### A.3 BASELINES

Based on the preliminaries, we introduce four baselines that are grouped into three categories to address OV-EIS. We specify the implementation details for our proposed baselines in the following. Tab. 7 further presents a detailed comparison of the proposed baselines in terms of training setting.

*AR-CDG* category leverages the rich semantic knowledge derived from the large-scale image datasets to understand the event-streams as *cross-domain generalization*. Specifically, SAM-based open-vocabulary models (Yuan et al., 2024a; Li et al., 2023; Han et al., 2025; Wang et al., 2024; Chen et al., 2023) which are pre-trained on richly annotated image datasets are directly applied to event streams to achieve OV-EIS. To mitigate the inherent domain gap between images and events, we reconstruct events into gray-scale images using E2VID (Rebecq et al., 2019), which are then employed for inference. It can be formulated as:

$$M, C = \mathcal{M}(\mathcal{R}^{e2vid}(I^{evt}), P), \qquad (4)$$

where $\mathcal{R}^{\text{e2vid}}(\cdot)$ denotes E2VID operation and $\mathcal{M}$ represents the unified model for segmentation and mask classification. In the implementation, OVSAM (Yuan et al., 2024a) is adopted as AR-CDG baseline since it is fully open-sourced. Given the visual prompt $P$, mask predictions $M$ and semantics labels $C$ are directly obtained from OVSAM via unified framework.

*AF-DA* category focuses on adapting neural networks to event streams without relying on the dense event annotations where it is named as *Annotation-Free Domain Adaptation* in this study. It is comprised of two-stage framework: 1) Class-agnostic mask proposal module first generates the instance masks in the events domain. 2) Obtained instance proposals are subsequently classified by the open-vocabulary classifier. The whole pipeline can be formulated as:

$$M = \mathcal{M}^{\text{mask}}(I^{evt}, P), \tag{5}$$

$$\mathbf{f}^{cls} = \text{pool}(\mathcal{M}^{\text{cls}}(I^{evt}), M), \quad \mathcal{M}^{\text{cls}}(I^{evt}) \in \mathbb{R}^{H \times W \times D}, \tag{6}$$

where $\mathcal{M}^{\text{mask}}$ and $\mathcal{M}^{\text{cls}}$ denotes the mask proposal module and the mask classifier, respectively. Given the event representation $I^{evt}$, classifier $\mathcal{M}^{\text{cls}}$ produces per-pixel features which are subsequently pooled by the binary mask obtained from $\mathcal{M}^{\text{mask}}$. Specifically, RoI-Align (He et al., 2017) is used for the $\text{pool}(\cdot, \cdot)$ operation. The resulting pooled mask feature $\mathbf{f}^{cls}$ is then employed for the mask classification conditioned on user-provided text prompts.

For implementation, we adopt EventSAM (Chen et al., 2024) as a class-agnostic event segmentor which is pretrained by following the original paper. Mask classifier is implemented with three different types of event backbones inspired by OpenESS (Kong et al., 2024), where each corresponds to a distinct type of event representation:

- **frame2voxel** converts raw events $\epsilon$ into regular voxel grids that are compatible with convolutional neural networks since it encodes both spatial and temporal information. Given the predefined number of events or time window, the voxel grids can be formulated as:

$$I^{evt} = \sum_{\text{e}_{\text{j}} \in \epsilon_i} p_j \delta(\mathbf{x}_j - x) \delta(\mathbf{y}_j - y) \max(1 - |t_j^* - t|, 0), \quad t_j^* = (B-1)\frac{t_j - t_0}{\Delta T}, \tag{7}$$

  where $\Delta T$ is the time window and B is the number of temporal bins. $t_0$ denotes the timestamp of the first event in the current time window, and $\delta(\cdot)$ is the Kronecker delta function. In the implementation, we follow Sun et al. (2022) to produce voxelized event representations. We set $B = 3$ and use 25 ms of time window for the DSEC-based benchmarks, and a 15 ms window is employed for the DDD17-based benchmarks. E2VID (Rebecq et al., 2019) is adopted as backbone for this type where it outputs per-pixel event embeddings $I^{evt} \in \mathbb{R}^{H \times W \times D}$ with feature dimension $D$. In the implementation, we add additional ResNet blocks on top of the E2VID backbone. During training, we initialize the E2VID backbone with the official pretrained weights. We then keep it frozen while optimizing only the newly added ResNet blocks. This training scheme can be seen as linear probing which improves training stability and convergence speed.

- **frame2recon** adopts event reconstructions via E2VID as the pre-processing step for events. The image-based model ResNet-50 (He et al., 2016) is used for the backbone, where it ingests events-reconstructed frames as input. E2VID operation can be described as follows:

$$\mathbf{z}_k^{rec} = \mathcal{E}^{\text{e2vid}}(I_k^{evt}, \mathbf{z}_{k-1}^{rec}), \quad k = 1, \cdots, N, \tag{8}$$

$$I^{rec} = \mathcal{D}^{\text{e2vid}}(\mathbf{z}^{rec}), \tag{9}$$

  where $\mathcal{E}^{\text{e2vid}}$ and $\mathcal{D}^{\text{e2vid}}$ denotes the encoder and decoder of E2VID. The recurrent network is adopted to leverage information from previous events for the reconstruction of the gray-scale images with high fidelity. However, it should be noted that E2VID operation often introduces unwanted artifacts or loses details due to fast motions or low event rates.

- **frame2spike** represents the spiking neural network, where it first converts events into spikes and processes them a with spiking network. We follow Kim et al. (2022b) to implement spiking-based classifier where the Spiking-DeepLab (Kim et al., 2022b) is employed as backbone. We adopt rate coding as the spike encoding strategy (Kim et al., 2022b; Kong et al., 2024). Specifically, at each time step, a random value sampled from the range $[s_{min}, s_{max}]$ is assigned to every pixel where $s_{min}$ and $s_{max}$ correspond to the lower and upper bounds of pixel intensity. A spike with amplitude 1 is generated by the Poisson spike generator if the sampled value exceeds the pixel

Table 8: **Configurations of MHSG module.** We collect approximately 24K number of event-image pairs from two widely-used ESS datasets (Alonso & Murillo, 2019; Sun et al., 2022). All of the pairs are processed to generate multimodal semantic guidance.

| Source Dataset | | DSEC-Semantic (Sun et al., 2022) | | | | | | | | DDD17-Seg (Alonso & Murillo, 2019) | | | | |
|---|---|---|---|---|---|---|---|---|---|---|---|---|---|---|
| Seq | | 00_a | 01_a | 02_a | 04_a | 05_a | 06_a | 07_a | 08_a | dir0 | dir3 | dir4 | dir5 | dir7 |
| # Frames & Events | | 938 | 680 | 234 | 700 | 1752 | 1522 | 1462 | 786 | 5549 | 1320 | 6945 | 1 | 647 |
| # Mask nums | *semantic*-level | 18,641 | 12,260 | 3,506 | 14,412 | 30,647 | 23,156 | 27,401 | 13,062 | 66,333 | 13,977 | 82,432 | 1 | 6,578 |
| | *instance*-level | 66,773 | 31,218 | 14,291 | 41,322 | 85,119 | 66,529 | 86,091 | 30,375 | 205,352 | 37,056 | 226,818 | 10 | 22,351 |
| | *part*-level | 115,645 | 47,842 | 29,682 | 84,616 | 133,185 | 113,847 | 144,252 | 65,893 | 401,950 | 82,185 | 389,481 | 23 | 33,767 |
| Resolution | | $640 \times 440$ | | | | | | | | $352 \times 200$ | | | | |

intensity. Otherwise, no spike is emitted. Spikes accumulated within a predefined temporal window are then aggregated into a frame $I^{evt}$, which is subsequently used as input to the spiking semantic segmentation network.

All types of classifier are trained by following Eq. 2 (Kong et al., 2024), where the foundational knowledge of CLIP is distilled to an event neural network. However, semantic supervision in the text domain $\mathcal{T}^{clip}$ in Eq. 2 is excluded. In this work, *annotation-free* refers to training without access to any human-annotated semantic information. Since $\mathcal{T}^{clip}$ is constructed from the closed set of class candidates provided by the dataset, it contradicts the condition of *annotation-free* and is therefore removed. Note that the mask proposal module and mask classifier are trained separately using the event–image pairs collected from the two ESS datasets (Alonso & Murillo, 2019; Sun et al., 2022).

*Hybrid* combines the AF-DA mask generation approach with the AR-CDG mask classification strategy. It follows the two-stage framework of the AF-DA category (*cf.* Eq. 5-6) while the mask classifier $\mathcal{M}^{cls}$ is replaced with pretrained RGB-based models which fall into the AR-CDG category. To match the input modality of events with the RGB models, event representation is first reconstructed to a gray-scale frame via E2VID which is then passed into the mask classifier $\mathcal{M}^{cls}$.

This category is comprised of two types of baseline:

- **Image Crop Baseline** crops the E2VID-reconstructed frame using the predicted binary mask, and then classifies the resulting image patch with a mask classifier. It can be formulated as:

$$\mathbf{f}^{cls} = \mathcal{M}^{cls}(\text{crop}(\mathcal{R}^{\text{e2vid}}(I^{evt}), M)), \tag{10}$$

where $\text{crop}(\cdot, \cdot)$ denotes the cropping operation. We employ CLIP ViT-L/14@336px (Radford et al., 2021) and ViT-L/14 based OVSeg (Liang et al., 2023) as an open-vocabulary classifier. It should be noted that OVSeg represents the fine-tuned variant of CLIP, where additional learnable parameters are introduced to the original CLIP for the transfer of its image classification capabilities to the mask classification task.

- **Feature Pooling Baseline** adopts the pretrained models that are specifically designed to conduct per-pixel classification or mask classification. Given the predicted binary mask $M$ and event-reconstructed frame $\mathcal{R}^{\text{e2vid}}(I^{evt})$, the mask classifier $\mathcal{M}^{cls}$ pools the CLIP-aligned mask feature $\mathbf{f}^{cls}$ by using RoI-Align or additional pooling network. We employ MaskCLIP (Zhou et al., 2022), OpenSeg (Ghiasi et al., 2022), MaskCLIP++ (Zeng et al., 2024) and Mask-Adapter (Li et al., 2024) as open-vocabulary mask classifier for this baseline. Given the gray-scale frame reconstructed from events, MaskCLIP and OpenSeg generate per-pixel CLIP features which are then pooled by binary mask via the RoI-Align operation. Conversely, MaskCLIP++ and Mask-Adpater employ additional mask-pooling layers which are specially designed to achieve the mask classification task. These pooling layers consist of several learnable modules which are pretrained with large-scale annotated datasets (Lin et al., 2014). Note that MaskCLIP represents the structure-modified variant of CLIP where only the structure of CLIP is modified without any fine-tuning. In contrast, OpenSeg, MaskCLIP++ and Mask-Adpater denote fine-tuned variants of clip where rich semantic knowledge of CLIP is transferred to pixel-level understanding task through additional training.

### A.4 MHSG

In this study, we propose MHSG which is a novel multimodal learning framework that enables our SEAL to learn semantic-rich event representations across multiple levels of granularity. In the following, we provide additional details and visualizations for the proposed MHSG framework.

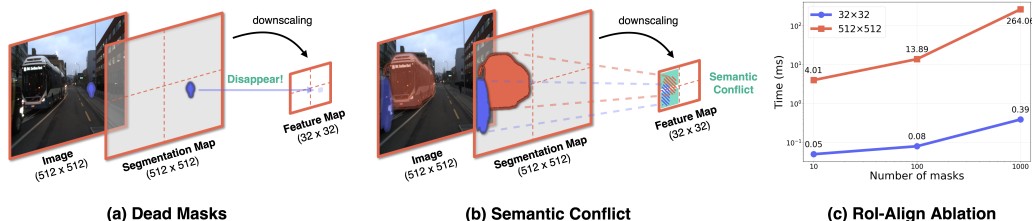

Figure 7: **(a) Dead Masks:** Small size of masks often disappear in the pooling operation due to the severe downscaling. **(b) Semantic Conflict:** Multiple masks with different semantics are projected into same feature patches, which leads to indiscriminative events representation. **(c) RoI-Align operation** in high-resolution feature maps substantially increases inference time (**Red**), especially when the large number of masks are given.

**Motivation.** Our MHSG is closely related to superpixel-driven contrastive learning (Hénaff et al., 2021; Sautier et al., 2022) where superpixels group the pixels into conceptually meaningful regions and transfer the corresponding semantics into other modalities, *i.e.*, point-cloud (Lee et al., 2024; Wang et al., 2025b), sensory data (Guan et al., 2024; Wang et al., 2025a), medical images (Wang et al., 2022) *etc*. This superpixel-driven approach can facilitate the events representation learning, which is inherently challenging due to the sparse and asynchronous nature of events. For example, Kong et al. (2024) converts raw events into frame-like representations such as voxel grids or gray-scale images and then groups the pixels into higher granularity, mitigating the sparsity of events. MHSG extends this strategy to enable hierarchical event representation learning by aggregating events into multiple levels of granularity including *semantic*-level, *instance*-level and *part*-level.

**Implementation Details.** MHSG is comprised of two types of guidance: *visual guidance* and *text guidance*. We use OpenSeg (Ghiasi et al., 2022) to extract visual features from the predicted masks, and Osprey (Yuan et al., 2024b) is used to generate mask captions that serve as text guidance. Osprey supports two caption-generation modes, namely short and long. We generate both short and long captions where short captions are used as guidance and long captions are given to the backbone feature enhancer for langauge fusion during the training. To generate the masks at multiple levels of granularity, we use a custom SAM proposed by Qin et al. (2024). We curate 24,032 event-image pairs from the two widely-used ESS datasets (Alonso & Murillo, 2019; Sun et al., 2022) which are named as ***Mixed-24K*** where all of the pairs are used to generate MHSG. Tab. 8 presents the specific configurations of MHSG.

**Visualization.** We visualize the generated masks and captions of MHSG module in Fig. 11-12.

## A.5    SEAL

Here, we provide the architectural details of our SEAL and supplement the motivation of the Spatial Encoding module with additional ablation.

**Model Architecture.** Given the voxelized events $I^{evt}$ with $512 \times 512$ spatial resolution, EventSAM backbone with ViT-B (Dosovitskiy et al., 2020) based architecture outputs $32\times$ downscaled event embeddings. Events tokens $\mathbf{I}^{evt}$ are fed into the *Backbone Feature Enhancer* which consists of 6 layers of multimodal fusion module. Each fusion module comprises a self-attention layer, a cross-attention layer, and an MLP layer. The text-domain representations are given to the cross-attention layer as keys and values, and $\mathbf{I}^{evt}$ serve as queries, fusing the language prior explicitly. Following Kirillov et al. (2023); Li et al. (2022), the self-attention layers in the first five fusion modules adopt $14 \times 14$ windowed attention, while the standard global attention is adopted in the last self-attention layer to allow token-wise global fusion. A *Spatial Encoding Module* with $\mathrm{concat}(\cdot, \cdot)$ operation is followed by MLP layers to reduce the feature dimension. Finally, a single cross-attention layer with masked attention (Cheng et al., 2022) is used as a *Mask Feature Enhancer* to further refine both semantic and spatial priors.

**Motivation of Spatial Encoding.** Given the event representation $I^{evt} \in \mathbb{R}^{H \times W}$, backbone feature enhancer outputs language-fused feature maps $\hat{\mathbf{I}}^{evt} \in \mathbb{R}^{\frac{H}{32} \times \frac{W}{32}}$ which are 32 times downscaled from the original resolution. RoI-Align is then applied to the feature maps, pooling the feature of the predicted mask $M \in \mathbb{R}^{H \times W}$ in high resolution. Despite its effectiveness, sole reliance of the pooling

Table 9: **Quantitative results in cross-dataset evaluation.**

| Method | *DDD17-Ins* ➔ *DSEC11-Ins* | | | | | | *DSEC11-Ins* ➔ *DDD17-Ins* | | | | | |
| | Point | | | Box | | | Point | | | Box | | |
| | AP | $AP_{50}$ | $AP_{25}$ | AP | $AP_{50}$ | $AP_{25}$ | AP | $AP_{50}$ | $AP_{25}$ | AP | $AP_{50}$ | $AP_{25}$ |
| `frame2recon` (He et al., 2016) | 14.9 | 15.3 | 15.7 | 15.5 | 16.5 | 16.8 | 23.0 | 25.4 | 27.1 | 25.0 | 28.6 | 29.7 |
| `frame2voxel` (Rebecq et al., 2019) | 15.9 | 17.8 | 19.1 | 17.3 | 20.3 | 21.1 | 24.3 | 28.4 | 31.1 | 28.1 | 35.1 | 36.8 |
| `frame2spike` (Kim et al., 2022b) | 16.3 | 18.6 | 20.1 | 17.6 | 21.3 | 22.2 | 23.2 | 26.8 | 29.0 | 26.0 | 31.4 | 32.8 |
| **SEAL (Ours)** | **20.4** | **26.6** | **30.8** | **24.4** | **33.7** | **36.2** | **30.1** | **39.4** | **44.2** | **35.2** | **48.4** | **51.7** |
| | (+4.1) | (+8.0) | (+10.7) | (+6.8) | (+12.4) | (+14.0) | (+5.8) | (+11.0) | (+13.1) | (+7.1) | (+13.3) | (+14.9) |

layer to obtain mask features results in two critical issues as mentioned in Sec. 4.2. **First**, small sized masks often disappear when we match the resolution of binary mask into low-resolution of the feature map in the pooling operation, consequently resulting in zero vectors. We name this problem ***Dead Masks***, and illustrates it in Fig. 7a. **Second**, as we illustrate in Fig. 7b, multiple masks with different semantics (*Red* and *Blue*) are often projected into the same feature patches (*Green*) due to severe downscaling. We refer to this issue as ***Semantic Conflict*** which yields indistinguishable mask features across distinct semantics (*cf.* Fig. 4a). The naive solution is to upscale the resolution of the feature map to the mask resolution prior to the pooling operation. However, applying RoI-Align on high-resolution feature maps substantially increases the inference time, particularly when processing a large number of masks. In Fig. 7c, we report the inference time of the RoI-Align operation at different spatial resolutions across varying numbers of masks. Specifically, we estimate the runtime of the pooling operation at mask resolution ($512 \times 512$) and feature map resolution ($32 \times 32$), with the channel dimensionality fixed at 768. We run the experiment in NVIDIA RTX 6000 Ada. Given the 10 number of masks, high-resolution pooling requires 4.01ms inference time, which is roughly 80 times longer than low-resolution pooling time (0.05ms). The pooling time increases drastically to 264.06ms in high resolution when the number of masks increases to 1,000. However, the low-resolution pooling time remains largely unchanged ($< 0.4$ms) on different mask numbers. Overall, we conclude that pooling in low resolution is essential to achieve real-time inference. To mitigate the *Dead Masks* and *Semantic Conflict* issues without upscaling the feature maps, we introduce the ***Spatial Encoding*** module to complement the mask feature with geometry information derived from SAM mask decoder. The 2nd row of Tab. 5 and Fig. 4b shows that spatial encoding leads to performance improvement by yielding the more discriminative event feature space.

# B    ADDITIONAL EXPERIMENT RESULTS

In the following, we present additional experimental results to supplement the effectiveness and efficiency of our framework. In Sec. B.1, we evaluate the cross-dataset generalizability of our framework compared to the baselines. In Sec. B.2, we present SEAL++ which is the variant of our SEAL to support generic spatiotemporal EIS that does not take the visual prompts from the users. In Sec. B.3, we further exhibit the breakdown comparison in runtime and parameter size between our framework and the proposed baseline. Finally, in Sec. B.4 and Sec. B.5, we show additional qualitative results with class-wise EIS performance and quantitative segmentation results of our fremwork across *noun*-level and *sentence*-level text conditions.

## B.1    CROSS-DATASET EVALUATION

Tab. 9 analyzes cross-dataset adaptation of our SEAL against event backbone baselines. All methods are trained on event-image pairs from the *DDD17-Ins* dataset and evaluated on the *DSEC11-Ins* benchmark, and vice versa. Our SEAL shows significantly better semantic generalizability compared to baselines trained in the same setting on both of *DSEC11-Ins* and *DDD17-Ins* benchmarks.

## B.2    PROMPT-FREE OV-EIS

Here, we present the additional variant of our SEAL, named as **SEAL++**, to support the prompt-free spatiotemporal OV-EIS by combining the event-based object detection model into our framework.

**Model Architecture.** Our overall model architecture is illustrated in Fig. 8. We attach the spatiotemporal object detection branch (*Red*) on top of pretrained backbone of our SEAL. Specifically, we follow DEOE (Zhang et al., 2025), a class-agnostic event-based detector that leverages the RVT architecture (Gehrig & Scaramuzza, 2023). Our object detection branch consists of a single layer of

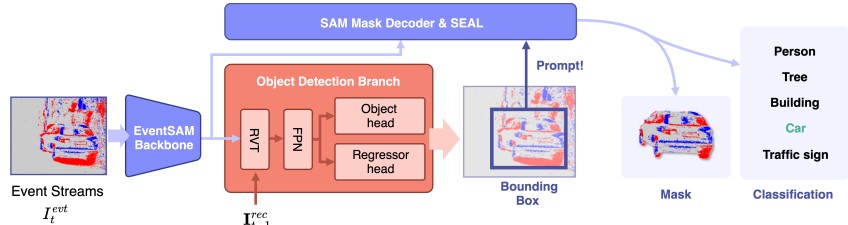

Figure 8: **Overall architecture of our SEAL++.**

Table 10: **CAOD results on DSEC-*Detection* (Gehrig & Scaramuzza, 2024) dataset.**

| Method | AUC | $AR_{10}$ | $AR_{30}$ | $AR_{50}$ | $AR_{100}$ | $AR_{300}$ | Time (ms) |
|---|---|---|---|---|---|---|---|
| DEOE (Zhang et al., 2025) | 41.35 | 50.0 | 52.0 | 52.5 | 53.3 | 53.8 | **6.16** |
| **SEAL++ (Ours)** | **43.32** | **52.9** | **54.5** | **55.0** | **55.3** | **55.6** | 11.61 |
| | (+1.97) | (+2.9) | (+2.5) | (+2.5) | (+2.0) | (+1.8) | |

RVT and FPN (Lin et al., 2017) that are followed by two distinct detection heads: 1) *Disentangled Obj. Head* and 2) *Dual Regressor Head*. The RVT block ingests event features $\mathbf{I}_t^{evt}$ extracted from the backbone at the current timestamp $t$, together with the recurrent event embeddings $\mathbf{I}_{t-1}^{rec}$ propagated from the previous timestamp $t-1$. It outputs temporal-aware event features $\mathbf{I}_t^{rec}$ which are then upsampled by the FPN layer and processed by two detection heads, and finally obtaining the class-agnostic bounding boxes for the detected proposals. Here, the final bounding boxes are acquired by combining the objectness scores from *Disentangled Obj. Head* and potential bounding boxes derived from *Dual Regressor Head*. Refer to the original paper (Zhang et al., 2025) for more specific details about the detection module.

**Discussion.** RVT (Gehrig & Scaramuzza, 2023) and DEOE (Zhang et al., 2025) use 4 layers of the RVT block to interpret spatiotemporal events. However, since our SEAL has a strong backbone which is distilled from the image segmentation foundation model (Kirillov et al., 2023), we empirically find that adopting a single layer of the recurrent network is enough to generate high quality bounding boxes when it is built on top of our backbone (*cf.* Tab. 10). This design choice endows our SEAL with prompt-free EIS capabilities without sacrificing the real-time inference and parameter-efficient architecture.

**Training.** Building the detection module on top of our pretrained SEAL, we train only the detection branch while keeping all other components frozen. We follow the DEOE (Zhang et al., 2025) framework to train the detection branch where the objective terms consist of IoU loss, object confidence loss and spatial consistency regularization term. To secure training efficiency and convergence speed, we set the sequence length as 5 and batch size as 6. The maximum learning rate is set to $2 \times 10^{-4}$ and optimization is performed using the Adam optimizer (Kingma & Ba, 2014) for 300,000 iterations. Refer to the DEOE paper (Zhang et al., 2025) for more details in training and objective terms.

**Inference.** Given the event representation $I^{evt}$, the SAM mask decoder, our SEAL and the object detection branch share a common backbone. Given the event features outputted from the backbone of our SEAL, the detection branch first generates class-agnostic bounding boxes. These bounding boxes are then used as box prompts for mask generator and mask classifier to get the segmentation masks and class labels.

**Dataset.** We train the detection branch of our SEAL++ on the training split of DSEC-*Detection* (Gehrig & Scaramuzza, 2024) with only using the ground-truth bounding boxes. DSEC-*Detection* is built on top of DSEC (Gehrig et al., 2021) by annotating the bounding boxes with the pretrained multiobject tracking module QDTrack (Fischer et al., 2023; Pang et al., 2021). It contains 8 semantic classes which are: `pedestrians`, `rider`, `cars`, `bus`, `truck`, `bicycle`, `motorcycle` and `train`.

**Experimental Settings.** We conduct experiments in two-settings: 1) Class-Agnostic Object Detection (CAOD) and 2) closed-set generic EIS with 8 semantic classes. In the CAOD setting, we compare our SEAL++ with DEOE which is pretrained on DSEC-*Detection*. On the evaluation of the closed-set generic EIS, SEAL++ is compared with the composition of DEOE and proposed baselines in Sec. 3, where the DEOE serves as the class-agnostic detection module and the baselines are exploited as

Table 11: **Quantitative results on generic EIS in DSEC-*Detection* (Gehrig & Scaramuzza, 2024).**

| Method | AP | $AP_{50}$ | $AP_{75}$ | $AP_{large}$ | $AP_{medium}$ | $AP_{small}$ | Time (ms) |
|---|---|---|---|---|---|---|---|
| DEOE + OVSAM (Yuan et al., 2024a) | 11.7 | 17.4 | 11.9 | 11.3 | 11.1 | 0.7 | 361.2 |
| DEOE + CLIP (Radford et al., 2021) | 12.1 | 18.2 | 12.8 | 12.9 | 13.0 | 1.2 | 335.59 |
| DEOE + OVSeg (Jia et al., 2021) | 12.4 | 18.7 | 14.3 | 12.7 | 13.1 | 1.4 | 299.02 |
| DEOE + MaskCLIP (Zhou et al., 2022) | 12.2 | 18.3 | 14.2 | 12.4 | 12.7 | 1.1 | 302.17 |
| DEOE + OpenSeg (Ghiasi et al., 2022) | 16.1 | 21.3 | 18.2 | 17.2 | 16.9 | 3.2 | 433.17 |
| DEOE + MaskCLIP++ (Zeng et al., 2024) | 16.3 | 21.8 | 18.5 | 17.6 | 17.0 | 3.4 | 400.77 |
| DEOE + Mask-Adapter (Li et al., 2024) | 15.6 | 21.3 | 17.9 | 16.9 | 16.5 | 2.2 | 293.65 |
| DEOE + frame2recon (He et al., 2016) | 14.8 | 20.9 | 16.2 | 15.1 | 14.5 | 1.7 | 284.51 |
| DEOE + frame2voxel (Rebecq et al., 2019) | 14.4 | 20.4 | 15.8 | 14.7 | 14.4 | 1.8 | 94.35 |
| DEOE + frame2spark (Kim et al., 2022b) | 14.2 | 20.1 | 15.3 | 14.4 | 14.3 | 1.4 | 328.80 |
| DEOE + SEAL | 16.4 | 22.1 | 18.8 | 17.8 | 17.4 | 3.4 | 28.44 |
| **SEAL++ (Ours)** | **17.8** | **23.6** | **19.2** | **18.5** | **18.2** | **3.5** | **24.12** |
| | (+1.4) | (+1.5) | (+0.4) | (+0.7) | (+0.8) | (+0.1) | |

mask generator and mask classifier. Furthermore, we adopt the naive integration of DEOE and our SEAL as an additional comparison, where DEOE and SEAL serve as individual module without sharing the backbone. All evaluations are conducted on three testing sequences of DSEC-*Detection* dataset comprising zurich_city_13_a, zurich_city_14_c and zurich_city_15_a. Since DSEC-*Detection* does not provide segmentation mask of events to evaluate EIS, we annotate the ground-truth masks using the paired RGB images and SAM (Kirillov et al., 2023). Ground-truth bounding boxes from the dataset are used as prompts for mask generation.

**Metrics.** In the CAOD setting, we adopt Average Recall (AR) and area under the curve (AUC) as the main metrics following the existing CAOD works (Saito et al., 2022; Kim et al., 2022a; Zhang et al., 2025; Wang et al., 2021c). For the generic EIS evaluation, we adopt Average Precision (AP) from RVT (Gehrig & Scaramuzza, 2023) as the main metric.

**Class-Agnostic Object Detection (CAOD).** Tab. 10 presents the CAOD results on DSEC-*Detection* dataset. Our SEAL++ significantly outperforms DEOE in all metrics by leveraging a stronger backbone endowed with foundational priors from SAM. The tradeoff between performance and inference speed is also observed, where DEOE shows slightly faster inference speed with lightweight backbone. However, the runtime of SEAL++ still exhibits real-time speed with only spending 11.61ms to generate box proposals. Note that operations for mask generation and mask classification are excluded from the runtime estimation since they are not necessary for CAOD.

**Closed-set generic EIS.** Tab. 11 shows the quantitative results on the closed-set generic EIS in the DSEC-*Detection* dataset. We make the following observations: 1) Our SEAL++ shows superior performance compared to the naive combination of DEOE and SEAL (DEOE + SEAL) since our SEAL++ can provide better quality of bounding boxes compared to DEOE as shown in Tab. 10. 2) The performance gap between DEOE + SEAL and the baselines in the Hybrid category is less pronounced than in the evaluations on our EIS benchmarks reported in the main paper. We empirically find that SEAL struggles to distinguish "car" class from other semantics within the "vehicle" category when they are provided as joint text conditions. This is likely from class imbalance in the training sequences, where "car" class dominates the vehicles in the scene while other vehicle types are rare. Since most of the classes in DSEC-*Detection* dataset are fine-grained subclasses within the "vehicle" category, our model frequently produces false positives for the "car" class by mislabeling other vehicles as cars. In contrast, the Hybrid baselines exhibit robust recognition under the vehicle-dominated label space with narrowing performance gap since they use mask classifiers trained on large-scale annotated datasets. 3) By adopting the single-backbone architecture design, our SEAL++ achieves the fastest inference time with parameter-efficient architecture at 24.12ms to process single timestamp.

### B.3 EFFICIENCY

We supplement the breakdown comparisons in runtime and parameters size between our SEAL and all of the proposed baselines. Specifically, Tab. 12 shows the runtime breakdown along the model configurations. We observe that events reconstruction via E2VID (Rebecq et al., 2019) is the main bottleneck of the baselines in the ARCDG and Hybrid categories due to the recurrent reconstruction process. This underscores the importance of excluding the reconstruction operation to enable the real-time events understanding. Our SEAL shows the fastest runtime by only taking 6.16ms in the mask classification step. In Tab. 13, we further report a detailed parameter-size breakdown to

Table 12: **Runtime breakdown with model configurations.**

| Method | Frame Reconstruction | Mask Generator | Mask Classifier | Frame Reconstruction | Mask Generator | Mask Classifier | Total (ms) |
|---|---|---|---|---|---|---|---|
| | **Model Configurations** | | | **Inference Time (ms)** | | | |
| OVSAM (Yuan et al., 2024a) | | OVSAM | | | 102.27 | | 355.04 |
| CLIP (Radford et al., 2021) | E2VID (Rebecq et al., 2019) | EventSAM (Chen et al., 2024) | CLIP | 252.77 | 16.12 | 60.54 | 329.43 |
| OV-Seg (Liang et al., 2023) | | | OV-Seg | | | 23.97 | 292.86 |
| MaskCLIP (Zhou et al., 2022) | | | MaskCLIP | | | 27.12 | 296.01 |
| OpenSeg (Ghiasi et al., 2022) | | | OpenSeg | | | 158.12 | 427.01 |
| MaskCLIP++ (Zeng et al., 2024) | | | MaskCLIP++ | | | 125.72 | 394.61 |
| Mask-Adapter (Li et al., 2024) | | | Mask-Adapter | | | 18.6 | 287.49 |
| Frame2Recon (He et al., 2016) | | | Frame2Recon | | | 9.46 | 278.35 |
| Frame2Voxel (Rebecq et al., 2019) | ✗ | | Frame2Voxel | ✗ | | 72.07 | 88.19 |
| Frame2Spike (Kim et al., 2022b) | | | Frame2Spike | | | 306.52 | 322.64 |
| **SEAL (Ours)** | | | SEAL | | | 6.16 | **22.28** |

Table 13: **Parameters size breakdown with model configurations.**

| Method | Frame Reconstruction | Mask Generator | Mask Classifier | Frame Reconstruction | Mask Generator | Mask Classifier | Total (M) |
|---|---|---|---|---|---|---|---|
| | **Model Configurations** | | | **Params Size (M)** | | | |
| OVSAM (Yuan et al., 2024a) | | OVSAM | | | 304 | | 314.7 |
| CLIP (Radford et al., 2021) | E2VID (Rebecq et al., 2019) | EventSAM (Chen et al., 2024) | CLIP | 10.7 | 91.3 | 427.9 | 529.9 |
| OV-Seg (Liang et al., 2023) | | | OV-Seg | | | 428.4 | 530.4 |
| MaskCLIP (Zhou et al., 2022) | | | MaskCLIP | | | 427.9 | 529.9 |
| OpenSeg (Ghiasi et al., 2022) | | | OpenSeg | | | 126.4 | 228.4 |
| MaskCLIP++ (Zeng et al., 2024) | | | MaskCLIP++ | | | 199.7 | 301.7 |
| Mask-Adapter (Li et al., 2024) | | | Mask-Adapter | | | 214.7 | 316.7 |
| Frame2Recon (He et al., 2016) | | | Frame2Recon | | | 39.7 | 141.7 |
| Frame2Voxel (Rebecq et al., 2019) | ✗ | | Frame2Voxel | ✗ | | 17.8 | 109.1 |
| Frame2Spike (Kim et al., 2022b) | | | Frame2Spike | | | 4.6 | **95.9** |
| **SEAL (Ours)** | | | SEAL | | | 7.8 | 99.1 |

highlight the parameter efficiency of our framework. Most parameters of our SEAL reside in the ViT-B backbone where the mask classifier adopts lightweight design with 7.8M parameters.

## B.4 QUANTITATIVE RESULTS

We present the detailed quantitative comparisons with per-class EIS results across DSEC-*Part*, DDD17-*Ins*, DSEC11-*Ins* and DSEC19-*Ins* in Tab. 14-17.

## B.5 QUALITATIVE RESULTS

In this section, we present diverse visualization results of our SEAL and SEAL++ across diverse settings, showing their capabilities to understand the multiple levels of granularity in response to free-from language.

**Visualization results of SEAL with box prompts.** Fig. 13 shows the OV-EIS capabilities of our SEAL based on the box prompts. We give multiple bounding boxes that cover all of the objects in the current frame and ask our SEAL to classify the correct instance masks accroding to the given free-form language query. Following Qin et al. (2024), we attach predefined canonical phrases which are "object", "things", "stuff" and "texture" to the text query, making the single language query to classification setting among 5 number of classes: 1 free-form language query + 4 canonical phrases. Our visualization results demonstrate that our SEAL can recognize the diverse instances in response to both noun-level and sentence-level queries.

**Visualization results of SEAL with point prompts.** In Fig. 14, we give single point prompt to our SEAL and ask to classify the generated mask in the multiple levels of granularity. Since the mask generator of SEAL follows the architecture of SAM, it generates three masks with different granularity according to the single point prompt. We use two of them for the visualization, the coarsest and the finest. For the classification, we use DSEC11-*Ins* class setting for the coarse mask

Table 14: **The per-class segmentation results (AP) on DSEC-*Part* benchmark.**

| Method | car wheel | car window | car light | car side mirror | car license plate | building window | building roof | building door | building terrace | car wheel | car window | car light | car side mirror | car license plate | building window | building roof | building door | building terrace |
|---|---|---|---|---|---|---|---|---|---|---|---|---|---|---|---|---|---|---|
| | | | | | *point* | | | | | | | | | *box* | | | | |
| VLPart (Sun et al., 2023) | 8.0 | 13.3 | 3.9 | **0.2** | **2.7** | 71.2 | 4.5 | **7.4** | 4.9 | 12.8 | 16.8 | 4.1 | **0.2** | **4.7** | 74.3 | 9.6 | **15.6** | 6.8 |
| **SEAL (Ours)** | **8.2** | **13.6** | **4.1** | 0.1 | 2.5 | **81.7** | **4.6** | 2.4 | **5.1** | **21.1** | **25.3** | **6.6** | 0.1 | 2.5 | **85.4** | **13.3** | 3.2 | **7.3** |

and DSEC-*Part* class setting for the fine mask. Fig. 14 shows that our SEAL can recognize the objects at both *instance*-level and *part*-level granularities.

**Video visualization of SEAL++.** We further provide video demo of our SEAL++ to demonstrate the spatiotemporal OV-EIS. Video contains visualizations of 1) Class-Agnostic Object Detection (CAOD) in events, 2) OV-EIS with free-form language and 3) generic OV-EIS with closed-set of classes. For the generic OV-EIS, we use two-classes setting where it consists of `vehicle` and `people`.

## C  LIMITATION

While our method has demonstrated open-vocabulary capabilities across diverse linguistic forms and multiple levels of granularity, its recognition still suffers from the inherent challenges posed by the asynchronous nature of events data. Specifically, we empirically find that our SEAL and SEAL++ often fail to detect the objects in region with low event rates. For example, objects far from the camera occupy a minimal footprint in the camera frame, thereby producing highly sparse event streams. This case is easily happened in the driving scenes with straight roads such as DSEC (Gehrig et al., 2021) dataset where they exhibit a single dominant forward vanishing point. Objects (*e.g.*, cars) near the vanishing point or far from the camera position generate few events due to the minimal apparent motion, leading to missed detections.

Furthermore, our model is trained on widely-used outdoor scenes such as DSEC (Gehrig et al., 2021) and DDD17 (Binas et al., 2017), thereby showing the poor recognition capabilities in the indoor scene due to the huge domain gap. However, our MHSG module and training framework of SEAL can be readily extended to other datasets without requiring the additional human annotations. Hence, exploiting more diverse events dataset such as VisEvent (Wang et al., 2023) and COESOT (Tang et al., 2022) will improve the generalizability of our method.

Table 15: **The per-class segmentation results (AP) on DDD17-*Ins* benchmark.**

| Method | construction+sky | object | nature | human | vehicle | construction+sky | object | nature | human | vehicle |
|---|---|---|---|---|---|---|---|---|---|---|
| | | | point | | | | | box | | |
| OVSAM (Yuan et al., 2024a) | 52.0 | 0.7 | 20.1 | 5.0 | 23.0 | 52.6 | 0.9 | 20.5 | 7.5 | 26.2 |
| CLIP (Radford et al., 2021) | 54.8 | 0.6 | 21.4 | 0.7 | 25.6 | 56.6 | 0.6 | 23.4 | 0.8 | 30.0 |
| OVSeg (Jia et al., 2021) | 51.5 | 0.6 | 19.5 | 0.9 | 33.0 | 52.1 | 0.6 | 20.0 | 1.2 | 33.6 |
| MaskCLIP (Zhou et al., 2022) | 51.8 | 0.6 | 20.2 | 0.7 | 24.2 | 52.3 | 0.8 | 21.0 | 0.8 | 25.1 |
| OpenSeg (Ghiasi et al., 2022) | 55.0 | 0.9 | 28.0 | 20.3 | 44.8 | 63.6 | 0.9 | 35.7 | 23.4 | 51.3 |
| MaskCLIP++ (Zeng et al., 2024) | 57.8 | 1.3 | 26.3 | 6.1 | 45.6 | 67.8 | 2.0 | 33.9 | 6.0 | 54.2 |
| Mask-Adapter (Li et al., 2024) | 56.8 | 0.6 | 24.5 | 20.4 | 34.0 | 61.1 | 0.6 | 30.8 | 21.4 | 35.9 |
| frame2recon (He et al., 2016) | 54.8 | 0.7 | 27.1 | 16.6 | 45.1 | 65.2 | 0.7 | 34.0 | 21.8 | 52.2 |
| frame2voxel (Rebecq et al., 2019) | 55.3 | 0.6 | 27.5 | 8.4 | 46.3 | 65.6 | 0.7 | 35.8 | 11.1 | 54.8 |
| frame2spark (Kim et al., 2022b) | 54.4 | 0.6 | 24.6 | 10.1 | 42.4 | 62.9 | 0.6 | 29.8 | 11.2 | 48.8 |
| **SEAL (Ours)** | **59.1** | 0.9 | 27.0 | **24.5** | **50.0** | **70.3** | 1.1 | 33.1 | **28.9** | **57.7** |

Table 16: **The per-class segmentation results (AP) on DSEC11-*Ins* benchmark.**

| Method | building | fence | person | pole | vegetation | car | traffic sign | building | fence | person | pole | vegetation | car | traffic sign |
|---|---|---|---|---|---|---|---|---|---|---|---|---|---|---|
| | | | | point | | | | | | | box | | | |
| OVSAM (Yuan et al., 2024a) | 22.7 | 3.3 | 8.7 | 6.5 | 39.9 | 25.4 | 10.5 | 35.4 | 3.9 | 13.3 | 7.1 | 41.1 | 40.6 | 14.0 |
| CLIP (Radford et al., 2021) | 25.7 | 3.5 | 2.6 | 6.9 | 39.9 | 25.4 | 4.2 | 36.1 | 3.7 | 2.8 | 7.0 | 40.5 | 30.2 | 4.4 |
| OVSeg (Jia et al., 2021) | 30.4 | 3.5 | 2.7 | 7.2 | 40.5 | 24.2 | 4.7 | 39.1 | 3.9 | 2.8 | 7.7 | 42.7 | 24.6 | 5.9 |
| MaskCLIP (Zhou et al., 2022) | 21.8 | 3.3 | 2.6 | 6.3 | 38.5 | 20.9 | 3.9 | 21.9 | 3.4 | 2.7 | 6.4 | 39.5 | 21.0 | 4.0 |
| OpenSeg (Ghiasi et al., 2022) | 25.1 | 3.7 | 15.6 | 6.2 | 43.0 | 40.2 | 9.2 | 31.6 | 4.3 | 17.9 | 6.3 | 46.7 | 47.8 | 10.8 |
| MaskCLIP++ (Zeng et al., 2024) | 25.3 | 4.0 | 10.5 | 10.1 | 45.0 | 35.0 | 14.9 | 30.6 | 5.5 | 12.9 | 11.0 | 52.4 | 44.1 | 21.5 |
| Mask-Adapter (Li et al., 2024) | 27.5 | 3.7 | 12.1 | 7.2 | 41.9 | 27.3 | 6.6 | 36.8 | 4.9 | 14.9 | 7.3 | 46.3 | 31.2 | 7.6 |
| frame2recon (He et al., 2016) | 25.1 | 4.2 | 10.1 | 6.4 | 43.3 | 30.0 | 7.3 | 32.0 | 5.2 | 10.1 | 6.6 | 49.7 | 35.8 | 9.0 |
| frame2voxel (Rebecq et al., 2019) | 26.0 | 3.7 | 8.7 | 6.6 | 42.7 | 32.2 | 6.2 | 33.5 | 4.1 | 8.7 | 6.9 | 49.8 | 38.4 | 7.6 |
| frame2spark (Kim et al., 2022b) | 24.5 | 3.5 | 10.6 | 6.7 | 42.3 | 29.7 | 7.9 | 30.5 | 3.6 | 10.8 | 7.0 | 48.1 | 34.9 | 10.0 |
| **SEAL (Ours)** | 27.1 | **5.0** | **15.6** | **10.2** | 44.9 | 38.4 | **15.9** | 35.2 | **7.5** | **22.8** | **13.1** | **54.0** | 45.7 | **23.5** |

Table 17: **The per-class segmentation results (AP) on DSEC19-*Ins* benchmark.**

| Method | building | fence | pole | traffic light | traffic sign | vegetation | person | rider | car | truck | bus | train | motorcycle | bicycle | building | fence | pole | traffic light | traffic sign | vegetation | person | rider | car | truck | bus | train | motorcycle | bicycle |
|---|---|---|---|---|---|---|---|---|---|---|---|---|---|---|---|---|---|---|---|---|---|---|---|---|---|---|---|---|
| | | | | | | | point | | | | | | | | | | | | | | box | | | | | | | |
| OVSAM (Yuan et al., 2024a) | 23.7 | 3.5 | 6.9 | 1.1 | 9.0 | 37.5 | 11.0 | 0.2 | 24.5 | 2.6 | 1.8 | 0.0 | 0.3 | 1.2 | 34.7 | 4.2 | 7.7 | 1.6 | 12.8 | 38.7 | 15.0 | 0.2 | 34.8 | 6.3 | 2.7 | 0.0 | 0.7 | 2.4 |
| CLIP (Radford et al., 2021) | 25.1 | 3.6 | 6.9 | 0.4 | 4.0 | 37.8 | 2.7 | 0.2 | 25.7 | 3.3 | 3.7 | 0.0 | 0.1 | 0.5 | 34.8 | 4.1 | 7.5 | 0.4 | 4.5 | 38.9 | 2.8 | 0.2 | 28.9 | 3.9 | 4.5 | 0.0 | 0.1 | 1.0 |
| OVSeg (Jia et al., 2021) | 28.5 | 3.6 | 6.8 | 0.4 | 5.0 | 38.9 | 2.5 | 0.2 | 26.0 | 4.3 | 3.8 | 0.0 | 0.1 | 0.2 | 38.8 | 4.1 | 7.2 | 0.4 | 5.5 | 41.6 | 2.7 | 0.2 | 26.8 | 6.2 | 4.6 | 0.0 | 0.1 | 0.2 |
| MaskCLIP (Zhou et al., 2022) | 21.5 | 3.3 | 6.4 | 0.3 | 3.6 | 36.3 | 2.6 | 0.2 | 18.3 | 0.2 | 0.8 | 0.0 | 0.1 | 0.2 | 21.8 | 3.5 | 6.6 | 0.4 | 3.8 | 37.1 | 2.7 | 0.2 | 19.1 | 2.0 | 0.8 | 0.0 | 0.1 | 0.2 |
| OpenSeg (Ghiasi et al., 2022) | 25.6 | 3.8 | 6.3 | 6.2 | 6.1 | 39.4 | 16.4 | 0.2 | 35.8 | 5.2 | 5.7 | 0.0 | 0.1 | 1.9 | 31.1 | 4.3 | 6.4 | 6.8 | 6.5 | 42.0 | 18.8 | 0.2 | 40.9 | 8.0 | 7.8 | 0.0 | 0.1 | 9.2 |
| MaskCLIP++ (Zeng et al., 2024) | 21.9 | 3.8 | 10.6 | 2.6 | 12.4 | 38.6 | 12.5 | 0.3 | 30.5 | 4.3 | 13.3 | 0.0 | 1.4 | 3.2 | 27.8 | 4.8 | 13.5 | 3.3 | 15.8 | 49.0 | 15.9 | 0.4 | 38.7 | 5.5 | 16.9 | 0.0 | 1.8 | 4.0 |
| Mask-Adapter (Li et al., 2024) | 26.4 | 3.9 | 7.5 | 1.0 | 7.3 | 40.2 | 14.1 | 0.3 | 28.1 | 5.2 | 5.0 | 0.0 | 0.1 | 2.8 | 36.9 | 5.1 | 7.5 | 1.3 | 7.4 | 45.8 | 15.9 | 0.4 | 29.7 | 7.5 | 5.9 | 0.0 | 0.2 | 3.9 |
| frame2recon (He et al., 2016) | 25.6 | 4.5 | 6.5 | 1.1 | 5.6 | 40.0 | 12.6 | 0.2 | 34.0 | 4.4 | 0.8 | 0.0 | 0.1 | 0.3 | 29.6 | 3.8 | 6.7 | 1.1 | 4.9 | 44.1 | 11.6 | 0.2 | 39.7 | 4.3 | 0.8 | 0.0 | 0.1 | 0.2 |
| frame2voxel (Rebecq et al., 2019) | 26.0 | 3.8 | 6.6 | 2.3 | 4.3 | 40.2 | 10.0 | 0.2 | 35.2 | 4.3 | 0.8 | 0.0 | 0.1 | 0.2 | 33.1 | 4.1 | 6.7 | 3.3 | 4.5 | 46.2 | 10.2 | 0.2 | 41.8 | 7.0 | 0.8 | 0.0 | 0.1 | 0.2 |
| frame2spark (Kim et al., 2022b) | 24.5 | 3.6 | 6.5 | 1.0 | 4.4 | 40.0 | 11.1 | 0.2 | 34.1 | 3.2 | 0.8 | 0.0 | 0.1 | 0.4 | 32.3 | 5.2 | 6.5 | 1.4 | 6.6 | 44.9 | 13.6 | 0.2 | 39.7 | 7.1 | 0.8 | 0.0 | 0.1 | 0.4 |
| **SEAL (Ours)** | 24.7 | 4.9 | 11.8 | 2.7 | 14.7 | 44.5 | 18.7 | 0.2 | 35.4 | 4.1 | 0.8 | 0.0 | 0.1 | 0.6 | 30.7 | 7.1 | 15.4 | 4.5 | 20.7 | 52.5 | 22.4 | 0.2 | 40.6 | 7.3 | 0.8 | 0.0 | 0.1 | 4.4 |

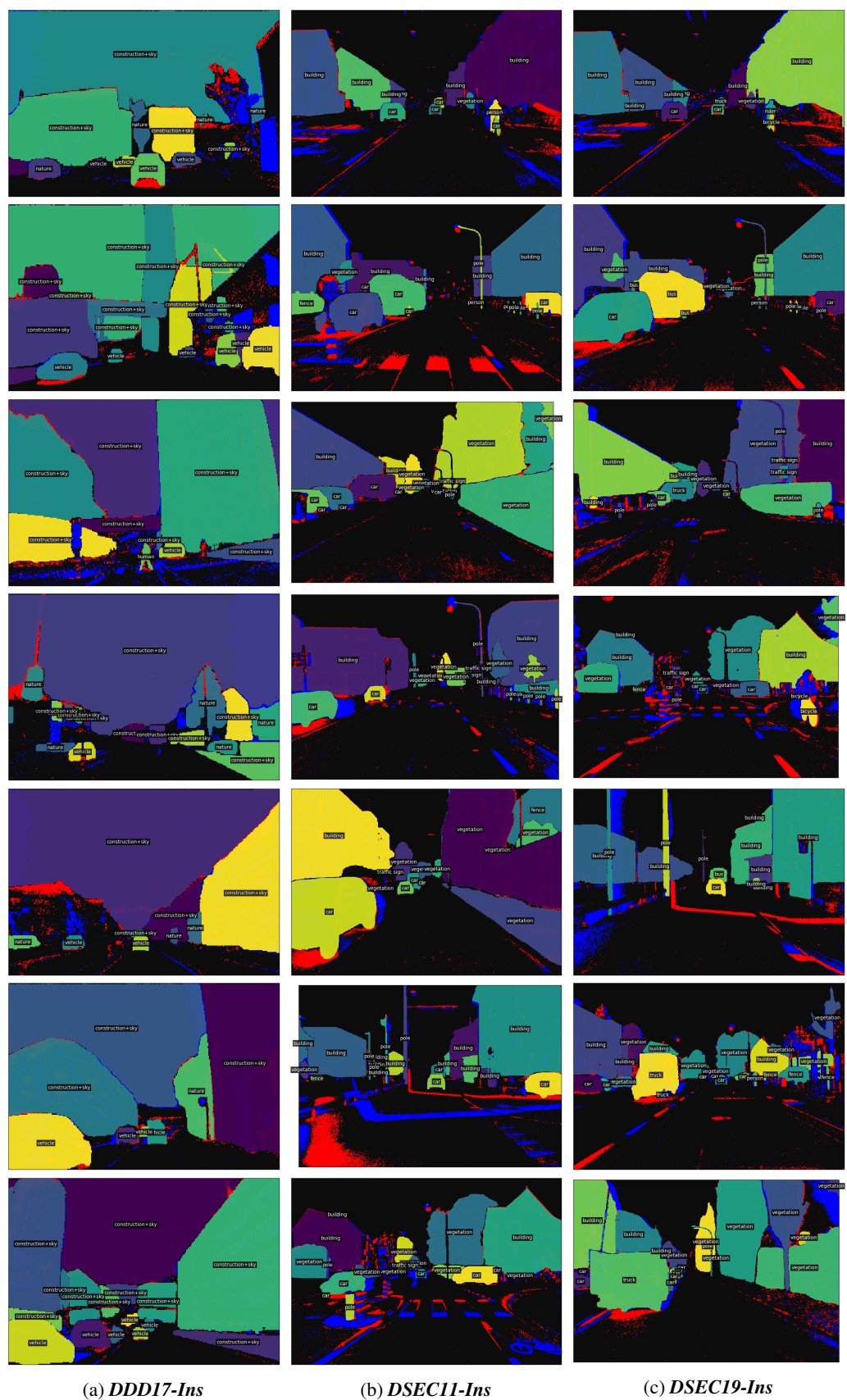

(a) *DDD17-Ins*          (b) *DSEC11-Ins*          (c) *DSEC19-Ins*

Figure 9: **Visualizations for our proposed EIS benchmarks**

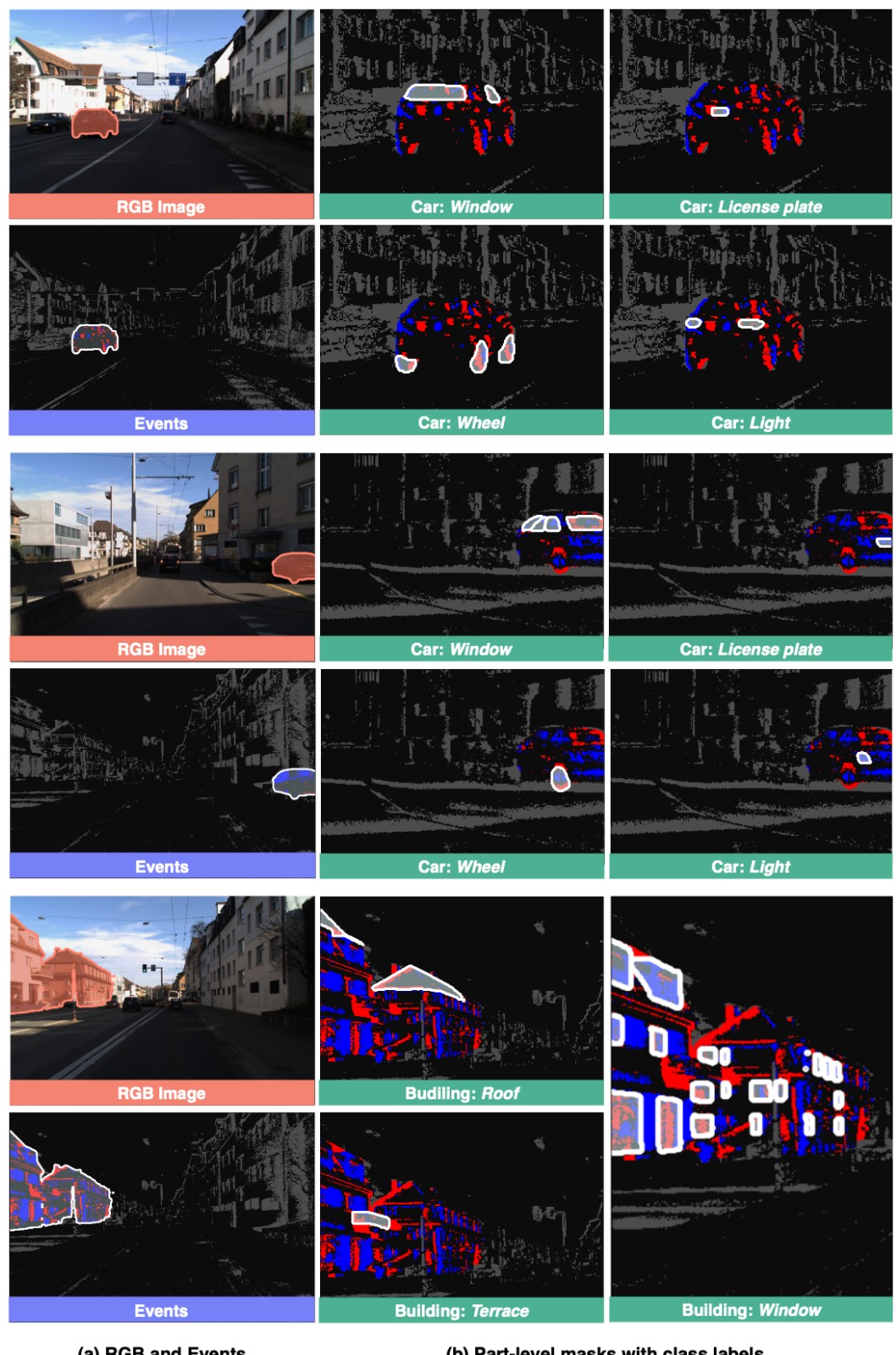

Figure 10: **Visualizations for our *DSEC-Part* benchmark.** All semantic labels are annotated by human.

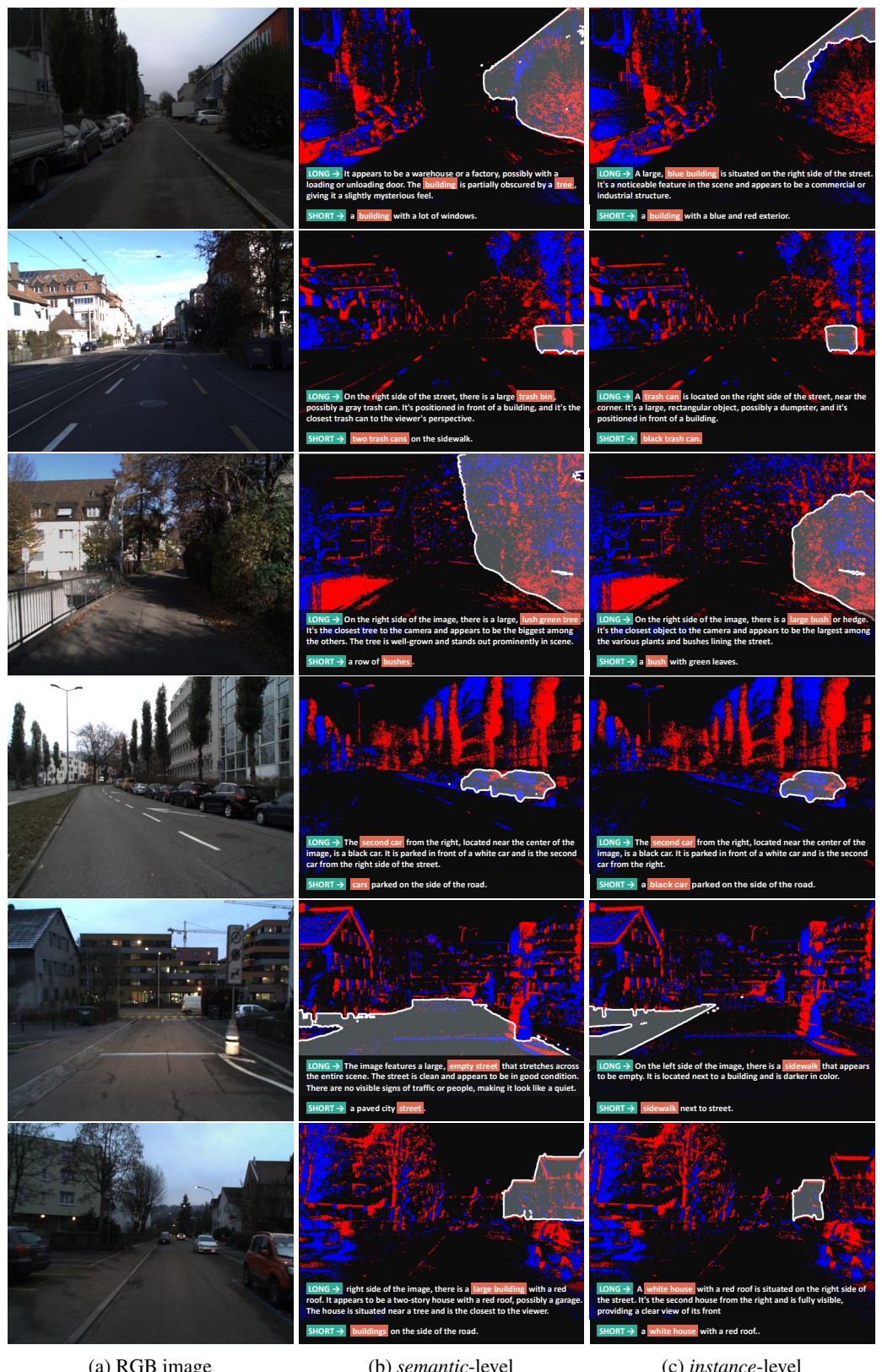

(a) RGB image      (b) *semantic*-level      (c) *instance*-level

Figure 11: **Visualizations of mask captions in *semantic* and *instance* level generated by MHSG module.**

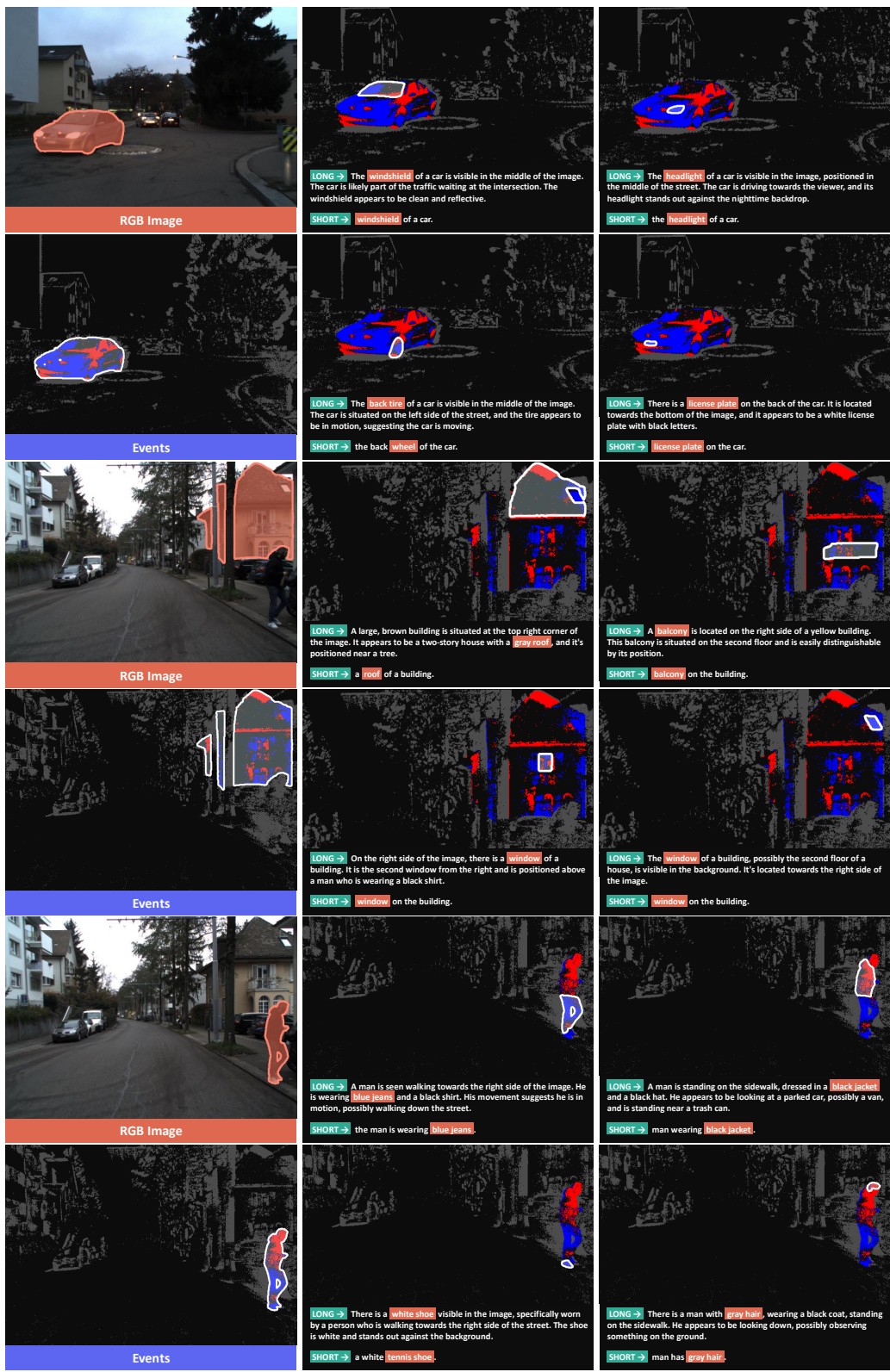

Figure 12: **Visualizations of mask captions in *part*-level generated by MHSG module.**

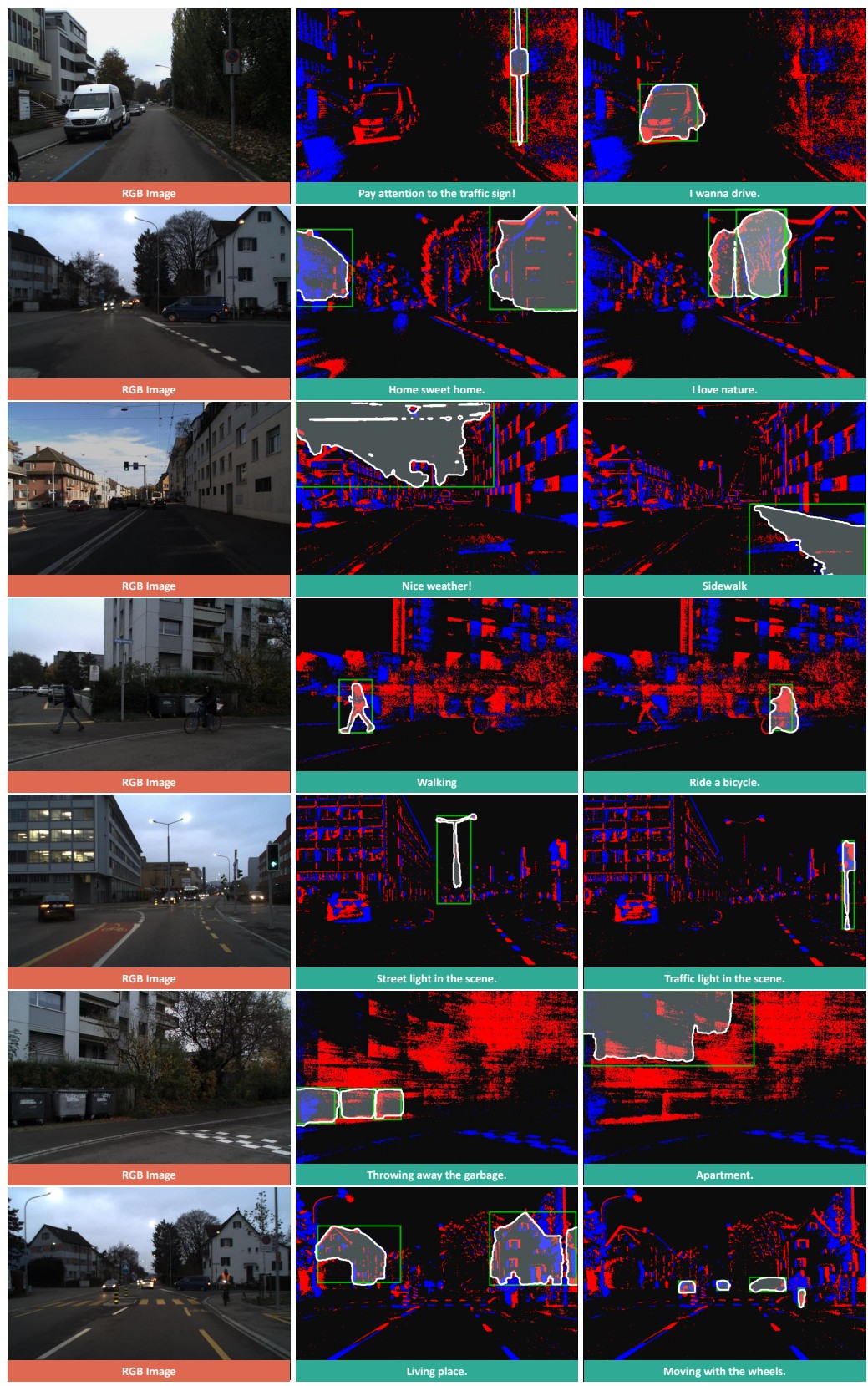

Figure 13: **Qualitative results of our SEAL with box prompts.**

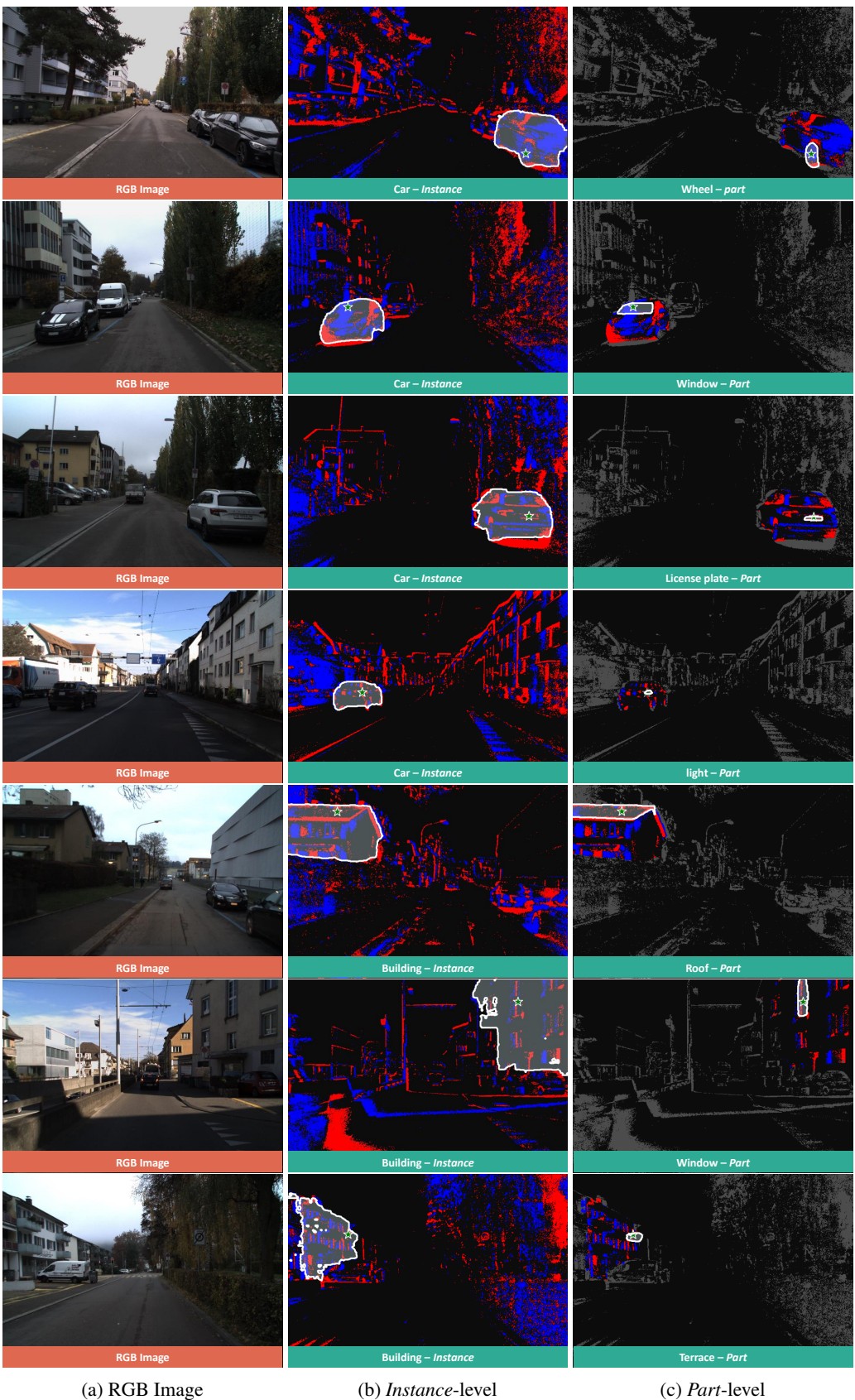

(a) RGB Image      (b) *Instance*-level      (c) *Part*-level

Figure 14: **Qualitative results of our SEAL with point prompts.**

