# OpenReview forum: "Segment Any Events with Language"
_ICLR.cc/2026/Conference — ICLR 2026 Poster_

### Official Review · Reviewer_5RxY · 2025-10-23

**Soundness:** 3
**Presentation:** 3
**Contribution:** 3
**Rating:** 6
**Confidence:** 4

**Summary:**

Authors propose the first text-guided instance segmentation framework for event streams, achieving open-vocabulary segmentation, and introduce four benchmark levels to validate the effectiveness of the proposed model.

**Strengths:**

1.  Proposes SEAL, the first open-vocabulary instance segmentation framework for event streams, achieving effective instance-level segmentation.
2.  Introduces a decoupled training–inference strategy, featuring a complex training framework and a lightweight inference framework.
3.  Constructs a scarce open-vocabulary segmentation dataset and designs four benchmark levels for training and evaluation.

**Weaknesses:**

1. Incomplete related work analysis. In the Open-World Events Understanding section (line 126), the authors discuss event-based vision foundation models such as EventBind and EventCLIP, but fail to include analyses of event-based vision MLLMs such as EventGPT and EventVL, which also fall within this domain.
2. Dataset availability concerns. For open-vocabulary segmentation in event streams, the authors propose a complex training framework. However, the scarcity of open-vocabulary segmentation datasets in this domain remains a key limitation. The authors should clearly state whether their dataset will be publicly released.
3. Fairness of experimental analysis. In the main experiments, several RGB-based models are used for comparison. It is unclear whether these models were fine-tuned on the proposed dataset (with event-image representations). If not, the comparison is unfair, and results after fine-tuning should be reported.
4. Unclear experimental presentation. The authors introduce four benchmark levels for open-vocabulary segmentation, which vary significantly in word-level and semantic-level difficulty. In Table 1, the evaluation for each benchmark level should be clearly separated and 5. Dataset quality issues. For semantic-level open-vocabulary segmentation, the authors mainly rely on the DSEC dataset, where all scenes are road environments. These scenes are highly repetitive, and event streams lack color and texture information, leading to low semantic diversity and a risk of overfitting. The authors should clarify how they mitigate or address these issues.

**Questions:**

See Weaknesses.

---

> ### Author Response · Authors · 2025-11-22
> **Response to Reviewer 5RxY (1)**
>
> ## Response for Reviewer `5RxY`
>
> We sincerely appreciate the reviewer `5RxY` for the careful reading of our work and for recognizing our paper as the first approach to address Open-Vocabulary Event Instance Segmentation (OV-EIS), as well as for acknowledging our effort in curating four benchmarks to thoroughly assess OV-EIS performance. Below, we respond to your comments in details.
>
> ---
>
> > `Q1`: "Discussion for events-LLM works such as EventGPT and EventVL is necessary as related works".
>
> **A**: Thank you for pointing this out and for your valuable feedback. We will add concrete discussion about events-LLM works in *Sec. 2. Related Works* in the camera ready version.
>
> ---
>
> > `Q2`: "Will you make benchmarks public?"
>
> **A**: This is indeed an important point. We believe that open-sourcing both the code and the benchmark to facilitate reproducibility is essential for making a meaningful contribution to the research community. Hence, the code for SEAL, proposed baselines and curated benchmarks will be all publicly available upon acceptance of the paper. We will revise the Reproductibility Statement to clarify this in the camera ready version.
>
> ---
>
> > `Q3`: "The performance of fine-tuned RGB-based models should also be reported."
>
> **A**: Thank you for your thoughtful and valuable comments. In the main experiment, we discuss the direct transfer of RGB-based models into the event modalities without further fine-tuning them. We explain the motivation underlying our proposed baselines below.
>
> **No existing baseline for OV-EIS**: Since our SEAL is the **first** framework to address OV-EIS, there is no existing baseline we can compare the performance. Hence, we need to build the **most straightforward approaches** that enable OV-EIS as our baselines **by combining the existing image or events modules**. First, we define two types of module which are indicated as *Red* and *Blue* colors in Fig. 2.
>
> * **RGB-based models + E2VID (*Red*)**: This baseline is designed to evaluate the zero-shot transfer of RGB-based segmentation models into the event modalities. Here, we treat RGB-based models as a ***generalist*** where it is trained by a large scale of image datasets with rich human annotations (*cf.* Tab. 7) which are readily obtained in the image domain. To enable the direct inference of RGB models on events modality, we reconstruct the intensity images through E2VID [a]. Kindly note that showing the direct transfer of RGB models with E2VID has been widely adopted as baselines in events perception by the previous literatures [b, c] due to its simplicity.
> * **Events-adapted models (*Blue*)**: This module adopts domain adaptaion approach where the neural network is adapted to events domain through feature distillation via events-image pairs. Specifically, the rich representations obtained from the RGB generalist are distilled into events network in an annotation-free manner using only event–image pairs.  We use EventSAM [c] when the events model serves as a mask segmentor, and adopt the OpenESS [b] framework when the events model functions as a mask classifier. Kindly note that annotation-free setup in this module reflects the scarcity of human annotations in event data compared to image datasets (*cf.* Tab. 7).
>
> We construct the baselines by leveraging aforementioned modules. All baselines follow two stages framework: 1) Mask Segmentor -> 2) Mask Classifier. By trying the diverse combinations of aforementioned two modules (RGB and Events) within the two-stage pipeline, we design four types of baselines (*cf* Fig. 2a-2d) which can be summarized as:
>
> | Types | Mask Segmentor | Mask Classifier|
> | --------------------------------- | --------------- | --------------- |
> | Fig. 2a | RGB (*Red*) | RGB (*Red*)
> | Fig. 2b-2c | Events (*Blue*) | RGB (*Red*)
> | Fig. 2d | Events (*Blue*) | Events (*Blue*)
>
> **Table. a. Summary of our proposed baselines.**
>
> Overall, we understand the reviewer's concern and further present the comparison with fine-tuned RGB-models through events-image represenations. Specifically, inspired by OpenESS, we adopt a linear probing (LP) approach by attaching multiple MLP layers on top of the instance-level events features derived from RGB-based methods. The MLP layers are trained using the same set of event–image pairs as SEAL in an annotation free manner, while the remaining parts are kept fixed to preserve its generalizability.

---

> > ### Author Response · Authors · 2025-11-22
> > **Response to Reviewer 5RxY (2)**
> >
> > (continue from `Q3`...)
> >
> > | Method                            | Point AP        | Box AP          |
> > | --------------------------------- | --------------- | --------------- |
> > | OVSAM + LP     | 22.1            | 23.3           |
> > | CLIP + LP      | 24.4            | 25.1            |
> > | OVSeg + LP        | 23.4            | 24.6           |
> > | MaskCLIP + LP     | 21.2            | 24.5            |
> > | OpenSeg + LP      | 30.5            | 36.9            |
> > | MaskCLIP++ + LP   | 28.8            | 34.4            |
> > | Mask-Adapter + LP | 28.5            | 32.2            |
> > | **SEAL (Ours)** | **32.3** | **38.2** |
> >
> > **Table b. DDD17-*Ins* results under point- and box-prompt settings**
> >
> > | Method                            | Point AP        | Box AP          |
> > | --------------------------------- | --------------- | --------------- |
> > | OVSAM + LP     | 17.2            | 23.1           |
> > | CLIP + LP      | 17.4            | 19.4            |
> > | OVSeg + LP        | 18.3            | 20.1           |
> > | MaskCLIP + LP     | 15.1            | 18.2            |
> > | OpenSeg + LP      | 21.1            | 24.1            |
> > | MaskCLIP++ + LP   | 21.3            | 26.5            |
> > | Mask-Adapter + LP | 20.4            | 22.3            |
> > | **SEAL (Ours)** | **22.4** | **28.8** |
> >
> > **Table c. DSEC11-*Ins* results under point- and box-prompt settings**
> >
> > | Method                            | Point AP        | Box AP          |
> > | --------------------------------- | --------------- | --------------- |
> > | OVSAM + LP     | 9.2            | 12.1           |
> > | CLIP + LP      | 8.4            | 10.9            |
> > | OVSeg + LP        | 8.9            | 11.4           |
> > | MaskCLIP + LP     | 7.3            | 8.2            |
> > | OpenSeg + LP      | 11.3            | 13.5            |
> > | MaskCLIP++ + LP   | **11.8**            | 14.5            |
> > | Mask-Adapter + LP | 10.8            | 12.9            |
> > | **SEAL (Ours)** | 11.7 | **14.8** |
> >
> > **Table d. DSEC19-*Ins* results under point- and box-prompt settings**
> >
> > Tab. b-d show that our SEAL consistently outperforms the fine-tuned RGB models with the only exception being the point-prompt setting on the DSEC19 benchmark. The performance gap between SEAL and the RGB models narrows as the class configuration becomes more fine-grained. We provide concrete discussion toward this issue in the Sec. B.2 of the Supplementary material (Lines 1331-1340).
> >
> > However, it is important to note that **RGB models lack the practicality in the events-based perception** due to their prolonged inference time (*cf.* Tab. 1, Tab. 12). In contrast, our SEAL provides real-time inference speed with lightweight architecture, enabling the efficient exploration of events with high temporal resolution.
> >
> > ---
> >
> > > `Q4`: "Tab. 1 should be separated by benchmark for better presentation."
> >
> > **A**: We appreciate to this feedback and agree with the author. We will revise the presentation of the Table according to the reviewer's feedback in the camera ready version.
> >
> > ---

---

> > > ### Author Response · Authors · 2025-11-22
> > > **Response to Reviewer 5RxY (3)**
> > >
> > > > `Q5`: "This work focuses on driving scene. However, driving scene has low semantic diversity."
> > >
> > > **A**: This is an another important point. We would first like to discuss the rationale behind focusing on driving scenes in this study: **Why driving scene?**
> > >
> > >
> > > * **Driving scenes provide favorable conditions for using event sensors.**: Driving scenes offer optimal conditions for deploying event sensors due to the combination of fast vehicle ego-motion, dynamic interactions with surrounding traffic participants, and frequent illumination changes such as shadows, flickering lights, and nighttime glare. Conventional RGB cameras usually suffer from these challenging conditions due to the motion blur, saturation, or loss of detail in low-light regions. In contrast, Event sensors are specifically designed to handle these situations, providing microsecond-level temporal resolution, high dynamic range, and asynchronous responses that capture meaningful scene changes even under extreme lighting or motion. Hence, most of the literatures in *Events Perception/Segmentation* [d, e, f, g] focus on driving scene such as DSEC [h], DDD17 [i] and 1 Mpx. Following this research trend, we also primarily focus on driving scenes such as DSEC and DDD17.
> > > * **Scarcity of the Events Dataset with Pixel-wise Semantics.**: Most of the available events datasets equipped with pixel-wise semantics or object-level labels are focused on driving scene due to the first reason. Hence, we curate the four benchmarks by labeling additional instance-level masks and classes on top of existing benchmarks with driving scenes.
> > >
> > > Still, the reviewer’s concern regarding the limited semantic diversity in driving scenes is an important and valuable point. We mitigate this issue by **extracting as much semantic diversity as the data allows** through our MHSG module. Specifically:
> > > * **Multiple levels of granularity from MHSG**: Our MHSG module divides the images into three different levels of granularity (*semantic-level*, *instance-level*, and *part-level*), thereby enabling the extraction of event groups that span a diverse range of semantic categories. For example, given an image containing cars, MHSG groups the events into 'vehicles' at the *semantic level*, 'cars' at the *instance level*, and finer components such as 'car wheels' and 'car windows' at the *part level*. Kindly refer to Fig. 11 and Fig. 12 for the qualitative visualizations of our MHSG.
> > > * **Semantic Augmentation with MLLM**: MHSG further augments the semantic diversity by leveraging the pretrained MLLM [j]. Specifically, we generate the rich captions for every mask proposals from MHSG through pretrained MLLM [j], where each caption contains rich contextual information such as *states*, *functionality* and *actions*. Fig. 11 and Fig. 12 demonstrate that our MHSG generates semantically rich captions, even containing classes such as *hair*, *shoes*, *jackets*, and *trash bins*, which are not closely related to driving scenes.
> > >
> > > Lines 1433-1438 in Sec. C further discusses the semantic diversity of the dataset and the semantic generalizability of the model. As our MHSG module doesn't require any human labors and solely rely on pretrained 2D VLM, kindly note that it can be easily extended to another events dataset which will potentially improve the semantic diversity of the training priors.
> > >
> > > ---

---

> > > > ### Author Response · Authors · 2025-11-22
> > > > **Response to Reviewer 5RxY (4)**
> > > >
> > > > ### References
> > > >
> > > > [a] *Henri Rebecq, René Ranftl, Vladlen Koltun, and Davide Scaramuzza.* High speed and high dynamic range video with an event camera.
> > > >
> > > > [b] *Lingdong Kong, Youquan Liu, Lai Xing Ng, Benoit R Cottereau, and Wei Tsang Ooi.* Openess: Event-based semantic scene understanding with open vocabularies.
> > > >
> > > > [c] *Zhiwen Chen, Zhiyu Zhu, Yifan Zhang, Junhui Hou, Guangming Shi, and Jinjian Wu.* Segment any event streams via weighted adaptation of pivotal tokens.
> > > >
> > > > [d] *Zhaoning Sun, Nico Messikommer, Daniel Gehrig, and Davide Scaramuzza.* Ess: Learning event-based semantic segmentation from still images.
> > > >
> > > > [e] *Inigo Alonso and Ana C Murillo.* Ev-segnet: Semantic segmentation for event-based cameras.
> > > >
> > > > [f] *Lin Wang, Yujeong Chae, Sung-Hoon Yoon, Tae-Kyun Kim, and Kuk-Jin Yoon.* Evdistill: Asynchronous events to end-task learning via bidirectional reconstruction-guided cross-modal knowledge distillation.
> > > >
> > > > [g] *Lin Wang, Yujeong Chae, and Kuk-Jin Yoon.* Dual transfer learning for event-based end-task prediction via pluggable event to image translation.
> > > >
> > > > [h] *Mathias Gehrig, Willem Aarents, Daniel Gehrig, and Davide Scaramuzza.* Dsec: A stereo event camera dataset for driving scenarios
> > > >
> > > > [i] *Jonathan Binas, Daniel Neil, Shih-Chii Liu, and Tobi Delbruck.* Ddd17: End-to-end davis driving dataset.
> > > >
> > > > [j] *Hugo Touvron, Thibaut Lavril, Gautier Izacard, Xavier Martinet, Marie-Anne Lachaux, Timothée Lacroix, Baptiste Rozière, Naman Goyal, Eric Hambro, Faisal Azhar, et al.* Llama: Open and efficient foundation language models.
> > > >
> > > > [k] *Yuqian Yuan, Wentong Li, Jian Liu, Dongqi Tang, Xinjie Luo, Chi Qin, Lei Zhang, and Jianke Zhu.* Osprey: Pixel understanding with visual instruction tuning.

---

> ### Comment · Reviewer_5RxY · 2025-11-27
>
> Thank you for the authors’ responses and for their commitment to contributing datasets and evaluation protocols to the open-source community. I believe these efforts will greatly benefit other researchers. However, I still have some concerns regarding the fact that the model is trained and evaluated solely on autonomous-driving scenarios, which may be insufficient for supporting open-vocabulary segmentation across such a broad and general domain.
>
> Furthermore, presenting the results for all four hierarchy levels separately would more clearly demonstrate the model’s open-vocabulary segmentation capability at each level. These results were not included in the current response. Should the authors provide them, I would be willing to reconsider and potentially increase my score.

---

> ### Author Response · Authors · 2025-11-27
> **Sincerely appreciate your engagement and valuable feedback.**
>
> We sincerely appreciate your valuable feedback. We are currently preparing the additional results that we believe will address your concerns. We will follow up as soon as these results are available.
>
> Thank you again for your active engagement throughout the rebuttal.
>
> Best regards,
>
> The Authors.

---

> ### Author Response · Authors · 2025-12-03
> **Response to Reviewer 5RxY (5)**
>
> > `Q6`: "Tab. 1 should be separated to the three individual tables for better readability."
>
> Thanks for your valuable feedback. We have already addressed this point in `Q4` above. We will separate the Tab. 1 into 3 individual tables to better indicate the hierarchical experimental settings in the camera ready version.
>
> ---
>
> > `Q7`: "Can you provide the open-vocabulary segmentation results in the diverse domain?"
>
> The reviewer raised a concern that open-vocabulary capability of SEAL is demonstrated only in driving scenes. Before addressing this point, we would like to emphasize the inherent challenge of evaluating OV-EIS *across diverse scenes*:
> * **No existing benchmarks to evaluate events instance segmentation:** As we mentioned in the Line 83 of the main paper, there is no existing benchmarks with multiple semantic labels to evaluate OV-EIS. Only a few benchmarks offer event data with pixel-level annotations for evaluating event-based *semantic* segmentation. **However, all of these benchmarks [d, e] are restricted to driving scenes** and consequently, prior works on event-based segmentation [b, c, d, e, f, g] have evaluated their performance exclusively in those driving scenes. We follow the evaluation settings of prior works by focusing on driving scenes while our proposed four benchmarks additionally extend the pixelwise semantic labels to *instance*-level and *part*-level annotations, enabling a comprehensive assessment of event-based *instance* segmentation.
>
> In conclusion, we generally agree with the reviewer’s concern. Nevertheless, this limitation stems from the lack of diverse event-based benchmarks with semantic labels in the research community.
>
> To address the reviewer's concern, we conducted additional experiment which is outlined below:
>
> * **Training**: We first collect additional event–image pairs from existing datasets that cover a diverse range of indoor and outdoor scenes. Specifically, we use RGBE-SEG dataset proposed by [c] where it provides 65,957 image-event training pairs across a diverse range of scene domains. Those image-event pairs are processed by our MHSG module to produce the rich open-vocabulary training guidance. Kindly note that our MHSG module relies solely on pretrained 2D VLMs and MLLMs without requiring any human labor. Therefore, it can be easily extended to additional image–event pairs for training which we discuss in Sec. C (Lines 1435-1438). Finally, our SEAL is trained by MHSG guidance derived from existing driving scenes [d, e] and additional RGBE-SEG dataset [c].
> * **Evaluation**: RGBE-SEG does not provide semantic annotations, which makes quantitative evaluation of OV-EIS challenging. While it is possible to manually design the new OV-EIS benchmark on top of image-event pairs from RGBE-Seg, this is not feasible within the rebuttal period due to time constraints. Hence, we provide *qualitative results* of our SEAL on the several RGBE-SEG testing samples  using free-form language queries.
> * **We add Fig. 15 into the main paper to present the qualitative OV-EIS results of our SEAL across diverse indoor and outdoor scenes from RGBE-SEG dataset.** Given the free-form of langauge queries, our SEAL successfully produces the corresponding event-instance masks across diverse scene domains. In the camera-ready version, we will include a detailed discussion and explanation of this experiment in the Supplementary Material.
>
> ---

---

### Official Review · Reviewer_55Qv · 2025-10-31

**Soundness:** 3
**Presentation:** 3
**Contribution:** 3
**Rating:** 4
**Confidence:** 4

**Summary:**

This paper introduce a framework to segment events using language.
It contains Multimodal Hierarchical Semantic Guidance and Multimodal Fusion Network.
And further propose four benchmarks to evaluate the results.

**Strengths:**

* This paper introduce a "Segment Any Events" framework, which can  generate open-world semantic predictions for event masks.
* This paper  attempt to address OV-EIS that supports free-form of language queries.
* This paper propose four benchmarks for evaluation.

**Weaknesses:**

* The architecture seems rather complex. Could the authors provide a clearer motivation for the inclusion of these modules? Are all of these modules necessary? Is there a simpler approach that could achieve the same results? Besides, it appears that the method is transferring concepts from open-vocabulary image segmentation techniques, such as OpenSeg, MaskCLIP, MaskCLIP++, and OVSeg to the event modality. Could the authors clarify this adaptation and its justification?

* The paper claims to handle open-vocabulary, but is there a benchmark to demonstrate this capability? For example, can the model generalize to unseen classes, or handle user-defined text queries effectively?

* MHSG uses SAM masks and CLIP features on images as supervision signals for the event modality, but there may have significant modality differences between events and images. How accurately can image masks correspond to event regions in this context?

* The part-level experiments seems conducted solely on the DSEC-Part dataset, which contains very few categories, with a small sample size and severe class imbalance. The model’s performance at a finer granularity (e.g., material, action, state) has not been evaluated.

* Additionally, the paper does not analyze conflicts or consistency between different granularities, for example, how the model handles the situation when a wheel is simultaneously recognized as both a “car” and a “wheel”.

I would be happy to revise my score if the author addresses these points.

**Questions:**

Please refer to the weakness

---

> ### Author Response · Authors · 2025-11-22
> **Response to Reviewer 55Qv (1)**
>
> ## Response for Reviewer `55Qv`
>
> We sincerly thank reviewer `55Qv` for the insightful feedback and questions. We have truly enjoyed addressing your question, as it touches on an interesting and important aspect of our work.
>
> ---
>
> > `Q1`: "what is the motivation of each component in the SEAL and baselines?"
>
> **A**: Thanks for giving us the opportunity to clarify the motivation of our SEAL's architecture and the configuration of the baselines.
>
> In this study, **1)** we first propose four types of simple baselines to achieve the OV-EIS (*cf*. Fig. 2). **2)** Then, we discuss the limitations of our proposed naive baselines (Lines 191-198) which motivate us to build more advanced framework, SEAL. **3)** We carefully design every components of the SEAL with **clear motivation** to overcome the limitations of the proposed baselines and **to effectively achieve OV-EIS across multiple levels of granularity.** Extensive ablations further support the effectiveness of each component. We provide detailed explanations to facilitate the reviewer's understanding below.
>
> **Baselines:** Since there is no prior work tackling OV-EIS, we first propose our own baselines by proposing **the most straightforward and simple strategies to achieve OV-EIS**. Specifically, all the baselines follow the two-stage framework [a]: 1) Mask segmentor → 2) Mask classifier. **We implement each stage by directly leveraging existing RGB or event models**, where we define two types of modules that can be selected for each stage, indicated in *red* and *blue* in Fig. 2.
>
> * **RGB-based models + E2VID (*Red*)**: This module directly applies the existing RGB-based open-vocabulary models into events modality. Our motivation is to evaluate the zero-shot transfer of RGB-based models (such as OpenSeg, OVSeg, MaskCLIP++ and etc) into events domain as *generalist*, given that they are pretrained on large-scale image datasets with rich human annotations. Since open-vocabulary models learned by large scale datasets with accurate text annotations are readily available in the image domain (*cf.* Tab. 7), directly applying them into events domain is a reasonable choice. To enable the direct inference of RGB models on events modality, we reconstruct the gray-scale images from events through E2VID [b] and use them for the inference of RGB models. Kindly note that demonstrating direct transfer of the RGB models into events modality has been widely adopted as baselines by the previous literatures [c, d] due to its simplicty.
> * **Events-adapted models (*Blue*)**: This module represents the domain adaptation approach where the neural network is adapted to events domain through feature distillation via events-image pairs. Specifically, the rich representations obtained from RGB generalist are distilled into events network in an annotation-free manner using only event-image pairs. This **annotation-free** feature distillation approach has been widely used in the events-based perception studies [c, d, e] since the human annotations are highly scarce in events domain. We use EventSAM [c] when the events model functions as mask segmentor whereas the OpenESS [d] framework is applied when the events model is used as a mask classifier.
>
> Overall, we define four types of baselines by exploring different combinations of these two modules within our two-stage framework, which can be summarized as follows:
> | Types | Mask Segmentor | Mask Classifier|
> | --------------------------------- | --------------- | --------------- |
> | Fig. 2a | RGB (*Red*) | RGB (*Red*)
> | Fig. 2b-2c | Events (*Blue*) | RGB (*Red*)
> | Fig. 2d | Events (*Blue*) | Events (*Blue*)
>
> **Table. a. Summary of the four types of proposed baselines.**
>
> Kindly refer fo Sec. A.3 for more details about our baseline.
>
> **Limitations of baselines**: We further discuss the limitations of the aforementioned baselines in Lines 191-198. **1)** RGB models still suffer from a huge domain gap between events and images despite the events are reconstructed to a gray-scale frame. This is because E2VID model often introduces severe artifacts in reconstructed frames, particularly in textureless regions or under low event rates [d]. **2)** All baselines require two distinct backbones, one for each stage of two-stage pipeline. It degrades the parameter and inference efficiency which is the crucial aspects in events-based perception tasks (*cf*. Lines 418-423). These limitations motivate us to build more advanced framework through a unified and efficient framework, called SEAL.

---

> ### Author Response · Authors · 2025-11-22
> **Response to Reviewer 55Qv (2)**
>
> (continue from `Q1`...)
>
> **SEAL**: In contrast to proposed baselines, our SEAL adopts the single-backbone architecture to preserve the parameter and inference efficiency. Furthermore, we attach **three main modules** on top of EventSAM's backbone to obtain high quality of language-aligned events mask features across multiple levels of granularity.
> *  **Backbone Feature Enhancer (Lines 261-286)**: The motivation of this module is to enhance the events backbone features $\mathbf{I}^{evt}$ by explicitly fusing the language prior (*semantic*) into events features through cross-attention layer. This early fusion technique reduces the domain gap between events and language, enabling better alignment between events and text. 4th-5th rows of Tab. 5 shows that backbone feature enhancer consistently leads to performance improvement across all benchmarks.
> *  **Spatial Encoding (Lines 287-306)**: Given the language-fused events feature $\hat{\mathbf{I}}^{evt} \in \mathbb{R}^{H/32 \times W/32}$, we pool the mask features corrdspond to events mask $M^{evt} \in \mathbb{R}^{H \times W}$ via Roi-Align layer. However, as resolution of features map ($H/32 \times W/32$) are much smaller than the resolution of events mask ($H \times W$), solely relying on the pooling method to obtain mask feature yields two main issues: ***1) Dead masks*** and ***2) Semantic conflicts***. These two issues are concretely explained in Lines 275-286, Lines 1185-1216 and Fig. 7. Due to the severe downscaling of mask resolution into features map resolution, the masks with small sizes often disappear (Fig. 7a) which is defined as *Dead masks* issue. Furthermore, multiple masks with different semantics are often projected onto the same feature patches (Fig. 7b) which is named as *Semantic conflict*. As reviewer mentioned, we first try the most simple strategy to resolve these issues: Upscale the resolution of features map into mask resolution. However, applying Roi-Align layer to high-resolution features map incurs huge computational overhead, particularly when the large number of masks are given as we discuss in Fig. 7c and Lines 1202-1212.
>
>     These explorations motivates us to add **Spatial Encoding** module which compensates for the spatial priors derived from mask decoder in pooled events mask feature. This module has three advantages: 1) By injecting the distinct spatial characteristic of each mask into the mask features, mask event representations becomes more discriminative, effectively solving the *semantic conflict*. 2) *Dead masks* issue is also resolved since every outputted masks have their own mask tokens regardless of their sizes. 3) No need to upscale the resolution of features map, avoiding the potential computational overhead from Roi-Align layer (*cf*. Fig. 7c). Kindly note that combining *Backbone Feature Enhancer* and *Spatial Encoding* enables mutual compensation between *semantic priors* from language fusion and *spatial priors* from mask decoder. Fig. 4 highlights the impact of the Spatial Encoding module where adding Spatial Encoding eliminates the dead mask issue and produces a more discriminative event-representation space (*cf*. Fig. 4b). 1st-2nd rows of Tab. 5 further show that Spatial Encoding (SE) leads to the performance improvement across all the benchmarks.
> *  **Mask Feature Enhancer (Lines 300-306)**: This module further enhances the incorporation of *semantic* and *spatial* priors from previous modules through cross-attention layer with masked attention. Combining the Mask Feature Enhancer (MFE) further yields performance improvement across all the benchmarks as shown in 2nd and 5th rows of Tab. 5.
>
> **Summary**: We design three main moduels on top of EventSAM's backbone to obtain discriminative events mask features by encoding both *semantic priors* and *spatial priors*. **1) Backbone Feature Enhancer** explicitly fuses *language priors* into events features. **2) Spatial Encoding** injects the *spatial priors* derived from SAM's mask decoder into the pooled mask features.  **3) Mask Feature Enhancer** further fuses the *semantic priors* and *spatial priors* together through cross-attention layer and masked-attention. Extensive abltaions (*cf*. Tab. 4-5 and Fig. 4) support the effectiveness of each component.
>
> If the reviewer has any further questions regarding the motivation of the components in SEAL's architecture, we would be happy to continue the conversation and provide further explanation.
>
> ---

---

> ### Author Response · Authors · 2025-11-22
> **Response to Reviewer 55Qv (3)**
>
> > `Q2`: "1) How can we evaluate open-vocabulary capability of the events model? 2) Is there a benchmark to demonstrate the open-vocabulary capability?"
>
> **A**: Reviewer's question highlights a central challenge in open-vocabulary events understanding, and we appreciate the opportunity to discuss this important aspect.
>
> **How can we evaluate open-vocabulary capability?**: To evaluate the open-vocabulary capability of the model, we adopt widely-used open-vocabulary evaluation setup:
> 1. **Annotation-free Training**: Training the open-vocabulary model *without* using the human-annotated (ground-truth) class labels.
> 2. **Open-vocabulary Evaluation**: Evaluating the model's performance with using the ground-truth class labels.
>
> Since the model didn't access to any ground-truth labels during the training, those classes are regarded as *unseen* classes and used in the evaluation. This setup has been widely adopted across diverse modlities, including-but not limited to-open-vocabulary 3D scene understanding [f, g, h, i] and open-vocabulary events understanding [d]. As we mention in the Lines 200-202, our SEAL is trained by only using the event-image pairs without accessing to the human-annoatated labels which is the common setup in events-based perception [c, d, e].
>
> **Is there any benchmark to evaluate OV-EIS?**: In contrast to image domain where the large-scale benchmarks with over 100 annotated classes are readily available, **events benchmark with human annotations are highly scarce (*cf*. Tab. 7).** Furthermore, there is no existing benchmark to evaluate the OV-EIS with multiple semantics as we mention in Lines 83-84. Hence, we define the '*absent of the benchmarks*' as one of the main challenges to achieve OV-EIS.
>
> To address this challenge, we curate four new benchmarks to evaluate the OV-EIS (*cf*. Sec. A.1). It should be noted that our benchmarks are carefully designed to cover diverse evaluation settings to throughly evaluate the open-vocabulary capability: **1)** ***Label granularity***: The model is evalulated from coarse-level class configurations to fine-grained class configurations where the number of classes increases from 5 to 14. **2)** ***Semantic granularity***: The model is evaluated on diverse levels of semantic granularity from instance-level to part-level segmentation. Kindly refer to Fig. 5b and Lines 939-949 for more detailed explanation about our experimental settings.
>
> To the best of our knowledge, **these diverse semantic evaluation settings have not been considered in the events-based perception community.** Given that the model is trained in an annotation-free setting, we believe our four benchmarks effectively support the evaluation of open-vocabulary capabilities across diverse scenarios.
>
> ---
>
> > `Q3`: "Can model handle user-defined text queries effectively? Can model understand fine-grained text queries such as material, action or state?"
>
> **A**: Due to the absence of OV-EIS benchmark with diverse forms of langauges which include material, action or state, we are not able to provide the quantitative evaluations in those conditions. However, we provide **qualitative results** to show the responsiveness of our SEAL on free-form of language in Fig. 13.
>
> Fig. 13 shows that our SEAL outputs the related masks based on the user-defined text queries. For example,
> * **Action & State**: SEAL outputs the mask of walking human when the the text query ''*walking*'' is given. In contrast, the model extracts the mask of human riding the bicycle when queried by ''*ride a bicycle*'', showing the responsiveness of the model based on the action or state.
> * **Functionality**: When the text query ''*living place*'' or ''*Throwing away the garbage.*'' is given to the model, SEAL produces masks corresponding to houses and trash bins, respectively. This demonstrates that SEAL can respond to queries with functional semantics.
>
> Reviewer also mention the *material*. However, events do not contain appearance cues such as *color* or *material*, as they only encode brightness changes rather than absolute visual attributes. Hence, we do not address the text queries with appearance features in this study.
>
> ---

---

> ### Author Response · Authors · 2025-11-22
> **Response to Reviewer 55Qv (4)**
>
> > `Q4`: "How accurately can image masks correspond to event regions?"
>
> **A**: We follow the previous studies [c, d, e] to accurately align the events with image.
>
> Given the image $I^{img}_t \in \mathbb{R}^{H \times W \times 3}$ at time step $t$, we first crop the events from $t - \Delta t$ to $t$, forming a short temporal window. The cropped events $\epsilon \in \mathbb{R}^{N \times 4}$ are converted to **frame-like representation** $I^{evt}_t \in \mathbb{R}^{H \times W \times B}$ by following the voxelization process [e] described in Lines 1048-1058 of our paper, where $N$ is the number of events generated from time step  $t - \Delta t$ to $t$ and $B$ denotes the temporal bins used in the voxelization process.
>
> Obtained events representation $I^{evt}_t \in \mathbb{R}^{H \times W \times B}$ is aligned pixel-by pixel with the image $I^{img}_t \in \mathbb{R}^{H \times W \times 3}$, enabling to use image mask to guide the events representation. Kindly note that voxelizing the events to frame-like representation is a common practice in event-based vision, as it enables direct integration of CNNs, transformers, and other image-centric architectures.
>
> Reviewer can further check the visualization of events-image pairs in Fig. 11, where it demonstrates the visualization of MHSG module obtained from events-image pairs.
>
> ---
>
> > `Q5`: "DSEC-Part dataset contains few categories with class imbalance and small sample size."
>
> **A**: We agree with the reviewer’s observation. we believe that the limited part-level semantic diversity, class imbalance, and small sample size are inherent to the **driving-domain outdoor scenes** on which our study is based. As Reviewer `5RxY` noted, driving scenes have less diverse semantic categories compared to the other domain such as indoor scene, and often exhibit highly repetitive structures. We provide additional clarifications below to support the reviewer’s considerations.
>
> **Why focus on driving scene?**: Driving scenes provide favorable conditions for using event sensors due to the combination of fast vehicle ego-motion, dynamic interactions with surrounding traffic participants, and frequent illumination changes such as shadows, flickering lights, and nighttime glare. Conventional RGB cameras usually suffer from these challenging conditions due to the motion blur, saturation, or loss of detail in low-light regions. In contrast, Event sensors are specifically designed to handle these situations, providing microsecond-level temporal resolution, high dynamic range, and asynchronous responses that capture meaningful scene changes even under extreme lighting or motion. Hence, most of the literatures in *Event-based Perception* [c, d, e] focus on driving scene such as DSEC [j], DDD17 [k] and 1 Mpx. Following this research trend, we also primarily focus on driving scenes such as DSEC and DDD17 which potentially leads to the lack of diverse semantic categories of our DSEC-*Part* benchmark.
>
> **Why only evaluate on DSEC-*Part* benchmark?**: The primary objective of our study is to develop an OV-EIS model capable of understanding semantics at multiple levels of granularity. However, **no existing benchmark provides part-level semantic annotations for event streams.** To address this gap, we construct DSEC-*Part* which evaluates the part-level understanding capability of our SEAL. We choose "*car*" and "*building*" as base categories since they have multiple distinguishable components and define the 5 part-level classes for "*car*" and 4 part-level classes for "*building*". Kindly refer to Sec. A.1 and Tab. 6 for more detailed explanation.
>
> Overall, due to the inherent characteristics of driving scenes and the scarcity of event-based datasets, we acknowledge that our DSEC-*Part* benchmark cannot be entirely free from the limitations that reviewer mentioned.
>
> Still, we would like to highlight the value of our study. To the best of our knowledge, our study is the **first attempt** to address the part-level understanding in the events-domain. Considering the severe scarcity of the events-domain benchmark, we believe our new part-level benchmark carries pioneering value to extend the events understanding into fine-grained granularity.
>
> ---

---

> ### Author Response · Authors · 2025-11-22
> **Response to Reviewer 55Qv (5)**
>
> > `Q6`: "Conflicts between different granularities."
>
> **A**: We are grateful that you raised this important point. We define the aformentioned problem as *semantic conflict* which is explained in the Lines 281-286, Lines 1199-1216 and Fig. 7b. Since *instance*-level and *part*-level masks can be overlapped together, they can be pooled from the same feature patches as it is illustrated in Fig. 7b. To mitigate this issue, we introduce the *Spatial Encoing* module in Lines 287-299 to enhance the events mask representation with *spatial prior*. Specifically, mask tokens derived from the mask decoder, which encode rich spatial and geometric priors of the mask, are further combined to events mask features.
>
> Even though *instance*-level masks and *part*-level masks are spatially overlapped, variations in their size, shape, and spatial extent introduce distinct geometric characteristics between them. Hence, adding the geometric priors through mask tokens further differentiates the feature spaces of *instance*-level masks and *part*-level masks, mitigating the conflict between them. To clarify this aspect, we provide an additional ablation on the Spatial Encoding in  DSEC-*Part* benchmark below.
>
> **Experimental setting**: To investigate the effectiveness of Spatial Encoding (SE) in cross-granularity conflict, we first construct two model settings: 1) Our original SEAL and 2) SEAL trained without SE. For each model, we estimate the average performance drop of part-level classes between two experiments: i) We evaluate part-level segmentation performance where only the part-level classes are given to the model which is identical with the setting of Tab. 2. ii) Two instance-level classes which are "*car*"" and "*building*" are additionally added to the class candidates to induce the semantic conflict and part-level segmentation performance is conducted. The larger performance drop in part-level classes implies a stronger vulnerability to semantic conflict of the model.
>
> **Experimental Result**: Tab. b shows that SEAL trained with SE exhibits a 2.9x smaller performance drop compared to SEAL trained without SE, demonstrating its effectiveness in addressing cross-granularity conflict.
>
> | Method                            | AP degradation $\downarrow$        |
> | --------------------------------- |--------------- |
> | without SE    | -3.5            |
> | with SE | **-1.2** |
>
> **Table b. Ablation on the effectiveness of SE in cross-granularity conflict**
>
> Even though our Spatial Encoding shows its effectiveness to address the cross-granularity conflict, the cases where SEAL misclassifies '*wheel*' as '*car*' still happens due to their semantic similarities if the two classes are jointly provided. We will provide  additional discussion about this on Sec. C in the camera ready version.
>
> ---
>
> ### References
>
> [a] *Xu, Mengde, et al.* A simple baseline for open-vocabulary semantic segmentation with pre-trained vision-language model.
>
> [b] *Henri Rebecq, René Ranftl, Vladlen Koltun, and Davide Scaramuzza.* High speed and high dynamic range video with an event camera.
>
> [c] *Zhiwen Chen, Zhiyu Zhu, Yifan Zhang, Junhui Hou, Guangming Shi, and Jinjian Wu.* Segment any event streams via weighted adaptation of pivotal tokens.
>
> [d] *Lingdong Kong, Youquan Liu, Lai Xing Ng, Benoit R Cottereau, and Wei Tsang Ooi.* Openess: Event-based semantic scene understanding with open vocabularies.
>
> [e] *Zhaoning Sun, Nico Messikommer, Daniel Gehrig, and Davide Scaramuzza.* Ess: Learning event-based semantic segmentation from still images.
>
> [f] *Songyou Peng, Kyle Genova, Chiyu Jiang, Andrea Tagliasacchi, Marc Pollefeys, Thomas Funkhouser, et al.* Openscene: 3d scene understanding with open vocabularies.
>
> [g] *Zhening Huang, Xiaoyang Wu, Xi Chen, Hengshuang Zhao, Lei Zhu, and Joan Lasenby.* Openins3d: Snap and lookup for 3d open-vocabulary instance segmentation.
>
> [h] *Phuc Nguyen, Tuan Duc Ngo, Evangelos Kalogerakis, Chuang Gan, Anh Tran, Cuong Pham, and Khoi Nguyen.* Open3dis: Open-vocabulary 3d instance segmentation with 2d mask guidance.
>
> [i] *Ayc¸a Takmaz, Elisabetta Fedele, Robert W Sumner, Marc Pollefeys, Federico Tombari, and Francis Engelmann.* Openmask3d: Open-vocabulary 3d instance segmentation.
>
> [j] *Mathias Gehrig, Willem Aarents, Daniel Gehrig, and Davide Scaramuzza.* Dsec: A stereo event camera dataset for driving scenarios
>
> [k] *Jonathan Binas, Daniel Neil, Shih-Chii Liu, and Tobi Delbruck.* Ddd17: End-to-end davis driving dataset.

---

> ### Author Response · Authors · 2025-11-23
> **Supplementary Responses for Reviewer 55Qv (1)**
>
> ## Supplementary Responses for Reviewer `55Qv`
>
> In this section, we provide additional **technical details** and **clarifications** that were not included in the main responses, to further support the reviewer’s understanding. If the reviewer has no additional questions after reading the main responses above, this section may be skipped.
>
> ---
>
> ### **Supplementary explanations for** `Q1`
>
> In the main response of `Q1`, we describe each component of our proposed baselines and SEAL, along with the motivation behind each module. Here, we further provide technical details for our **baselines**: *How are RGB-based models such as MaskCLIP++, Mask-Adapter, and OpenSeg adapted to the event modality?*
>
> First of all, reviewer mentioned that
> > It appears that the method is transferring concepts from open-vocabulary image segmentation techniques.
>
> We would like to clarify this point. Rather than *transferring the concepts* of the image-domain techniques, **we directly apply pretrained open-vocabulary image segmentation models to the event modality as baselines.** As we mentioned in the main response, our focus on designing the baseline is **1)** to explore the most straightforward approaches to enable the OV-EIS and **2)** analyze their limitations which further motivates our own framework, SEAL.
>
> Below, we further clarify the *technical details* on how RGB models are adapted to the event modality as baselines. **Kindly note that specific technical details for baselines are all provided in Sec. A.3.** For convenience, we offer a brief explanation here as well.
>
> * **AR-CDG (Fig. 2a)**: As described in Tab. a of the main response, this type adopts RGB model for both mask segmentor and mask classifier. Specifically, OVSAM is adopted as a baseline for this type. OVSAM is *Semantic-aware Segment Any Image* model which provides unifed framework for open-vocabulary mask segmentor and mask classifier. As OVSAM is image model, it can't be directly transferred to events modality. Hence, 1) we first convert the events to gray-scale image through E2VID which is generative model. 2) Obtained gray-scale image is fed to OVSAM for OV-EIS. Since OVSAM supports open-vocabulary interactive segmentation, we directly apply pretrained model into E2VID-reconstructed image without any modification.
> * **Hybrid (Fig. 2b-2c)**: As described in Tab. a of the main response, this type adopts events module as mask segmentor and RGB module as mask classifier. Specifically, voxelized events (*cf*. main response of `Q4`) are injected to EventSAM which is *events mask segmentor*, obtaining the class-agnostic events masks. These events masks are classified by using the RGB-based open-vocabulary model such as CLIP, OVSeg, MaskCLIP, MaskCLIP++, Openseg and Mask-Adapter. Similar with AR-CDG baseline, events are first reconstructed to gray-scale image before the mask classification for the direct transfer of RGB models. We define two types of baseline based on how they classify the masks. ***1) Image Crop Baseline (Fig. 2b)***: CLIP and OVSeg fall in this type. Given the reconstructed image from events, they crop the image patches of the resulting masks and subsequently classify them.   **2) Feature Crop Baseline (Fib. 2c)**: MaskCLIP, MaskCLIP++, Openseg and Mask-Adapter fall in this type. They ingest the entire reconstructed frame and output the open-vocabulary mask features, given by the class-agnostic masks generated from EventSAM. We further provide specific technical details on how the feature-crop baseline is implemented across the different methods below.
>     * OpenSeg, MaskCLIP: They output per-pixel CLIP features for reconstructed frame. We then conduct masked average pooling to the features map, obtaining the CLIP-aligned mask features which can be classified by CLIP text features.
>     * MaskCLIP++, Mask-Adapter: These methods attach additional mask classifier module on top of pretrained CLIP. Mask classifier is then trained to output the open-vocabulary mask features from the CLIP. In our study, given the entire image of reconstructed frame and class-agnostic masks, the mask classifier of these methods directly outputs CLIP-aligned mask features by referring to the CLIP visual features. Kindly refer to the original paper for further details.
>
> Although simple and effective, they incurs prolonged inference time with large parameter sizes (*cf*. Tab. 1). Furthermore, they suffer from artifacts or noises generated from E2VID when the events are converted to image domain. In contrast, our SEAL enables **1)** real-time OV-EIS with lightweight architecture and **2)** achieves the best performance across all the OV-EIS benchmarks with diverse scenarios. **Additionally,** it is free from E2VID-oriented artifacts or noises since our SEAL takes voxelized events as input.
>
> ---

---

> ### Author Response · Authors · 2025-11-23
> **Supplementary Responses for Reviewer 55Qv (2)**
>
> ### **Supplementary explanations for** `Q6`
>
> In the main response of `Q6`, we provided an additional ablation study of the Spatial Encoding module with respect to the cross-granularity conflict problem. **1)** Here, we further provide the detailed *conceptual explanation* about cross-granularity conflict problem: *Why does the cross-granularity conflict occur, and how does the Spatial Encoding module mitigate this issue?* **2)** Furthermore, we provide additional functionality of our SEAL which supports the instance-level segmentation and part-level segmentation at the same time.
>
> ----
>
> * **Why does cross-granularity conflict occurs?**: The key idea of our events representation learning framework with multi-levels of granularity is to pool the event mask features from the event backbone features map and align them with the corresponding open-vocabulary semantics obtained from our MHSG module. To learn rich event representations across both instance-level and part-level granularities, we pool mask features from both granularity levels and supervise them with the corresponding MHSG guidance. However, since part-level and instance-level masks frequently exhibit spatial overlap, identical event feature patches are pooled by both masks and subjected to supervisions from two different semantics at the same time. For example, event feature patches corresponding to ‘*wheel*’ may be simultaneously supervised by the semantics of both ‘*wheel*’ and ‘*car*’, because those patches also fall within the spatial extent of the ‘*car*’ mask. This problem is defined as *semantic conflict* in the paper.
> * **How does the Spatial Encoding module mitigate this issue?**: As we mentioned in the main response of `Q6`, part-level masks and instance-level masks have different geometric characteristic such as sizes, shapes and spatial extent. This motivates us to better separate the event features space of instance- and part-level masks by exploiting the geometric differences inherent to each granularity. Hence, we additionally incorporate the mask tokens produced by the mask decoder, into pooled event mask features. As mask tokens encode rich spatial priors of corresponding masks, it effectively enhances the events representation across different granularities, mitigating the semantic conflict issue.
>
> ----
>
> Now, we introduce additional functionality of our SEAL that supports instance-level segmentation and part-level segmentation at the same time which is described in Fig. 14.
>
> Since our SEAL is built on top of EventSAM which adapts SAM to the event modality, SEAL retains all functionalities provided by the original SAM model. For example, given the single point prompt, SEAL outputs both object level mask and part-level mask at the same time, just like SAM. Specifically, the mask decoder employs three mask tokens, where each token represents a distinct granularity level. These mask tokens produce three different masks of varying granularity in parallel. Our motivation is to exploit this functionality of SAM that supports multi-level segmentation at the same time.
>
> First, we treat the mask token corresponding to the coarsest granularity as the instance-level token, and the mask token corresponding to the finest granularity as the part-level token. Given the single point prompt, we extract two masks from both instance-level token and part-level token. We then provide instance-level categories such as those from the DSEC11-*Instance* benchmark, to the instance-level mask for mask classification. If an object-level label is obtained, we subsequently provide the corresponding part-level categories associated with that object to the part-level mask. For example, if the SEAL classifies the instance-level mask as '*car*', we further provide part-level classes for '*car*' to the part-level mask to further conduct part-level mask classification. **Quantitative results are provided in Fig. 14 with the explanations in Lines 1400-1416.** By leveraging the explicit division of granularity derived from SAM's mask decoder, our SEAL successfully conducts the segmentation in both granularities without conflict.

---

### Official Review · Reviewer_PR6s · 2025-11-01

**Soundness:** 4
**Presentation:** 2
**Contribution:** 3
**Rating:** 8
**Confidence:** 3

**Summary:**

This work introduces a multimodal, hierarchical semantic method  to align event-based segmentation with open-vocabulary language queries. It supports both instance-level and part-level mask generation. The proposed method uses three types of prompt-driven masks generated by SAM for each image (semantic, instance, part). The features are supervised by pooled CLIP embeddings and LLM-generated captions. The multimodal fusion network, with a single backbone, combines language and vision with explicit spatial encoding per mask, addressing previous inefficiencies and semantic conflicts in feature pooling observed in all AR-CDG, Hybrid, and AF-DA baselines.

**Strengths:**

This work is shown to outperform MaskCLIP and other leading open-vocabulary baselines by 3.4 AP on DSEC11-Ins and 3.2 AP on DDD17-Ins. Inference is 5-18x faster with fewer than 1/5th the parameters. This is a good contribution towards practical application.

Secondly, unlike prior methods, this work’s two-stage mask feature enhancement and spatial encoding overcome the "dead mask" issue, where small-event region masks are mapped to zero vectors; UMAP visualisations show tight semantic separation after adding these modules.

**Weaknesses:**

* DSEC19-Ins is a highly fine-grained dataset: on it, the improvement over MaskCLIP narrows to just 0.7 AP. This, it seems to me that, even with annotation-free training, suggests that the distilled representations are less robust when class granularity exceeds the capacity of available MHSG cues.
* I think that the main variant of the method benefits from GT-derived visual prompts for mask proposals; although a supplementary "prompt-free" variant exists, the claim of real-world flexibility is less convincing without broader prompt-agnostic validation.

**Questions:**

* For DSEC-Part, are part-level mask labels created
"by hand" (since event data is too impoverished for mask proposals alone to identify parts)? If so, doesn’t this somewhat undermine the claim of annotation-free scaling?
* Artefacts seem to be mitigated compared to the baselines. Even so, couldn’t reconstructing events to images or mapping event data to traditional vision models still introduce artefacts or domain discrepancy, especially under high-speed or low-event-rate scenarios?

---

> ### Author Response · Authors · 2025-11-22
> **Response for Reviewer PR6s (1)**
>
> We sincerely thank Reviewer `PR6s` for the time dedicated to reviewing our work and for the constructive feedback provided. We are encouraged by your recognition of the strong practicality of our ***SEAL*** and the clear instance-level event representations demonstrated through the UMAP visualization.
>
> ---
>
> > `Q1`: "Distilled representations seem less robust when class granularity exceeds the capacity of available MHSG cues"
>
> **A**: Thank you for highlighting this valuable point. We discuss this aspect in the Sec. B.2 of the Supplementary material (Lines 1331-1340). As you mentioned, DSEC19 [b] has fine-grained class configuration especially for the *vehicle* category which consists of *car*, *truck*, *bus*, *train*, *motorcycle* and *bicycle*. When those fine-grained *vehicle* classes are given as joint text conditions, our SEAL often struggles to distinguish *car* from the other semantics within *vehicle* category. We postulate that this issue primarily stems from semantic imbalance within the training dataset. In driving scenes such as DSEC [c] and DDD17 [d], the *car* class overwhelmingly dominates the vehicle category, while other types such as *truck*, *bus*, or *motorcycle* appear far less frequently. Since our MHSG is derived from the event-image pairs of DSEC and DDD17, it inherents the same class imbalance, causing SEAL to generate false positives for the *car* class. (For example, it sometimes misclassifies *bus* as *car*.)
>
> However, we would like to remark two important contributions of our SEAL below:
> 1. Our SEAL **still outperforms** the existing baselines in DSEC19-*Instance* benchmark with **much faster inference speed** and with **much lighter model architecture**. Real-time inference speed is crucial in events-based perception task [a], as the model must remain synchronized with the high temporal resolution of event streams. Tab. 1 shows that our framework is the only one capable of achieving real-time event instance segmentation.
> 2. Our MHSG is produced in **human-annotation free manner** by only exploiting the pretrained 2D foundation models. Hence, it can be readily scaled by collecting more event–image pairs, which would help mitigate the class imbalance problem by collecting more data that contain less frequent semantic classes.
>
> ---
>
> > `Q2`: "SEAL benefits from GT-derived visual prompts for mask proposals."
>
> **A**: We clarify our experimental settings more cleary below:
> * **Tab. 1 and Tab. 2**: We evaluate the interactive segmentation in Tab. 1 and Tab. 2, where the visual prompts such as point and box are given to the model. To ensure the fair comparisons, all methods including the baselines and our SEAL are provided by prompts sampled from the ground-truth masks. Since all methods are given same quality of GT-sampled visual prompts, SEAL **does not receive any additional advantage** in terms of visual prompt quality compared to the baselines.
> * **Tab. 11**: We evaluate the generic instance segmentation in Tab. 11 where we assume **no prompt is provided by user (prompt-free)**. Specifically, the event-based object detection model [e] is additionally combined as off-the-shelf module to provide box prompts while SEAL++ further integrates [e] by sharing the single backbone. Kindly note that all methods in Tab. 11 do not use visual propmts sampled from the GT masks to simulate the *prompt-free* setting.
>
> ---

---

> ### Author Response · Authors · 2025-11-22
> **Response for Reviewer PR6s (2)**
>
> > `Q3`: "Broader prompt-agnostic valiadatoin is necessary."
>
> **A**: We appreciate to this valuable feedback. As reviewer mention, we provide the evaluation on prompt-free setting in Tab. 11 to support the real-world flexibility. According to reviewer's feedback, we supplement additional experiment on *prompt-free* OV-EIS below:
>
> **Experimental Setting**: Since our SEAL is built on top of EventSAM architecture, it also supports auto-segmentation of original SAM where it generates the multiple masks that cover the whole image without requiring the explicit visual prompts from the user. By exploiting this auto-segmentation capability, we evaluate generic segmentation on our DDD17-*Instance* and DSEC11-*Instance* benchmarks. We adopt AP as metric, following the main paper.
>
> **Experimental Results**: Tab. a and Tab. b show that our SEAL achieves the best performance by outperforming all the baselines. This superior results are consistent with the interactive segmentation results, showing the robust open-vocabulary capability of our SEAL.
>
>
> | Method                            | AP        |
> | --------------------------------- | ---------------
> | OVSAM     | 16.5            |
> | CLIP      | 15.3            |
> | OVSeg         | 15.4        |
> | MaskCLIP     | 14.3        |
> | OpenSeg      | 20.4       |
> | MaskCLIP++   | 19.4  |
> | Mask-Adapter | 18.8 |
> | frame2recon | 18.7 |
> | frame2voxel | 17.9 |
> | frame2spike | 17.4 |
> | **SEAL (Ours)** | **22.1**     |
>
> **Table a. Generic EIS in DDD17 through auto-segmentation**
>
> | Method                            | AP        |
> | --------------------------------- | ---------------
> | OVSAM     | 12.1            |
> | CLIP      | 11.5           |
> | OVSeg         | 11.4       |
> | MaskCLIP     | 11.1        |
> | OpenSeg      | 16.3       |
> | MaskCLIP++   | 16.4  |
> | Mask-Adapter | 15.9 |
> | frame2recon | 15.5 |
> | frame2voxel | 15.3 |
> | frame2spike | 15.6 |
> | **SEAL (Ours)** | **18.9**     |
>
> **Table b. Generic EIS in DSEC11 through auto-segmentation**
>
> ---
>
> > `Q4`: "DSEC-Part are part-level mask labels created by hand. Doesn’t this undermine the claim of annotation-free scaling?"
>
> **A**: The DSEC-*Part* benchmark is annotated **strictly for evaluation, not training**. As we mentioned in Lines 360-362, we manually annotate the labels to ensure the accurate evaluation since no existing benchmark currently provides part-level semantic maps or supports evaluation on part-level event segmentation
>
> However, our MHSG module which provides training guidance to our SEAL is constructed entirely **without any human annotations**. Specifically, it provides rich vision-language priors for events represenation learning across multiple levels of granularity, **by only leveraging the existing 2D foundation models**. Hence, MHSG can be readily extended to other event–image pair datasets without requiring any human labors, enabling scalable expansion of the training priors for SEAL (*cf.* Lines 1435-1438).
>
> ---

---

> ### Author Response · Authors · 2025-11-22
> **Response for Reviewer PR6s (3)**
>
> > `Q5`: "Does the event representation also inherit artifacts when the reconstructed grayscale image from events contains artifacts?"
>
> **A**: We appreciate the reviewer for bringing up this point. We will clarify why using voxelized event representations (Lines 1048-1063) is preferable to using reconstructed images derived from events (Lines 1064-1073).
> * **Free from hallucinated artifacts**: E2VID is a *generative* model that aims to generate grayscale intensities from sparse event data. Hence, E2VID often hallucinates the missing information in textureless regions where few or no events are generated. This hallucination causes implausible artifacts which is not physically meaningful. This issue becomes more pronounced under high-speed motion, where rapid changes produce streak-like artifacts, unstable brightness normalization, and temporal inconsistencies across reconstructed frames. Baselines fall in *AR-CDG* and *Hybrid* categories suffer from this hallucinated noise since they adopt E2VID-reconstructed frames to perceive the events (Lines 191-196). In contrast, voxelized events used in SEAL are free from such hallucinations, as voxelization is a deterministic and non-generative mapping that converts raw events into frame-like tensors. This process preserves the true spatiotemporal structure of the event stream without inventing appearance information, while remaining fully compatible with CNN- or Transformer-based architectures commonly used in computer vision.
>
> However, as reviewer noted, event streams still contain the noise from the sensor in textureless regions which further exacerbates the domain gap with images. It may also suffer from low event rates when there is little motion in the scene. This inherent nature of events makes events-based perception more challenging. In our study:
>
> 1. Our MHSG module mitigates this challenge by exploiting the super-pixel driven approach with diverse levels of granularity which is specifically explained in Lines 1147-1156 of the Supplementary material. Specifically, we group the events into semantically meaningful regions to mitigate the sparsity of events and smooth the noise during the training.
> 2. To further reduce the domain gap between events and vision-language representations, we fuse language-modality priors into events backbone features $\mathbf{I}^{evt}$ through cross-attention layer in *backbone feature enhancer*. Tab. 5 shows that text fusion in the events network enhance the alignment between events and text (4th row vs **5th row**), consistently leading to better performance across all the benchmarks.
>
> ---
>
> ### References
>
> [a] *Mathias Gehrig and Davide Scaramuzza.* Recurrent vision transformers for object detection with event cameras.
>
> [b] *Zhaoning Sun, Nico Messikommer, Daniel Gehrig, and Davide Scaramuzza.* Ess: Learning event-based semantic segmentation from still images.
>
> [c] *Mathias Gehrig, Willem Aarents, Daniel Gehrig, and Davide Scaramuzza.* Dsec: A stereo event camera dataset for driving scenarios
>
> [d] *Jonathan Binas, Daniel Neil, Shih-Chii Liu, and Tobi Delbruck.* Ddd17: End-to-end davis driving dataset.
>
> [e] *Haitian Zhang, Chang Xu, Xinya Wang, Bingde Liu, Guang Hua, Lei Yu, and Wen Yang.* Detecting every object from events.

---

### Official Review · Reviewer_AXhG · 2025-11-04

**Soundness:** 3
**Presentation:** 3
**Contribution:** 2
**Rating:** 6
**Confidence:** 3

**Summary:**

The paper proposes the SEAL architecture, an open-vocabulary event segmentation architecture based on EventSAM that can segment events on the instance and part level using single nouns or whole sentences as prompts.

The paper adapts EventSAM, an event encoder aligned with the feature space of the Segment Anything Model (SAM). A Multimodal Hierarchical Semantic Guidance (MHSG) module uses SAM to produce 3 levels of masks, which are then used to pool CLIP features and generate captions for each masked region using a pretrained vision-language model (LLaMA). The multilevel captions are used as conditioning to train a feature enhancer network that takes in event features from EventSAM. The enhanced features are pooled with a region-of-interest pooling from the EventSAM masks and are further enhanced using mask tokens from the SAM decoder. The pooled CLIP features are used to test the alignment of the language-enhanced event encodings.

**Strengths:**

Strong performance against baselines

**Weaknesses:**

- Limited novelty: the method is essentially a merger of several pretrained foundation models and their data in a clever way.

 - Limited ablations: qualitative ablations on the Spatial Encoding or Mask Feature Enhancer are missing or not well explained; I did not understand Tables 4 and 5.

**Questions:**

See Weaknesses

---

> ### Author Response · Authors · 2025-11-22
> **Response for Reviewer AXhG (1)**
>
> ## Response for Reviewer `AXhG`
>
> We sincerely appreciate the time that Reviewer `AXhG` devoted to reviewing our paper and recognition of our strong performance compared to the baselines. To further clarify reviewer's understanding to our contributions and experiments, we supplement the additional explanations below.
>
> ---
>
> > `Q1`: "Limited Novelty."
>
> **A**: We thank the reviewer for giving us the opportunity to clairfy our contributions again.
>
> **Motivation & Objective:** Understanding the scene with free-form of language is crucial in diverese tasks such as autonomous driving, AR/VR and robotics which has motivated extensive exploration across multiple modalities including images, point clouds, and LiDAR. **However, related studies on event sensors are scarce or narrowly centered on semantic-level understanding.** To this end, **for the first time**, our study aims to enable real-time event segmentation in multiple levels of granularity based on free-form language or open-vocabulary queries. We define our objective as **Open-Vocabulary Events Instance Segmentation (OV-EIS).**
>
> **Challenges:** We encounter several key challenges in pursuing this goal.
> * There is no existing baseline that addresses OV-EIS, making direct performance comparison infeasible.
> * There is no benchmark to support the evaluation on OV-EIS with multiple class labels, due to the scaricty of events data in the research community.
> * Scarcity and asynchronous nature of events further exacerbates the challenge of understanding the events with human language.
>
> **Contributions:** We address the aforementioned issues with the contributions below:
> * We first define four types of baselines that are grouped into three categories to address OV-EIS, by simply combining the existing RGB models and events-based framework. Here, we focus on proposing the most straightforward approaches to achieve the OV-EIS as these serve as baseline methods. Then, we discuss the limitations of these naive designs in Lines 191-198, proposing the motivation to build our own advanced framework, **SEAL**.
> * Our **SEAL** consists of two key aspects: 1) Real-time OV-EIS model with lightweight architecture which effectively addresses the limitations of proposed baselines with superior performance. 2) Novel events-representation learning framework across multiple levels of granularity which allows the model to understand *semantic*, *instance* and *part*-level events. Unlike the previous events-representation learning framework [a], **we define the events into hierarchical semantic structure** and supervise the SEAL with diverse levels of semantic granularity through our **MHSG module** (*cf*. Lines 249-254). Furthermore, we reveal the two critical issues when we learn hierarchical events representations which are *Dead Masks* and *Semantic Conflict* problems. To mitigate these issues, we introduce Spatial Encoding and Mask Feature Enhancer modules to learn more discriminative events feature space across *part*-level, *instance*-level and *semantic*-level. As reviewer `PR6s` mentioned, **these problems have not been discussed in the prior events-based perception methods.**
> * We further curate the **new** four benchmarks to evaluate the OV-EIS performance in diverse experimental settings (*cf.* Fib. 5b and Lines 939-949). As reviwer `5RxY` mentioned, these new benchmarks carry non-trivial value, particularly for the event-based vision community where datasets remain highly scarce.
>
> **Claims:** We believe our study offers valuable insights toward achieving OV-EIS by: 1) analyzing four types of naive baselines and their limitations, 2) introducing our advanced framework, SEAL to overcome these limitations, and 3) establishing new benchmarks for a comprehensive evaluation of OV-EIS performance. Even though some parts of our work are inspired by existing open-vocabulary techniques, our study **is not limited to naive adaptation of them to the event modality.** We define the novel OV-EIS probem that hasn't been tackeld before, articulate the key challenges associated with it and propose our own solution (SEAL with MHSG module) with extensive analysis (Tab. 1-17). We believe our effort can serve as meaningful pioneer in open-vocabulary events understanding with fine-grained granularity which is important problem but has received very little attention.
>
> ---

---

> ### Author Response · Authors · 2025-11-22
> **Response for Reviewer AXhG (2)**
>
> > `Q2`: "Lack of qualitative ablations on Spatial Encoding and Mask Feature Enhancer."
>
> **A**: We present UMAP [b] visualization of events feature space to show the effectiveness of Spatial Encoding (SE) and Mask Feature Enhancer(MFE) modules in Fig. 4. We detailed more explanations about it to aid the reviewer's understanding below:
>
> **Objective**: The objective of Fig. 4 is to demonstrate the effeciveness of SE and MFE on learning distinct object-level event represenations across multiple semantics.
>
> **Experimental Setting**: We construct two experimental configurations: 1) Events representation leaarned *without* SE and MFE, 2) Event representation learned *with* SE and MFE. Based on these two experimental settings, two sets of predicted event-mask features are collected across all frames in the DSEC-*Part* benchmark. We then visualize the two feature sets using UMAP in 2D space where each feature is color-coded according to its ground-truth label.
>
> **Results**: Fig. 4a presents the events feature space learned *without* SE and MFE which demonstrates two critical issues: **1) *Dead Masks***: Red box in Fig. 4a indicates the events features for dead masks where multiple dots with different colors are clustered together. This is because they are all mapped to zero values in the RoI-Align layer due to their small sizes which is explained in Lines 275-281. **2) *Semantic Conflict***: Purple box in Fig. 4a shows *semantic conflict* problem where green and blue dots are not clearly separated. Since these two colors fail to preserve distinct feature spaces despite their semantic differences, their embeddings collapse and overlap in the same region of the 2D plane. We mitigate these two issues by additionally combining the SE and MFE modules. SE compensates the events feature with rich spatial priors derived from the SAM, resulting in more discriminative feature space. MFE further aggregates both the semantic and spatial priors through additional cross-attention layer. Fig. 4b shows the effectiveness of these two modules where the dots are more distinctly separated by color compared to the Fib. 4a, indicating a more discriminative feature space. Dead masks are also not observed.
>
> **Conclusions (TLDR)**: Object-level events representations exhibit a more discriminative feature space when they are learned with SE and MFE, as shown in Fig. 4b compared to Fig. 4a.
>
> Kindly refer to Lines 1185-1216 of the Supplementary material and Fig. 7 for more detailed explanations about *dead masks* and *semantic conflict* problems.
>
> ---
>
> > `Q3`: "Cannot understand Tab. 4 and Tab. 5."
>
> **A**: We add more explanations about Tab. 4 and Tab. 5 to facilitate reviewer's understanding.
>
> **Tab. 4**: Explanations for Tab. 4 are shown in Lines 459-461 of the main paper. It demonstrates the ablation study on the multi-modal guidance of our MHSG module. As we explain in the Sec. 4.1, our MHSG module provides guidance in two modalities: ***1) Visual-domain Guidance (VG)***: Visual features for each SAM-generated mask are obtained from CLIP [c] as visual guidance. ***2) Text-domal Guidance (TG)***: Rich captions for each mask are generated from pretrained MLLM [d] as text-domain guidance. 3rd row of Tab. 4 shows that combining the guidance from both modalities (VG, TG) yields superior performance across all three benchmarks, compared to the 1st and 2nd rows of Tab. 4 where only a single guidance modality is used.
>
> **Tab. 5**: Explanations for Tab. 5 are shown in Lines 475-480 of the main paper. It provides the quantitative ablations for the components of our SEAL: *Backbone Feature Enhancer*, *Spatial Encoding* and *Mask Feature Enhancer* which are denoted as 'Fusion', 'SE' and 'MFE' in the Tab. 5, respectively. We make following observations: ***1)*** Using SE and MFE together (5th row) yields the best performance across all benchmarks, outperforming configurations where only one of the modules is used (2nd–3rd rows) or where neither module is applied (1st row). ***2)*** Fusion of text knowledge in the backbone feature enhancer consistently leads to performance improvement in all the benchmarks as shown in comparison between 4th and 5th row.
>
> ---
>
> ### References
>
> [a] *Lingdong Kong, Youquan Liu, Lai Xing Ng, Benoit R Cottereau, and Wei Tsang Ooi.* Openess: Event-based semantic scene understanding with open vocabularies.
>
> [b] *Leland McInnes, John Healy, and James Melville.* Umap: Uniform manifold approximation and projection for dimension reduction.
>
> [c] *Alec Radford, Jong Wook Kim, Chris Hallacy, Aditya Ramesh, Gabriel Goh, Sandhini Agarwal, Girish Sastry, Amanda Askell, Pamela Mishkin, Jack Clark, et al.* Learning transferable visual models from natural language supervision.
>
> [d] *Yuqian Yuan, Wentong Li, Jian Liu, Dongqi Tang, Xinjie Luo, Chi Qin, Lei Zhang, and Jianke Zhu.* Osprey: Pixel understanding with visual instruction tuning.

---

> > ### Comment · Reviewer_AXhG · 2025-11-26
> > **Official Comment by Reviewer AXhG**
> >
> > Thanks for addressing my concerns in detail.
> >
> > You made it clear that Open-Vocabulary Event Instance Segmentation is a new and unsolved problem, which you addressed in your work and compared to four baselines that you defined for this purpose. Thus, my novelty concerns have been addressed.
> >
> > Thanks for pointing out and explaining how you ablated the Spatial Encoding or Mask Feature Enhancer.
> >
> > Since my concerns have been addressed, I raised my score to 8.

---

> > > ### Author Response · Authors · 2025-11-26
> > > **Sincerely appreciate to your engagement.**
> > >
> > > We sincerely appreciate the time and effort you invested in evaluating our paper and your active engagement during the rebuttal. As you noted, we believe that SEAL can serve as a **strong starting point** for open-world, fine-grained event understanding, and we are pleased that the reviewer recognized this aspect of our work.
> > >
> > > Best regards,
> > >
> > > The Authors.

---

### Author Response · Authors · 2025-11-22
**General Responses to ALL Reviewers**

## General Responses to ALL Reviewers

We sincerely thank the reviewer for their dedicated time and thoughtful evaluation of our work. We truly enjoyed addressing each of your questions and engaging with your insightful comments. If you have any further questions or would like to continue the discussion, please feel free to raise additional points. We would be happy to elaborate further.

**Note:** In some responses, we include line numbers (e.g., Lines 111–222) to help the reviewer more easily reference the corresponding parts of our paper.

---

> ### Author Response · Authors · 2025-12-03
> **Discussion overview for AC (1)**
>
> Dear AC,
>
> We sincerely appreciate the significant effort required during this critical decision period and thank you and the reviewers for the time and dedication in upholding high review standards.
>
> To support your final assessment, we provide a concise summary of our paper’s key contributions and the points of agreement reached during the rebuttal phase.
>
> ---
>
> ### 1. Strengths recognized by the reviewers
>
> We are encouraged by the reviewers’ recognition of our contributions, including:
> * Reviewer `AXhG` acknowledges the **strong performance** of our SEAL and further admits the **novelty of our work** as the first to address the OV-EIS problem during the discussion.
> * Reviewer `PR6s` highlights SEAL’s **superior performance, fast inference speed, and lightweight model architecture.** Furthermore, reviewer recognizes **our effort to handle the *dead mask* and *semantic conflict* issues** that have not been addressed from the prior works.
> * Reviewer `55Qv` endorses **our four new benchmarks for evaluating OV-EIS** and recognizes **our effort to address the novel OV-EIS problem**.
> * Reviewer `5RxY` underscores the **significant value of our proposed OV-EIS benchmarks and evaluation protocol** to the event-based research community during the discussion. Furthermore, reviewer recognizes our work as **first and effective solution for OV-EIS** in the initial review.
>
> ---
>
> ### 2. Overview of the discussion
>
> We got initial scores of **6, 8, 4, 6**. We summarize the score changes and discussion highlights for each reviewer below:
> * `AXhG`: **Score 6 -> 8 (Accept)**. Initial evaluation was positive while reviewer raised concerns regarding the novelty of our work and the interpretation of our ablation studies. We provided an additional summary of our contributions and further clarification of our extensive ablations. Reviewer recognized our effort and raised the score to 8 by saying ***"my concerns have been addressed, I raised my score to 8."***.
> * `PR6s`: **Score 8 (Accept)**. Initial evaluation was **Accept**. During the discussion, we provided:
>     *  1) Additional analysis of SEAL’s open-vocabulary capability under fine-grained class configurations.
>     *  2) Further clarification of our OV-EIS experimental settings.
>     *  3) Further clarification of the human-free scalability of our MHSG module.
>     *  4) Additional experiment on generic events instance segmentation under the *prompt-free* setting.
>     *  5) Further explanations about inherent characteristics of the event streams.
> * `55Qv`: **Score 4**. The reviewer primarily raised several questions regarding the **1)** motivation behind our SEAL architecture and proposed baselines, **2)** the evaluation pipeline of OV-EIS in our study, **3)** the limitations of our part-level benchmarks, **4)** responsiveness of our SEAL on user-defined language queries, and **5)** potential semantic conflicts across different levels of granularity. Reviewer further mentioned that ***"I would be happy to revise my score if the author addresses these points."***. To address the reviewer’s concerns:
>     * 1) We provided detailed explanations of the motivation behind our SEAL’s architecture and the configurations of the proposed baselines. Extensive ablations in the main paper supports the necessity of each component in our SEAL.
>     * 2) We further clarified our evaluation pipeline and remark its novelty which thoroughly assesses the model's OV-EIS performance in diverse scenarios such as *label granularity* and *semantic granularity* (*cf* Fig. 5b). These diverse scenarios have not been considered in the prior events-based perception works.
>     * 3) We discussed the limitations of our part-level benchmark and clarified that these limitations stem from the inherent characteristics of driving scenes themselves. Nevertheless, we reiterate the pioneering values of our DSEC-*Part* benchmark as the first attempt to address part-level events understanding in the events research community.
>     * 4) Fig. 13 of the main paper demonstrates the qualitative results of our SEAL based on free-form of languages given by the users. Our SEAL successfully outputs related instance masks in response to both noun-level and sentence-level queries.
>     * 5) We provided addtional ablation to show the effectiveness of our SEAL on mitigating the cross-granularity conflict. Our ablation shows that *Spatial Encoding* module effectively alleviates the semantic conflict issue across different granularities.
>
>     In addition, reviewer also asked the technical details for aligning the events with image. We provided concrete explanations with referring the Lines 1048-1057 of the main paper. Specifically, the events are converted to frame-like representation via voxelization process and then aligned pixel-by-pixel with the RGB image.

---

> ### Author Response · Authors · 2025-12-03
> **Discussion overview for AC (2)**
>
> * `5RxY`: **Score 6**. Reviwer raised several concerns on **1)** fairness of experimental settings and **2)** limited semantic diversity of the driving scenes. **3)** The reviewer also provided several suggestions for revising the paper. To address the reviewer's questions:
>     * 1) We detailed our experimental settings to clarify the fairness of our evaluations and supplemented additional experiments for fine-tuned RGB baselines according to reviewer's feedback.
>     * 2) We discussed the inherent characteristics of driving scenes and highlighted our efforts to maximize semantic diversity within this domain. Specifically, our MHSG module enriches the semantics by dividing each scene into multiple levels of granularity and generating detailed text captions for each level.
>     * 3) Finally, we outlined our revision plans according to reviewer's feedback.
>
>   After the first round of discussion, Reviewer recognizes our effort by saying ***"I believe these efforts will greatly benefit other researchers"*** while requesting additional experimental results of our SEAL on scenes beyond driving environments. Reviewer also mentioned that ***"I would be willing to reconsider and potentially increase my score if these results are provided."***. To further satisfy the reviewer’s request, we presented additional qualitative results of our SEAL on a diverse range of indoor and outdoor scenes in Fig. 15 of the main paper.
>
> ---
>
> We believe that our detailed rebuttal and additional ablation studies have thoroughly addressed all of the reviewers’ concerns. We hope this summary supports you in the final decision-making process.
>
> Finally, we woule like to remark our key-contributions below:
> * For the first time, we address the Open-Vocabulary Event Instance Segmentation (OV-EIS) problem, an important yet largely overlooked research challenge.
> * We propose four types of simple baselines that enables OV-EIS and discuss the limitations of them.
> * Given the limitations of the proposed baselines, we introduce SEAL, an advanced OV-EIS framework that achieves the best performance and fastest real-time inference speed with a lightweight architecture.
> * We further propose four new benchmarks to thoroughly evaluate the OV-EIS with diverese scenarios.
>
> Thank you once again for your service to the community.
>
> Best regards,
>
> The Authors.

---

### Meta-Review · Area_Chair_Q7J1 · 2026-01-06

**Summary:**

In the initial phase, reviewers recognized the advantages of the proposed method, including strong performance (Reviewer AXhG, Reviewer PR6s), fast inference (Reviewer PR6s), lightweight model design (Reviewer PR6s), and new benchmarks (Reviewer 55Qv, Reviewer 5RxY).

However, they also raised concerns regarding novelty (Reviewer AXhG), underlying motivation (Reviewer 55Qv), limitations of benchmarks (Reviewer 55Qv), robustness (Reviewer 55Qv), fairness of experimental settings (Reviewer 5RxY), dataset availability (Reviewer 5RxY), and unclear presentation (Reviewer 5RxY).

**Reviewer Concerns:**

Reviewer AXhG explicitly mentioned that his/her concerns have been addressed and raised the score to 8.

Reviewer PR6s has no major concerns and his/her questions have been well addressed.

The authors provided detailed analysis and additional experimental results to address all concerns of Reviewer 55Qv and Reviewer 5RxY. Since the responses are clear and reasonable with new evidence and information, they are sufficient to solve the concerns raised in the initial phase.

**Reviewer Scores:**

Reviewer AXhG raised the score to 8.

Reviewer PR6s will keep the initial score.

Reviewer 55Qv will raise the score to 6.

Reviewer 5RxY will keep the score.

---

### Decision · Program_Chairs · 2026-01-26

Accept (Poster)